# DISENTANGLING REPRESENTATIONS THROUGH MULTI-TASK LEARNING

**Pantelis Vafidis**[*]**, Aman Bhargava**[*]
Computation and Neural Systems
California Institute of Technology
{pvafeidi,abhargav}@caltech.edu

**Antonio Rangel**
Humanities and Social Sciences
California Institute of Technology
arangel@caltech.edu

## ABSTRACT

Intelligent perception and interaction with the world hinges on internal representations that capture its underlying structure ("disentangled" or "abstract" representations). Disentangled representations serve as world models, isolating latent factors of variation in the world along approximately orthogonal directions, thus facilitating feature-based generalization. We provide experimental and theoretical results guaranteeing the emergence of disentangled representations in agents that optimally solve multi-task evidence accumulation classification tasks, canonical in the neuroscience literature. The key conceptual finding is that, by producing accurate multi-task classification estimates, a system implicitly represents a set of coordinates specifying a disentangled representation of the underlying latent state of the data it receives. The theory provides conditions for the emergence of these representations in terms of noise, number of tasks, and evidence accumulation time, when the classification boundaries are affine in the latent space. Surprisingly, the theory also produces closed-form expressions for extracting the disentangled representation from the model's latent state $\mathbf{Z}(t)$. We experimentally validate these predictions in RNNs trained on multi-task classification, which learn disentangled representations in the form of continuous attractors, leading to zero-shot out-of-distribution (OOD) generalization in predicting latent factors. We demonstrate the robustness of our framework across autoregressive architectures, decision boundary geometries and in tasks requiring classification confidence estimation. We find that transformers are particularly suited for disentangling representations, which might explain their unique world understanding abilities. Overall, our framework establishes a formal link between competence at multiple tasks and the formation of disentangled, interpretable world models in both biological and artificial systems, and helps explain why ANNs often arrive at human-interpretable concepts, and how they both may acquire exceptional zero-shot generalization capabilities.

## 1 INTRODUCTION

The ability to construct representations that capture the underlying structure of the world from data, is a hallmark of intelligence. Humans and animals leverage their experiences to construct such faithful representations of the world ("world models", "cognitive maps"), resulting in a near-effortless ability to generalize to new settings (Lake et al., 2015; 2016). Modern foundation models also display emergent out-of-distribution (OOD) generalization abilities, in the form of zero- or few-shot learning (Brown et al., 2020; Pham et al., 2021; Jia et al., 2021; Oquab et al., 2023); however whether artificial systems learn world models remains unclear. Understanding the conditions under which that occurs is bound to lead to better generalizable systems, and explain why artificial systems often converge to human interpretable, aligned representations of the world (Templeton et al., 2024).

A promising direction towards understanding the construction of world models is abstract, or disentangled representations (Higgins et al., 2017; Kim & Mnih, 2018; Johnston & Fusi, 2023). These two concepts are interrelated yet somewhat distinct (see definitions adapted from Ostojic & Fusi (2024) in Appendix A.1). Shortly, an abstract representation of $x_1, \ldots, x_n$ represents each $x_i$ linearly and

---

[*]Equal contribution.

approximately mutually orthogonally. Disentangled representations encode each $x_i$ orthogonally, without the necessity of linearity. Both representations preserve the latent structure present in the world in their geometry by isolating *factors of variation* in the data, which facilitates downstream generalization. When a representation is abstract, a linear decoder (i.e. downstream neuron) trained to discriminate between two categories can readily generalize to stimuli not observed in training, due to the structure of the representation. Furthermore, the more disentangled the representation is, the lower the interference from other variables and hence the better the performance. This corresponds to decomposing a novel stimulus into its familiar features, and performing feature-based generalization. For instance, imagine you are at a grocery store, deciding whether a fruit is ripe or not. If the brain's internal representation of food attributes (ripeness, caloric content, etc.) is disentangled, then learning to perform this task for bananas would lead to zero-shot generalization to other fruit (e.g. mangos, Figure 1a). Crucially, the visual representation of a mango is high-dimensional, non-linear and noisy, making it particularly challenging to extract a low dimensional latent like "ripeness".

Several brain areas including the amygdala, prefrontal cortex and hippocampus encode variables of interest in an abstract format (Saez et al., 2015; Bernardi et al., 2020; Boyle et al., 2022; Nogueira et al., 2023; Courellis et al., 2024). This raises the question of under which conditions do such representations emerge in biological and artificial agents alike. Previous work showed that feedforward neural networks develop abstract representations when trained to multitask (Johnston & Fusi, 2023). However, real-world decisions typically rely on imperfect, noisy information, evolving dynamically over time (Britten et al., 1992; Krajbich et al., 2010). To account for this important feature of the world, we train autoregressive models (RNNs, LSTMs, transformers) to multitask canonical neuroscience tasks involving the accumulation of evidence over noisy streams. The tasks tie closely to Bayesian filtering theory, and should be solved by any agent that deals with a noisy world.

**Contributions.** Our main contributions are the following:

- We prove that any optimal multi-task classifier is guaranteed to learn an abstract representation of the ground truth contained in the noisy measurements in its latent state, if the classification boundary normal vectors span the input space (Appendix B). Furthermore, the representations are guaranteed to be disentangled as the number of tasks $N_{\text{task}}$ greatly exceeds the input dimensionality $D$. Intriguingly, noise in the observations is necessary to guarantee the latent state would compute a disentangled representation of the ground truth.

- We confirm that RNNs trained to multitask develop abstract representations that zero-shot generalize OOD, when $N_{\text{task}} \geq D$, and orthogonal, disentangled representations for greater $N_{\text{task}}$. The computational substrate of these representations is a 2D continuous attractor (Amari, 1977) storing a ground truth estimate in a product space of the latent factors. In addition, the representations are sparse and mixed, attributes of biological neural networks.

- We reproduce these findings in GPT-2 transformers, which generalize better due to them learning disentangled representations already from $N_{\text{task}} \geq D$, confirming their appropriateness for constructing disentangled world models.

- We demonstrate that our setting is robust to a number of manipulations, including correlated inputs, interleaved learning of tasks and free reaction-time tasks canonical in the cognitive neuroscience literature (Britten et al., 1992; Krajbich et al., 2010).

- Finally, we discuss implications for generalizable representation learning in biological and artificial systems, and demonstrate the strong advantage of multi-task learning over previously proposed mechanisms of representation learning in the brain (Mante et al., 2013).

Although framed in the context of canonical neuroscience tasks, our results are general; they apply to any system aggregating noisy evidence. While our experiments focus on supervised multi-task learning for tractability, the theory only assumes **competence** at multiple tasks, thus enabling alternative methods of acquiring such competence, such as self-supervised or unsupervised pre-training.

## 1.1 RELATED WORK

Disentanglement has long been recognized as a promising strategy for generalization (Bengio et al., 2012) (although note Locatello et al. (2019); Montero et al. (2020) for a contrarian view), yet most classic work focuses on feedforward architectures (Higgins et al., 2017; Kim & Mnih, 2018; Whittington et al., 2022; Maziarka et al., 2023). In autoregressive models, Hsu et al. (2017); Li & Mandt

(2018) showed that variational LSTMs disentangle representations of underlying factors in sequential data allowing style transfer; however the underlying representational geometry was not characterised. Other work focuses on fitting RNNs to behavioral data while enforcing disentanglement for interpretability (Dezfouli et al., 2019; Miller et al., 2023). Work on context-dependent decision making has shown that RNNs re-purpose learned representations in a compositional manner when trained in related tasks (Yang et al., 2019; Driscoll et al., 2022); however, the abstractness of the resulting representations was not established. Finally John et al. (2018) show that multitasking results in disentanglement, however unlike us they directly enforce latent factor separation through their adversarial optimization objectives. Our approach is most closely related to weakly supervised disentanglement, without comparing across samples (Shu et al., 2019).

Our work relates to previous work on linear identifiability. Roeder et al. (2021) show that representations of models trained on the same distribution must be linear transformations of each other; yet we go beyond their results to show that abstract representations are **guaranteed** to emerge under moderate conditions, irrespectively of the dimensionality of the input and model architecture. Lachapelle et al. (2023) proved that disentangled representations emerge in feedforward architectures from multitask learning in sparse tasks when a sparsity regularization constraint is placed on the predictors; we place no such constraints and still uncover disentangled representations.

Previous neuroscience-inspired work showed that multitasking feedforward networks learn abstract representations, as quantified by regression generalization (Johnston & Fusi, 2023). We expand upon these findings in several ways. First, we extend the framework to autoregressive architectures (RNNs, LSTMs, transformers) that can update their representations as further information arrives. Second, we prove theorems that guarantee the emergence of abstract representations **in any optimal multitask classifier** if the number of tasks exceeds the input dimensionality $D$, and showcase disentanglement in our trained networks. Third, we rigorously analyze the role of noise in forming disentangled representations, extending the noise-free regime studied in Johnston & Fusi (2023). Finally, we explore a range of values for $D$, providing experimental validation of our theory.

## 2 PROBLEM FORMULATION

**Multi-Task Classification with Evidence-Aggregation:** We study the evidence accumulation multi-task classification paradigm shown in Figure 1b. An agent with latent state $\mathbf{Z}(t)$ receives noisy, non-linearly mapped observations $\{f(\mathbf{X}(t))\}_{t=1}^{T}$ where each $\mathbf{X}(t) = \mathbf{x}^* + \sigma \mathcal{N}(0, I_D)$ is a noisy measurement of unknown ground truth vector $\mathbf{x}^* \in \mathbb{R}^D$ ($x_i^* \sim Uniform(-0.5, 0.5)$), $\mathcal{N}$ being Gaussian noise. The noisy measurements are transformed by an injective observation map $f$, which can be non-linear and high dimensional, representing the wide range of sensory transformations found in real-world scenarios. The agent is tasked with simultaneously solving $N_{task}$ classification problems by accumulating information over time (a canonical neuroscience task (Britten et al., 1992)), each defined by a random linear decision boundary[1] in the ground truth space $\mathbb{R}^D$ i.e.

$$y_i(\mathbf{x}^*) = \begin{cases} 1 & \text{if } \mathbf{c}_i^\top \mathbf{x}^* > b_i \\ 0 & \text{otherwise} \end{cases} \tag{1}$$

where $\mathbf{y}(\mathbf{x}^*) \in \{0, 1\}^{N_{task}}$ represents the $N_{task}$ classifications of $\mathbf{x}^*$, $\{(\mathbf{c}_i, b_i)\}_{i=1}^{N_{task}}$ are the classification boundary normal vectors and offsets, and let $\hat{\mathbf{Y}}(t) = g(\mathbf{Z}(t)) \in [0, 1]^{N_{task}}$ represent the agent's predicted likelihood of $y_i(\mathbf{x}^*) = 1$ over each of the binary classifications $i$ at time $t$. The classification lines reflect criteria based on which decisions will be made. Imagine for example that $x_1$ corresponds to food and $x_2$ to water reward. Depending on the agent's internal state, one takes precedence over the other, and the degree of preference is reflected in the slope of the line.

**Criterion for Disentangled Representation Learning:** We investigate how solving the multi-task classification problem (Figure 1b) leads to agents learning disentangled representations of the latent ground truth $\mathbf{x}^*$ in its internal state $\mathbf{Z}(t)$. Specifically, we ask whether there exists a linear-affine transformation $(\mathbf{A}, \mathbf{b})$ such that $\mathbf{x}^* = \mathbf{A}\mathbf{Z}(t) + \mathbf{b}$. Such a mapping would imply $\mathbf{Z}(t)$ linearly represents $\mathbf{x}^*$. If the rows of $\mathbf{A}$ are approximately orthogonal, the representation is disentangled.

---

[1]Due to the observation map $f$, the tasks may appear non-linear from the perspective of the multi-task classification agent.

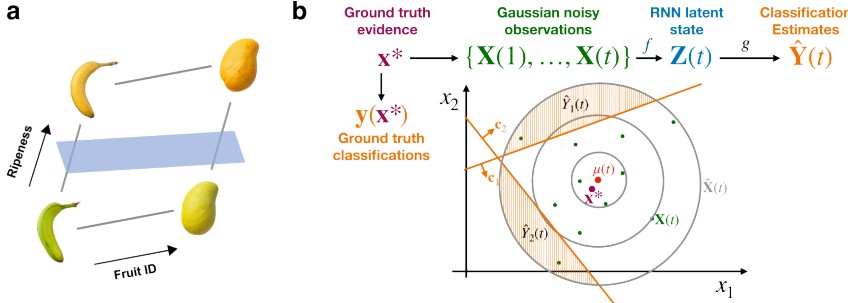

Figure 1: **Disentangled representations and a framework to learn them.** **(a)** A disentangled representation directly lends itself to OOD generalization: a downstream linear decoder that can differentiate ripe from unripe bananas can readily generalize to mangos, even though it has never been trained on mangos. **(b)** Overview of our multi-task classification framework. A ground truth $\mathbf{x}^*$ is sampled and Gaussian noise is added to arrive at observations $\{\mathbf{X}(1), ..., \mathbf{X}(t)\}$. These observations are processed by the filter-based model illustrated graphically in Figure S7, maintaining a latent state $\mathbf{Z}(t)$. The latent state $\mathbf{Z}(t)$ is then used to produce classification outputs $\hat{Y}_1(t)$, $\hat{Y}_2(t)$. Theorem B.6 proves that $\mathbf{Z}(t)$ must encode the optimal estimator of $\mathbf{x}^*$ given the noisy observations, $\mu(t)$.

## 3 THEORETICAL RESULTS

Here we provide conditions and guarantees for the emergence of disentangled representations in optimal multi-task classifiers with latent state $\mathbf{Z}(t)$ in the paradigm described in Section 2 and Figure 1. By "optimal multi-task classifier", we refer to any agent or system whose outputs $\hat{\mathbf{Y}}(t)$ correspond to the correct posterior classification probabilities given the noisy, non-linearly transformed observations; that is, for each task $i = 1, \ldots, N_{task}$

$$\hat{Y}_i(t) = \Pr\left(y_i(\mathbf{x}^*) = 1 \mid f(\mathbf{X}(1)), \ldots, f(\mathbf{X}(t))\right) \tag{2}$$

The notion of optimality allows us to make precise statements about the informational content of the agent's internal state since $\hat{\mathbf{Y}}(t) = g(\mathbf{Z}(t))$. Let $\mathbf{C} \in \mathbb{R}^{N_{task} \times D}$ be a matrix where each row is a decision boundary normal vector. Then

**Theorem 3.1** (Disentangled Representation Theorem). *If $\mathbf{C} \in \mathbb{R}^{N_{task} \times D}$ is a full-rank matrix and $N_{task} \geq D$ and noise $\sigma > 0$, then*

1. *Any optimal estimator of $\mathbf{y}(\mathbf{x}^*)$ **must encode a finite-sample, maximum likelihood estimate** $\mu(t)$ of the ground truth evidence variable $\mathbf{x}^*$ in its latent state $\mathbf{Z}(t)$.*

2. *If the activation function is sigmoid-like, $\mu(t)$ will be **linearly decodable from $\mathbf{Z}(t)$**, thus implying that $\mathbf{Z}(t)$ contains an abstract representation of $\mu(t)$ (Ostojic & Fusi, 2024).*

3. *The representation is guaranteed to be disentangled (orthogonal) as $N_{task} \gg D$ for random decision boundaries.*

*Specifically, $\mu(t)$ is the maximum likelihood estimate (MLE) of $\mathbf{x}^*$ given observations $f(\mathbf{X}(1)), \ldots, f(\mathbf{X}(t))$. A closed-form expression for extracting $\mu(t)$ from $\mathbf{Z}(t)$ if $N_{task} \geq D$ is:*

$$\mu(t) = (\mathbf{C}^\top \mathbf{C})^{-1} \mathbf{C}^\top \left(\tfrac{\sigma}{\sqrt{t}} \Phi^{-1}\big(g(\mathbf{Z}(t))\big) + \mathbf{b}\right) \tag{3}$$

*where $\Phi$ is the CDF of the normal distribution, $\sigma$ is the noise magnitude and $t$ the trial duration. Furthermore, if the activation function $g$ is of the sigmoid family of functions ($\tanh, \text{sigmoid}$), then the term $\Phi^{-1}\big(g(\cdot)\big)$ approximately cancels out, leading to:*

$$\mu(t) \approx \underbrace{\frac{a_g \sigma}{\sqrt{t}}(\mathbf{C}^\top \mathbf{C})^{-1}\mathbf{C}^\top \mathbf{Z}(t)}_{\text{Linear Function of } \mathbf{Z}(t)} + \underbrace{(\mathbf{C}^\top \mathbf{C})^{-1}\mathbf{C}^\top \mathbf{b}}_{\text{Affine Term}} \tag{4}$$

*where we have approximated $a_{\tanh} = \frac{2\sqrt{3}}{\pi}$ for $g = \tanh$ and $a_\sigma = 0.5886$ for $g = \text{sigmoid}$. For Gaussian IID noise, $\mu(t)$ is the sample mean of $\{\mathbf{X}(t)\}_{t=1}^T$, i.e. with non-linearity $f$ removed.*

*Proof.* Point 1 and Equation 3 are proven in Appendix B in Theorem B.6. Point 2 and Equation 4 are proven in Corollary B.8 for `tanh` and Corollary B.9 for sigmoid. Point 3 is proven in Corollary B.10. □

The key conceptual insight in the proof of Theorem 3.1 is that each of the multi-task classification probability estimates $\hat{Y}_i(t)$ represents an estimated projection distance between the MLE $\mu(t)$ and the given classification boundary $(\mathbf{c}_i, b_i)$. Once distances to classification boundaries are recovered, $\mu(t)$ can be inferred if the $N_{\text{task}}$ classification boundaries span the $D$-dimensional space of $\mathbf{x}^*$.

**Robustness of results** While Theorem 3.1 applies to optimal multi-task classifiers, Corollary B.7 shows that a sub-optimal multi-classifier with zero-mean independent errors will represent $\tilde{\mu}(t)$ in state $\mathbf{Z}(t)$ (Equation 3) with residual errors w.r.t. optimal $\mu(t)$ expected to decrease at a rate of approximately $\mathcal{O}(1/\sqrt{N_{task}})$. See Appendix B.4, B.9 for extensions of the theory to anisotropic and non-Gaussian noise distributions (Elliptical, t-distribution, Laplace distributions). The linear approximation for decoding $\mu(t)$ from $\mathbf{Z}(t)$ in Equation 4 is enabled by the remarkable similarity between sigmoid functions and the Gaussian CDF $\Phi$ (Corollary B.8). The sigmoid-like structure of $\Phi$ suggests many similar activations $g$ (e.g., softmax) would exhibit approximate linear decodability.

**More general decision boundaries** Decision boundaries $y_i$ on latents $\mathbf{x}^*$ may appear non-linear in the image of observation map $f$, but Theorem 3.1 applies to linear boundaries $y_i$ on latent space (Eq. 1). Our results extend naturally to smooth manifold decision boundaries through local linearization when the manifold $y_i$'s reach $\tau_i^2$ is much larger than the noise scale $\sigma$. Intriguingly, classification boundary distances are only guaranteed to be recoverable when there is non-zero noise $\sigma > 0$ such that $\hat{Y}_i(t)$ does not saturate to 1 or 0, and thus still carries useful decision boundary distance information (see Lemma B.3) [3]. An intriguing open question is what conditions on manifolds $\{y_i\}_{i=1}^{N_{\text{task}}}$ are necessary and sufficient to preserve the decodability of $\mu(t)$. We leave a complete characterization of representation learning with multiple manifold decision boundaries for future work.

## 4 METHODS

We trained autoregressive models (RNNs, LSTMs, GPT-2 transformers) with latent state $\mathbf{z}(t)$, to output multi-task classifications $\mathbf{y}(\mathbf{x}^*) \in \{0,1\}^{N_{task}}$ given noisy and non-linearly mapped inputs $f(\mathbf{X}(1)), \ldots, f(\mathbf{X}(t))$ (Figure 2). We subsequently trained linear probes $\mathbf{A}$ on $\mathbf{z}(t)$ to estimate $\mathbf{x}^*$, denoted $\hat{\mu}(t) = \mathbf{A}\,\mathbf{z}(t)$. We here focus on leaky RNNs, representing a brain area making decisions; for more details on GPT-2 experiments see Appendix A.6. The networks contain $N_{neu}$ neurons, and their activations $\mathbf{z}(t)$ obey:

$$\tau \dot{\mathbf{z}} = -\mathbf{z} + \left[\,\mathbf{W}_{rec}\,\mathbf{z} + \mathbf{W}_{in}\,\mathbf{x}_{in} + \mathbf{b}\,\right]_+ \quad (5)$$

where $\mathbf{W}_{rec}$ is the recurrent weight matrix, $\mathbf{W}_{in}$ is the matrix carrying the input vector $\mathbf{x}_{in}$, $\mathbf{b}$ is a unit-specific bias vector, $\tau$ is the neuronal time constant, $[.]_+$ is the ReLU applied element-wise and time dependencies have been dropped for brevity. We discretize Equation 5 using the forward Euler method for $T = 20$ timesteps of duration $\Delta t = \tau = 100$ ms, which we find to

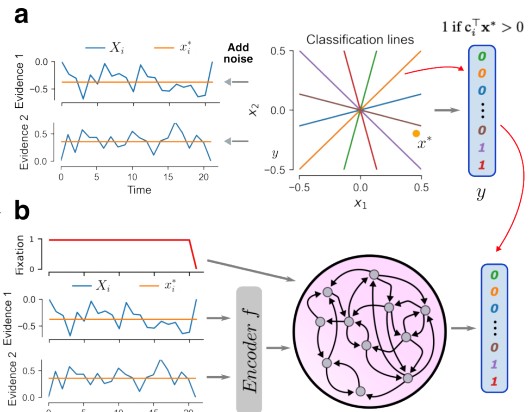

Figure 2: **Data generation and architecture. (a)** For each trial, we sample a ground truth vector $\mathbf{x}^*$, and add IID noise to arrive at $\mathbf{X}(t)$. The task is to report whether $\mathbf{x}^*$ lies above (1) or below (0) each of the classification lines (color matches corresponding boolean variable in $y$), given the noisy and non-linearly transformed samples $f(\mathbf{X}(1)), \ldots, f(\mathbf{X}(t))$. **(b)** Models (RNN depicted) are trained to report the outcome of all the binary classifications in **a** at the end of the trial (indicated by the fixation input turning 0).

---

[2]"Reach": maximum distance at which each point on a manifold has a unique closest point on the manifold
[3]In fact, Equations 3 and 4 do not hold for $\sigma \to 0$, as they were derived by means of Bayesian estimation which assumes the presence of noise.

be stable. The RNN's output $\hat{\mathbf{y}}(t) \in \mathbb{R}^{N_{task}}$ is given by $\hat{\mathbf{y}}(t) = g(\mathbf{W}_{out}\,\mathbf{z}(t))$, where $\mathbf{W}_{out}$ is a readout matrix and $g = $ sigmoid the output activation function applied elementwise. The encoder $f$ is a 3-layer MLP with hidden dimensions $100, 100, 40$ and ReLU non-linearities, and it is randomly initialized and kept fixed during training as it represents a static mapping from latents to observations (observation map). An additional fixation input is directly passed to the hidden layer. It is 1 during the trial and turns 0 at the end of the trial, indicating that the network should report its decisions (Figure 2b). The fixation input is concatenated with $f(\mathbf{X}(t))$ to form $\mathbf{x}_{in}$, and it precludes the RNN from learning a specific timing in its response. We refer to this kind of tasks as fixed reaction-time (RT). The network is trained with a cross-entropy loss and Adam default settings, except learning rate $\eta_0 = 10^{-3}$, to produce the target outputs $\mathbf{y}(\mathbf{x}^*)$. By minimizing loss across trials, the network is incentivized to estimate $\hat{\mathbf{Y}}(t) = \Pr\{y_i(\mathbf{x}^*) = 1\}$. Table S1 summarizes all hyperparameters and their values, which are shared across all architectures.

## 5 EXPERIMENTS

### 5.1 MULTI-TASK LEARNING LEADS TO DISENTANGLED REPRESENTATIONS

We train RNNs to do simultaneous classifications for $N_{\text{task}}$ linear partitions of the latent space for $D = 2$ (Figure 2a, 6 partitions shown). To quantify the disentanglement of the representations after learning, we evaluate regression generalization by training a linear decoder to predict the ground truth $\mathbf{x}^*$ while network weights are frozen. We perform out-of-distribution 4-fold crossvalidation, i.e. train the decoder on 3 out of 4 quadrants and test in the remaining quadrant (Appendix A.2 for details). We also evaluate in-distribution (ID) performance by training the decoder in all quadrants. An example of train and test losses is shown in Figure S11f. We find that the network's OOD and ID generalization performance are excellent (median $r^2 = 0.96, 0.97$ respectively across 5 example networks); therefore the network has learned an abstract representation that zero-shot generalizes OOD. In addition, ID performance increases with the number of tasks $N_{task}$, and the OOD generalization gap decreases (Figure 3a). Performance is identical when choosing a more nonlinear, power-law nonlinearity for the encoder (Appendix A.5). Therefore we conclude that multi-task learning leads to abstract representations in the RNN's hidden layer, when tasks span the latent space.

Since $\mathbf{x}^*$ can be decoded by this representation in unseen (by the decoder) parts of the state space, it follows that the representation can be used to solve **any** task involving the same latent variables, without requiring further pretraining. In other words, to solve any other task we do not need to deal with the denoising and unmixing of the latent factors $x_1$, $x_2$; we would just need to learn the (potentially non-linear) mapping from $x_1$, $x_2$ to task output. Furthermore, the representation scales linearly with input dimensionality $D$ (see Figure 5b). This marks a significant improvement from previously proposed models for representation learning in the brain where one task is executed at a time (Mante et al., 2013; Yang et al., 2019), which scale linearly with $N_{task}$, and exponentially with $D$ (see Appendix A.8 for details). Crucially, these findings are architecture-agnostic: they hold for non-leaky ("vanilla") RNNs, which outperform leaky ones for small $N_{task}$, LSTMs which perform the best, and GPT-2 transformers (details in Appendix A.6) which have excellent performance already from $N_{task} = 2$ (Figure 3b). Note that state-space models have superior asymptotic performance, which is expected due to the nature of the task. We focus on leaky RNNs because of their closer correspondence to biological neurons, which have a membrane voltage that decays over time.

So far we showcased abstractness, but not disentanglement. For disentanglement, it is crucial that the latents lie in orthogonal subspaces. Looking at the angles between the decoders of the latents, we find that they become orthogonal as $N_{\text{task}} \gg D$ for RNNs (Figure 3c), as predicted by our theory. Intriguingly, this already occurs from $N_{\text{task}} \geq D$ in transformers, showcasing their superior ability to separate latent factors. Furthermore, orthogonality strongly correlates with OOD generalization performance, which emphasizes the close link between disentanglement and abstractness: the more orthogonal the representations are, the cleaner the readout of the latent factors by linear decoders.

We further demonstrate the robustness of our setting by showing that abstract representations emerge for different noise distributions and correlated inputs (Appendix A.9), non-linear boundaries (Appendix A.10), and for cognitive neuroscience integrate-to-bound tasks where the agent can make their decision whenever confident enough, not at a fixed time (Krajbich et al., 2010) (Appendix A.11).

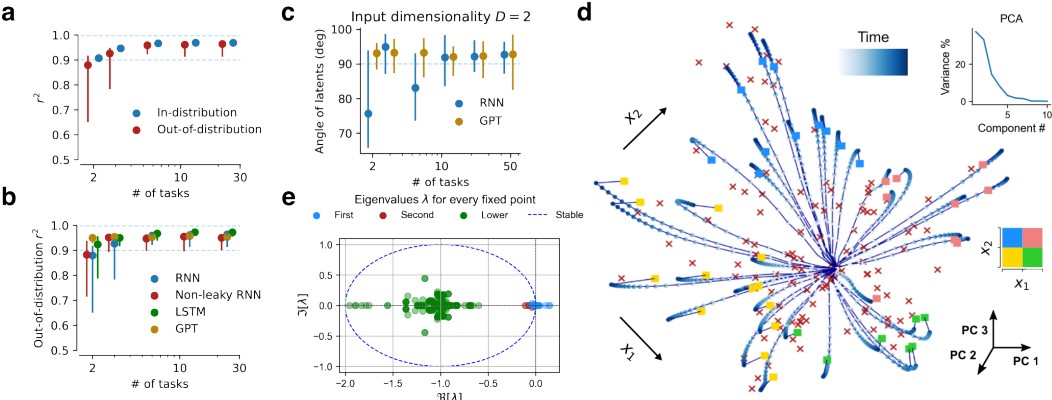

Figure 3: **Learning disentangled representations.** **(a)** ID and OOD generalization performance for networks trained in different number of tasks $N_{task}$. We report the 25, 50 and 75 percentile of $r^2$ for each network size (see Appendix A.2). ID and OOD performance increase with $N_{task}$, and the generalization gap decreases, indicating that the networks have learned abstract representations. **(b)** The results hold for other autoregressive architectures, including LSTMs and GPT-2 transformers. **(c)** Angles between latent factor decoders (see Appendix A.3 for how they were estimated). The angles approach 90 degrees as $N_{task} \gg D$ for RNNs, but already fror $N_{task} \geq D$ for transformers. Remaining errors around 90 degrees are attributed to variability in the linear decoder fits. **(d)** Top 3 PCs of RNN activity ($N_{task} = 24$, $D = 2$), capturing 85% of variance (see inset). Each line is a trial, while color saturation indicates time. All trials start from the center and move outwards, towards the location of $\mathbf{x}^*$ in state space. We color the last timepoint in each trial (squares) according to the quadrant this trial was drawn from. Red x's correspond to attractors (see Appendix A.7). Here we remove input noise so that trajectories can be visualized easier. The network learns a two-dimensional continuous attractor that provides a disentangled representation of the 2D state space. **(e)** Spectral plot resulting from linearizing RNN dynamics around every fixed point (Appendix A.7). First two eigenvalues of the difference system are near 0, while the rest decay much faster, indicating marginal stability across two dimensions for every fixed point, a signature of a 2D continuous attractor.

## 5.2 REPRESENTATIONAL STRUCTURE IN RNNS AND TRANSFORMERS

In this section, we open the black box and investigate the representations learned by the networks, starting with RNNs. Figure 3d shows the top 3 PCs (capturing $\sim 85\%$ of the variance) of network activity after training (final accuracy $\sim 95\%$) for multiple trials, along with the fixed points of network dynamics. To find the fixed points, we follow a standard procedure outlined in Sussillo & Barak (2013) (see Appendix A.7 for details). Looking at Figure 3d the fixed points span the entire two-dimensional manifold that the trials evolve in, which corresponds to a continuous attractor with stable states across a 2D "sheet". Linearizing the dynamics around each fixed point and computing the eigenvalues of the linearized system (Appendix A.7 for details), reveals marginal stability across two eigenvectors, i.e. near-0 eigenvalues which correspond to slow, integration dimensions in network dynamics, therefore confirming the continuousness of the attractor (Figure 3e). This implies that the network can store a short-term memory (Wang, 2001) of the current amount of accumulated evidence in a product space of the latent variables, and update it as further evidence arrives.

Furthermore, compared to the representations after the encoder which are non-linearly mixed, high-dimensional and overlapping (Figure S11a), the representation in Figure 3d looks disentangled as we would expect from the theory and metrics above. Individual trials with noise show how the representation maintains a sense of metric distances in the RNN representation space (Figure S11b). Figure S11c demonstrates how this representation comes about during learning, and Figure S11d that the short-term memory persists when a delay period is included before the decision. Therefore, multi-task learning has led to disentangled, persistent representations of the latent variables. Importantly, and in line with our theory, this only happens when noise is present in the input, which forces the network to learn a notion of distance from classification boundaries (Lemma B.3). Indeed, when the network is trained without input noise, it does not learn a 2D continuous attractor (Figure S11e).

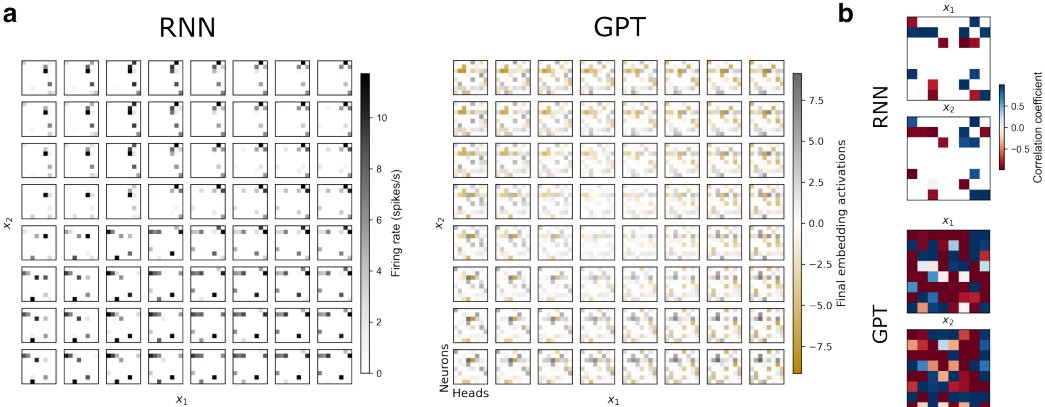

Figure 4: **RNN and GPT representations and relation to latent variables.** **(a)** Hidden layer activations of RNN in Figure 2b (left) and GPT-2 transformer (right), while systematically varying the latent factors $x_1$ and $x_2$ from -0.5 to 0.5. Activations are plotted in 8*8 grids, one for each value of $x_1$ and $x_2$. Each grid contains firing rates for a total of 64 neurons for the RNN, and activations for 8 units for each of the 8 heads from the final embedding of the sequence for GPT-2. **(b)** Correlation coefficient of activations for both models with $x_1$ and $x_2$, respectively.

Finally, we examined RNN and GPT unit activations, and their relation to the latent variables. In Figure 4a we plot activations for all 64 units for both networks, while regularly sampling $x_1$ and $x_2$. RNNs representations are sparse, with only $\sim 10\,\%$ of neurons active at any time, which is in line with sparse coding in the brain (see Appendix A.12 for quantification of sparsity as a function of $N_{task}$, $D$ and RNN architecture). In addition, the average firing rate is $\sim 1$ spike/s, which is surprisingly close to cortical values. Transformers on the other hand, do not have these features, shared by RNNs and their biological counterparts. Furthermore, we find that both networks display mixed selectivity, i.e. neurons are tuned to both variables, which is a known property of cortical neurons (Rigotti et al., 2013) (Figure 4b). This suggests that metrics of disentanglement that assume that individual neurons encode distinct factors of variation (Higgins et al., 2017; Kim & Mnih, 2018; Chen et al., 2018; Eastwood et al., 2022; Hsu et al., 2023) might be insufficient in detecting disentanglement in networks that generalize well. While recent work incorporates such axis-alignment in the definition of disentaglement, our work along with others (Johnston & Fusi, 2023) showcases the advantages of approaching disentanglement from a mixed representations perspective. Importantly, these properties were not imposed during training, nor was there any parameter fine tuning involved; they emerged from task and optimization objectives.

## 5.3 EXPERIMENTS CONFIRM AND EXTEND THEORETICAL PREDICTIONS

Here we expore the relation between the theory in Section 3 and Appendix B and experiments in Section 5 in more depth. First, we wondered why performance saturates in our networks to a high yet non-1 $r^2$. The central limit theorem predicts that the estimate of the ground truth $\mathbf{x}^*$ in any optimal multi-task classifier becomes more accurate with $\sqrt{t}$, providing a theoretical maximum $r^2$ given trial duration $T$ (Appendix A.4). Since the RNNs trained on the free reaction-time (free RT) task in Appendix A.11 are required to output their decision confidence at any time in the trial, we can compute OOD $r^2$ of free RT network predictions at any timepoint $t$, and compare that to the theoretical prediction. Figure 5a shows that indeed the highest RNN $r^2$ falls in the vicinity of or just short of the theoretical maximum. This indicates that RNNs trained with BPTT on these tasks behave like near-optimal multi-task classifiers that create increasingly accurate predictions with time, tightening the relation between our theoretical and experimental results.

An important prediction of our theory is that to learn abstract representations, $N_{task}$ should exceed $D$. To test this, we increase $D$ (adding more inputs to Figure 2b), while varying $N_{task}$. Sampling classification hyperplanes homogeneously (similar to Figure 2a, center) in high-dimensional spaces is non-trivial; therefore we resort to randomly sampling them. Figure 5b shows OOD generalization

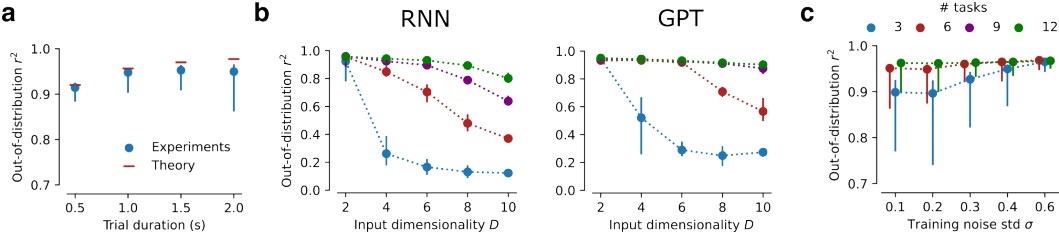

Figure 5: **Experiments confirm theoretical predictions.** (a) OOD $r^2$ for free RT RNN required to report its estimate of $\mathbf{x}^*$ at different times (see Appendix A.11 for details, $N_{task} = 24$, $D = 2$). Maximum network $r^2$ matches optimal multi-task classifier theory predictions (Equation 7 in Appendix A.4). (b) OOD $r^2$ as a function of input dimensionality $D$ and number of tasks $N_{task}$. Good values of $r^2$ are obtained when $N_{task} \geq D$, especially for GPT models, confirming our theoretical results. (c) Increasing amounts of noise in pretraining results in better OOD generalization ($D = 2$).

performance for various combinations of $D$ and $N_{task}$. We observe that performance is bad when the $N_{task} < D$, but it increases when $N_{task} \geq D$. For RNNs, this increase is abrupt for smaller $D$ and more gradual for higher, which is in line with remarks by Johnston & Fusi (2023) that it is easier to learn abstract representations when $D$ is high. Transformers on the other hand display higher generalization performance than RNNs, and always perform almost perfectly when $N_{task} \geq D$, demonstrating their superior performance in learning abstract representations. Looking at the angles between latents for higher $D$ (Figure S12), we find that transformers have excellent disentanglement as long as $N_{task} \geq D$, which might explain their superior generalization performance to RNNs for lower $N_{task}$. These results, together with Figure 3c demonstrate the superior ability of transformers in disentangling latent factors. Overall, our findings confirm our theory that abstract representations emerge when $N_{task} \geq D$, and even go beyond to suggest that disentangled representations emerge earlier than the theoretical condition $N_{task} \gg D$, as long as the architecture is appropriate. These results are remarkable, especially for high $D$, because they go against our intuition that $N_{task}$ should scale exponentially with $D$ to fill up the space adequately; instead it need only scale linearly.

**Importance of noise for generalization** Our theory and experiments provide insight on the importance of noise for developing efficient, abstract representations (Figure S11e). The closer to a classification boundary the ground truth $\mathbf{x}^*$ is, the more likely noise will cross over the boundary. Since, as our theory shows, any optimal multi-task classifier has to estimate $\Pr\{y_i(\mathbf{x}^*) = 1\}$, and said probability directly relates to the actual distance from the boundary, it follows that noise allows the model to learn distances from boundaries (Lemma B.3)), leading to efficient localization. We reasoned that additional noise might be even more beneficial, as it would allow more accurate estimation of $\Pr\{y_i(\mathbf{x}^*) = 1\}$, especially when $\mathbf{x}^*$ is far from the boundary. To test this, we increase noise strength $\sigma$ when pretraining RNNs, while testing with the same $\sigma = 0.2$. Indeed, increasing amounts of noise consistently result in better OOD generalization (Figure 5c). This benefit comes for smaller numbers of tasks, allowing us to consider less supervised tasks (e.g. 3), train on them with more noise, and achieve the same performance as more tasks (e.g. 12). So even though networks with more noise perform worse in pretraining (low 90%s classification accuracy), they learn more abstract representations. These findings are highly non-trivial, and have informed our thinking about generalization and inherent variability of the underlying latent factors.

## 6 DISCUSSION

In this work, we proved that disentangled, generalizable representations **must** emerge in agents optimally solving multi-task evidence accumulation tasks canonical in the neuroscience literature. We also conducted experiments in a suite of autoregressive models (RNNs, LSTMs, transformers) which confirmed all of the main theoretical predictions. A key takeaway is that transformers more readily disentangle representations, which may explain their unique world understanding abilities. Here we discuss the broader impact of this work for representation learning and neuroscience alike. Limitations of this study and how it can be extended in the future are discussed in Appendix A.15.

## 6.1 Implications for representation learning

**Topology-preserving representation learning**   Our work has profound implications for learning representations that inherit the topological structure of the world. We prove this naturally happens as long as there are enough tasks to uniquely identify the location of $\mathbf{x}^*$. Crucially, the constraints from different tasks should be placed simultaneously on the representation, which explains why representations from context-dependent computation (Mante et al., 2013) are typically not disentangled.

**Representational alignment across individuals**   Our results provide a new perspective on the Platonic representation hypothesis (Huh et al., 2024), which suggests that the convergence in deep neural network representations is driven by a shared statistical model of reality, like Plato's concept of an ideal "Platonic" reality. Theorem 3.1 suggests that the key factor driving convergence is the diversity and comprehensiveness of the tasks being learned. As long as individuals are faced with similar day-to-day tasks that collectively span the space of the underlying data representation, convergence to a shared, reality-aligned representation can occur. This could explain why for example modern LLMs come to encode high-level, human-interpretable concepts (Templeton et al., 2024).

**Manifold hypothesis**   While our problem is framed in terms of arbitrary injective observation map $f$, the formulation encompasses many scenarios relevant to the manifold hypothesis (Fefferman et al., 2013). The function $f$ can represent a smooth manifold embedded in a high dimensional space, directly modelling the manifold hypothesis of deep learning. In neuroscience, $f$ could be a non-linear encoding of stimuli in a neural population response, connecting our work to neural manifold research (Langdon et al., 2023). By developing and testing theoretical guarantees for the emergence of disentangled representations in this multi-task problem formulation, we provide insight on how neural networks can inherently discover and linearize low-dimensional manifolds within high-dimensional, non-linear observations, enhancing our understanding of how complex data structures are captured and represented in deep learning models and biological systems alike.

**Interplay between number of tasks and fine-grainness of representations**   Finally, the theorem and experimental results presented here are not a one-way-street from dimensionality $D$ of the latents to how many tasks $N_{\text{task}}$ are required to uncover them. Instead, there is a fundamental interplay between richness of tasks performed and detail of the representation learned. In a high-dimensional world, the richness of the tasks at hand directly affects the dimensionality $D$ of the latents that can be extracted, allowing for "ground truths" $\mathbf{x}^*$ at different levels of granularity to be explored. The richer the label information available, the more fine-grained the resulting world model will be.

## 6.2 Implications for neuroscience

The brain encodes variables of interest in a disentangled format, in processes as disparate as memory (Boyle et al., 2022), emotion (Saez et al., 2015), and decision making (Bongioanni et al., 2021). Furthermore, performance in tasks has been shown to degrade once abstract representations collapse (Saez et al., 2015), supporting their role in guiding generalizable behavior. Our findings put forth **parallel processing** as a unifying mechanism for generalization in brains. The cortex, with its massively parallel architecture (Markram et al., 2015; Hawkins et al., 2019), is a prime candidate area for the construction of disentangled, generalizable world models. Another candidate area is the thalamus; it is posited that thalamocortical loops operate in parallel, and combined with internal state-dependent mechanisms lead to state-dependent action selection (e.g. prioritizing water when thirsty), while evidence integration occurs in corticostriatal circuits (Rubin et al., 2020). The representations discovered here (continuous attractors, CANs) have been widely found in the brain when solving similar tasks, highlighting their role as a general computational substrate for cognitive functions in the brain (for relations of our work to the neuroscience literature, see Appendix A.13). Notably, the receipt of rich supervisory signals from the environment is not a requirement for our setting, as it can leverage the output of previously learned tasks (see Appendix A.14 on the biological plausibility of multi-task learning).

The algorithmic efficiency of multi-task learning compared to alternatives ("context-dependent computation"( Mante et al. (2013), Appendix A.8)), makes us think that it is no coincidence that the cortex can support parallel processing; all the pieces are there, and we feel that the brain has to leverage this feature to construct faithful models of the world, as it does.

## REPRODUCIBILITY STATEMENT

All code used to generate the results can be found in https://github.com/panvaf/DisentangleRes. The experiments are seeded, ensuring exact reproducibility of results. Five networks have been trained for every configuration shown, to provide sufficient statistics to support our conclusions. To ensure full clarity of the theoretical results in the main text, a full proof is provided in Appendix B.

## AUTHOR CONTRIBUTIONS

P.V. conceived the project, wrote the code, performed the experiments and wrote the manuscript. A.B. developed the theory, wrote the theory sections and parts of the manuscript. A.R. supervised the research and provided funding.

## ACKNOWLEDGMENTS

P.V. would like to thank the Onassis Foundation and A.R. the NOMIS Foundation for funding. A.B. thanks the NIH PTQN program for funding. No competing interests to declare. We would like to thank Yisong Yue for early discussions, Aiden Rosebush for discussions of proof methods and Stefano Fusi for early feedback.

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

# A  SUPPLEMENTARY MATERIALS

## A.1  DEFINITIONS

In the main text we used the terms "abstractness" and "disentanglement". Since there exists some ambivalence about their meaning in the literature, we would like to strictly define them here. We will be using definitions adapted from Ostojic & Fusi (2024):

- An abstract representation of latent factors $x_1, \ldots, x_n$ represents each $x_i$ linearly and approximately mutually orthogonally. Thus, abstractness ensures a simple linear map can decode each $x_i$ regardless of variation in $x_{j \neq i}$.
- Disentangled representations of $x_1, \ldots, x_n$ encode each $x_i$ orthogonally, without the necessity of linearity.

Note that under this definition, axis-alignment is not a requirement for disentanglement (also see Higgins et al. (2018)). Our work suggests that the computer science and neuroscience communities should adopt this broader definition of disentanglement, because otherwise we might be missing cases where the factors are not axis-aligned, but they are still orthogonal and can still be isolated by a linear decoder. Our argument is that there is nothing special about individual factors being encoded by individual neurons. Rather, we think that allowing for mixed representations within the definition of disentanglement leads to a more holistic view of disentanglement. A contribution of this work, along with others (Johnston & Fusi, 2023), is to bring this argument to the forefront.

## A.2  QUANTIFICATION OF GENERALIZATION PERFORMANCE

To assess OOD generalization performance, we keep the trained networks fixed and train a linear decoder $\mathbf{A}$ to predict the ground truth $\mathbf{x}^*$ from network activity at the end of the trial. We train the decoder in 3 out of 4 quadrants and test OOD in the remaining quadrant, repeating this process 5 times for each quadrant, which results in a total of 20 OOD $r^2$ values for each network. To account for randomness in initialization and sythetic generation of datasets, we train 5 networks for each combination of number of tasks $N_{\text{task}}$ and dimensionality $D$, resulting in a total of 100 OOD $r^2$ values for each pair of $(N_{\text{task}}, D)$. We report the 25, 50 (median) and 75 percentiles of those values in Figure 3a,b and throughout the text. For ID generalization performance, we train on all quadrants and test in one quadrant at a time. For input dimensionality $D > 2$, we keep the same logic by choosing every 4-th quadrant to be sampled only in testing, repeating the process for every $mod\,4$ group of quadrants.

## A.3  ESTIMATION OF ANGLES BETWEEN LATENT FACTORS

To estimate the angles between latent factors in the representation, we obtain the normal vectors of the decoders $\mathbf{A}$ for each of the latents, and compute pairwise angles for all of them. To account for variability in the decoder fits, we repeat the decoder fit 5 times for each out-of-distribution region (see Appendix A.2 for details). We also repeat this process across 5 trained networks for each combination of $(N_{\text{task}}, D)$, and report the 25, 50 (median) and 75 percentiles of all values for each $(N_{\text{task}}, D)$ combination in Figure 3c and Figure S12.

## A.4  DERIVATION OF THEORETICAL $r^2$ FOR OPTIMAL MULTI-TASK CLASSIFIERS

Here we derive the theoretical $r^2$ for the estimation of ground truth $\mathbf{x}^*$ from noisy data for a discrete time optimal multi-task classifier at time $t$. $r^2$ is defined as:

$$r^2 = 1 - \frac{MSE(\mathbf{x}^*, \mu)}{Var(\mathbf{x}^*)} \tag{6}$$

where $\mu$, the mean of $\mathbf{X}(1), \ldots, \mathbf{X}(t)$, is the prediction of the multi-task classifier (see Appendix B). The optimal estimator of $\mathbf{x}^*$ given observations $\mathbf{X}(1), \ldots, \mathbf{X}(t)$ is denoted $\hat{\mathbf{X}}(t) \sim \mathcal{N}(\mu(t), t^{-1}\sigma^2 I_D)$ where $\sigma$ is the noise strength. Note that $\mu(t) \to \mathbf{x}^*$ as $t \to \infty$ by the central limit theorem, and $\mu(t)$ is the optimal estimator of $\mathbf{x}^*$ given Gaussian-noised observations. Since

the dimensions in both noise and ground truth are independent, we can focus on one dimension at a time i.e.:

$$r^2 = 1 - \frac{MSE(x_i^*, \mu_i(t))}{Var(x_i^*)} = 1 - \frac{\mathbb{E}[(x_i^* - x_i^* + \mathcal{N}(0, t^{-1}\sigma^2))^2]}{Var(x_i^*)} = 1 - \frac{\sigma^2}{t\,Var(x_i^*)}. \quad (7)$$

Remembering that $x_i^* \sim Uniform(-0.5, 0.5)$ it follows that $Var(x_i^*) = \frac{2}{3}0.5^3$, and replacing $\sigma = 0.2$ from Table S1 we arrive to $r^2 = 1 - \frac{0.48}{T}$ for given trial duration $T$ which we compare to RNN OOD generalization performance in Figure 5a.

### A.5 MORE NONLINEAR ENCODING

In the main text we used ReLU nonlinearities for the encoder $f$. Here we extend our findings to more nonlinear observation maps which are likely to be encountered in the real world, e.g. ones with power-law nonlinearities. Recent work showed that the choice of activation function can influence the geometry of representations (Alleman et al., 2024). Interestingly, we find that replacing ReLUs in the encoder with a quadratic nonlinearity results in virtually identical OOD generalization performance compared to Figure 3a (Figure S1). Therefore we conclude that our setting is robust to the choice of encoder nonlinearity, even when the nonlinearity is not injective, going beyond our theoretical proofs (Appendix B.3).

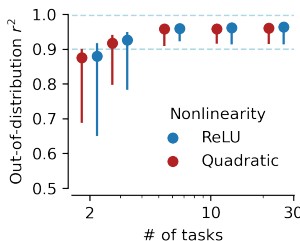

Figure S1: OOD generalization is robust to choice of encoder nonlinearity.

### A.6 GPT-2 EXPERIMENTS

We train GPT-2 causal transformers with $d_{model} = N_{neu} = 64$, $N_{layer} = 1$, $N_{head} = 8$ in the multi-task classification task of the main text. The networks receive continuous, noisy and non-linearly mapped inputs $f(\mathbf{X}(1)), \ldots, f(\mathbf{X}(t))$, and should output multi-task classifications $\mathbf{y}(\mathbf{x}^*) \in \{0, 1\}^{N_{task}}$. The output of the network is $\hat{\mathbf{Y}}(t) := g(\mathbf{Z}(t))$, where $\mathbf{Z}(t)$ is the last embedding of the sequence in the last layer and $g = $ sigmoid. Since the input is continuous, we omit the tokenization and embedding steps, and project the input directly to the hidden state with a linear map. Furthermore, since the inputs are IID, we do not include positional encodings. The networks are trained with binary cross-entropy loss for $N_{batch} = 2 * 10^4$ batches, while the rest of the parameters are identical to the fixed RT networks of the main text (Table S1).

### A.7 FINDING FIXED POINTS AND LINEARIZATION OF DYNAMICS

To find approximate fixed points of RNN dynamics after training, we follow a standard procedure outlined in Sussillo & Barak (2013). Specifically, we keep network weights fixed, provide no inputs to the network, and instead optimize over hidden activity. Specifically, we penalize any changes in the hidden activity, motivating the network to find stable states of the dynamics in the absence of input, i.e. attractors of the dynamics. This process finds all states of accumulated evidence that can be stored in this network as short-term memory. Network dynamics could then leverage these states to maintain and update the internal representation of the ground truth $\mathbf{x}^*$ on a single trial level, and drive downstream decisions.

Then for every approximate fixed point $\mathbf{z}^f$, we linearize RNN dynamics around it and estimate the eigenmodes which describe how the system behaves in a small region $\delta\mathbf{z}$ around $\mathbf{z}^f$. Specifically, following Sussillo & Barak (2013); Mante et al. (2013) we take the difference system $\delta\mathbf{z}(t + t_0) = \mathbf{z}(t + t_0) - \mathbf{z}(t_0)$ and linearize it, i.e.

$$\dot{\delta\mathbf{z}} = \mathbf{F}'(\mathbf{z}^f)\,\delta\mathbf{z} \quad (8)$$

where $\dot{\mathbf{z}} = \mathbf{F}(\mathbf{z})$ is the function describing the RNN dynamics and $\mathbf{M} \equiv \mathbf{F}'(\mathbf{z}^f)$ is its Jacobian at $\mathbf{z}^f$. To estimate $\mathbf{F}'(\mathbf{z}^f)$, we let network dynamics run in the absence of inputs for one time step $\Delta t$ starting from $\mathbf{z}^f$, i.e. $\delta\mathbf{z}(\Delta t) = \mathbf{z}(\Delta t) - \mathbf{z}^f$, and autodifferentiate $\delta\mathbf{z}(\Delta t)$. We then perform eigendecomposition of $\mathbf{M}$ and report the eigenvalues around each approximate fixed point.

Eigenvalues near 0 indicate that the difference system $\delta\mathbf{z}(t) = \mathbf{z}(t) - \mathbf{z}^f$ changes slowly over time, i.e. they correspond to "slow" dimensions in network dynamics which can integrate inputs and maintain them over time (continuous attractors) (Amari, 1977; Mante et al., 2013).

## A.8 State-space efficiency of context-dependent computation

The workhorse model for computational neuroscience has traditionally been context-dependent computation, where tasks are carried out one at a time and task identity is cued to the RNN by a one-hot vector Mante et al. (2013). However, this approach can be algorithmically inefficient, because as we show here it scales linearly with the number of tasks $N_{task}$, and exponentially with input dimensionality $D$. This is because context-dependent computation utilizes different parts of the state space for different tasks, and the resulting representations collapse to what is minimally required for each task (also see (Mante et al., 2013; Yang & Wang, 2020)). This can be detrimental for brains, which need to pack a lot of computation within a large yet limited neural substrate. In contrast, abstract representations are general, compact (Ma et al., 2022), can be used for **any** downstream task involving the same variables, scale linearly with $D$, and as we show readily emerge from relatively simple tasks.

To compare context-dependent decision making, where one task is performed at a time (Mante et al., 2013), to multitasking, in terms of state-space usage efficiency, we train RNNs to perform context-dependent decisions on the same tasks encountered in the main text. Compared to the network in the main text (Figure 2b), the RNN now also receives a one-hot task rule vector indicating the current task, and it outputs the decision for that task only (Figure S2b). We have also omitted the non-linear encoder, making the tasks easier. We train the RNN for two tasks, one task at a time, in interleaved batches (Figure S2a). In one task, the RNN is required to decide which stream has more evidence, and at the other whether the sum of evidence across streams exceeds a certain decision threshold (here 0).

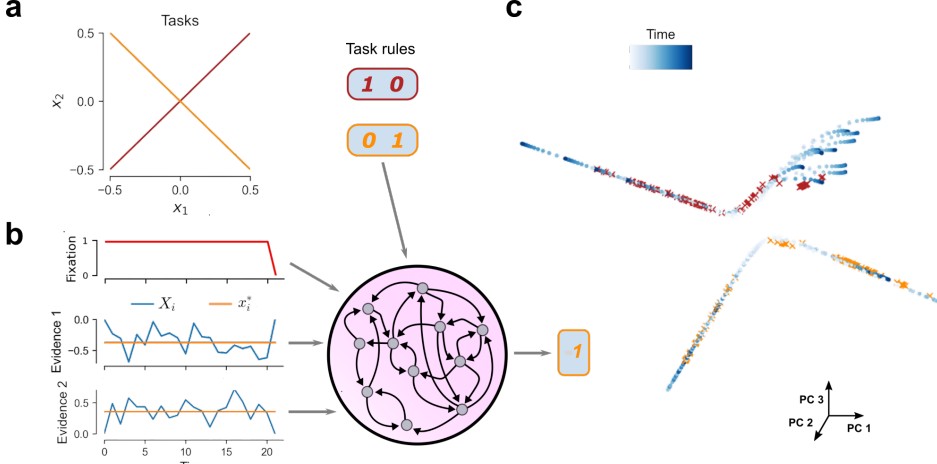

Figure S2: **Representations in an RNN trained in context-dependent decision making.** **(a)** We trained RNNs for two classification tasks: two-alternative forced choice (where the decision boundary is the $x_1 = x_2$ line) and evidence integration (corresponding to the $x_1 = -x_2$ line). Each task corresponds to a different one-hot task rule vector. **(b)** Network architecture. In addition to the inputs in Figure 2b, the network also a one-hot vector indicating the current task. **(c)**, Top 3 PCs of RNN activity example trials (40 in total). The task rule biases the network towards learning separate solutions in different parts of the state space for different tasks, in the form of separate line attractors; red x's for two-alternative forced choice and orange x's for evidence integration.

We find that in this setting the network is not incentivized to learn abstract representations. Instead a separate line attractor is present in the dynamics for each task (red and orange x's in Figure S2c); one of them is presumably tracking the difference of evidence (similar to Yang & Wang (2020) but

for independent evidence streams) and the other the sum of evidence. That is to say, the task rule biases the network to learn different computations in separate regions of state space, as in Mante et al. (2013). As a result, the 2D latent space has collapsed and cannot be decoded from network activity; therefore generalization to any task that involves these two variables is not possible.

It follows that the network can be inefficient in terms of state space usage, because instead of compressing all of its activity around the same region, it spreads it across multiple regions, one for each task, which scales badly (linearly with the number of tasks $N_{task}$ and exponentially with input dimensionality $D$). To demonstrate the latter, imagine a family of tasks with classification boundaries of the form $\oplus x_1 \oplus x_2 \oplus .. \oplus x_D = 0$, where $\oplus \in \{+1, -1, 0\}$ is an operator indicating contribution with a positive sign, negative sign or absence of contribution for a factor to a specific task, respectively. As just shown, each one of this tasks will require its own line attractor, resulting in a total of $3^D$ line attractors lying in separate regions of the state space, just for this simple family of tasks. As mentioned in the main text, such inefficiency can be detrimental for brains, which need to pack a lot of computation within a large yet limited neural substrate. Compare that to multitasking, which builds representations that can serve any task that involves the same latent variables, scaling linearly with $D$ (as we saw that we only need $N_{task} \geq D$ to learn them). Note that context-dependent computation can still be efficient, if tasks have a compositional structure where the solution for one task is part of the solution for another (Yang et al., 2019; Driscoll et al., 2022); in this case, representations developed for the former can act as a scaffold for representations for the latter.

Overall, we believe that multitasking may present a paradigm swift for generalizable representation learning in biological and artificial systems alike. That is not to say that context-dependent representations are not useful; they are great at leveraging the compositional structure of tasks (Yang et al., 2019; Driscoll et al., 2022), but tend to overfit to the specifics of the task, while multitask representations serve as world models applicable to various scenarios. Both types of representations are likely to be found in the brain. One possibility is that context-dependent representations may emerge as a first quick solution to a task, while disentangled representations come about with more experience or when more tasks are performed over time to support better generalization.

### A.9 ROBUSTNESS TO OTHER NOISE DISTRIBUTIONS AND CORRELATED INPUTS

We here show that our setting is robust to Gaussian anisotropic and autocorrelated noise, and other asymmetric distributions of noise (Gumbel) whose CDF no longer matches the sigmoid functions in shape, with almost no drop in performance, and correlated inputs. This demonstrates that abstract representations are also learned outside of the specific assumptions made by our theory.

Starting from anisotropic noise, we observe that doubling the standard deviation of noise across one dimension ($\sigma = 0.4$) does not result in a reduction in OOD generalization performance (median $r^2 = 0.96$ for $N_{task} = 24$, $D = 2$). This is in line with our theory that can be extended to cover anisotropic noise (Lemma B.11). Non-IID noise should not be a problem either, since we are training our network for many samples and the effects of correlations will cancel out over long ensembles. Indeed, we find that including autocorrelated AR(1) noise with an AR coefficient of 0.7 results in only minor reduction in performance (median $r^2 = 0.95$ for $N_{task} = 24$, $D = 2$).

We were also curious to see the impact of correlated inputs. A problem with high correlations is that they render parts of the state space virtually invisible to the network (Figure S3a). Surprisingly, OOD generalization performance is very weakly affected by input correlations, even though the state space is sampled uniformly in test (Figure S3b). The behavior is highly non-linear: performance is great until $\rho = 0.97$, but for perfectly correlated inputs ($\rho = 1$), the performance drop is sharp.

Finally, our theory pointed out sigmoid functions as a choice for activation function because of their close resemblance to the Gaussian CDF, resulting in the best OOD $r^2$. Still we find that for an asymmetric noise distribution (Gumbel) whose CDF does not match sigmoid functions well, there is only a slight drop in performance (median $r^2 = 0.95$ from 0.96). Therefore the conditions for the activation function/CDF should be quite lax; any monotonic bijective function should work with small performance drop. This drop in performance is because the representation would be "stretched out" and "compressed" in a non-linear manner in regions where there is discrepancy between the noise CDF and the activation function. But this nonlinear squishing (determined by the term $\Phi^{-1}(g(\mathbf{Z}(t)))$) would be geometrically inoffensive — no cutting or gluing together would be required to map from $\mathbf{Z}(t)$ to a linear representation of $\mu(t)$. As a result, the representations would

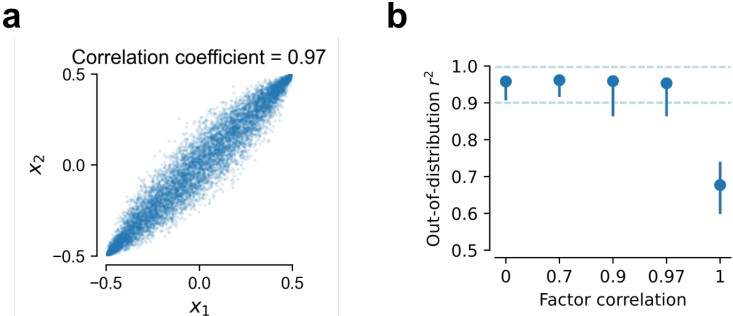

Figure S3: **Disentanglement and factor correlations.** **(a)** We introduce strong correlations in the latent factors, rendering parts of the state space virtually invisible to the network during pre-training (trained for a total of 24 classification tasks). **(b)** Despite that, generalization performance is excellent for correlations very close to 1. Once the factors are perfectly correlated, performance drops significantly. This implies that the network can learn an abstract representation from correlated inputs, as long as there is some signal about the factors independently. This finding goes beyond (Johnston & Fusi, 2023) to show that the multi-task learning setting allows OOD generalization when the distribution during training the RNN itself is vastly different that the one during testing.

remain approximately linearly decodable. Monotonicity and bijectivity are quite mild assumptions for the activation function used by neurons in the brain.

## A.10 NONLINEAR CLASSIFICATION BOUNDARIES AND INTERLEAVED LEARNING

In the main text we trained networks on linear classification boundaries. The tasks are still non-linear, since the encoder renders these boundaries non-linear to the network. However, there are cases where the latent factors themselves might need to be combined non-linearly, to make decisions. For instance, if the two factors represent the amount and probability of reward respectively, an agent needs to multiply the two and decide whether the expected value exceeds a certain (metabolic) cost $\gamma$ of performing an action to obtain said reward. Figure S4a shows the classification lines for the multiplicative task, where the network should decide whether the ground truth $\mathbf{x}^*$ lies above or below the curve $x_1 x_2 = \gamma$, for multiple values of $\gamma$. This family of tasks is not covered by Theorem B.6, because they violate the injectivity condition. Hence, we wondered how the representation would look like if the network was trained on both the linear and multiplicative boundaries, as animals do.

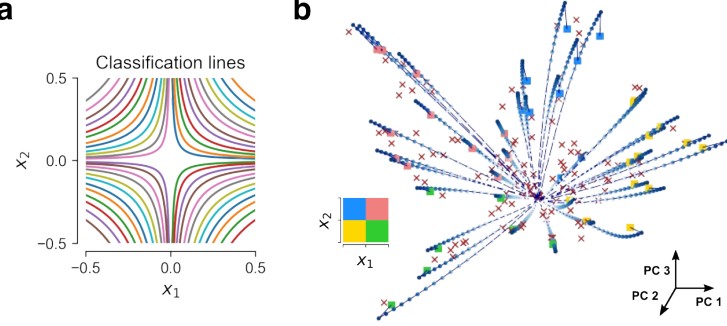

Figure S4: **Interleaved learning of linear and non-linear boundaries.** **(a)** Classification lines for the multiplicative task. There is a total of 48 classification lines, 12 per quadrant. **(b)** The network learns an abstract representation when trained for the linear and multiplicative boundaries in interleaved batches.

For that we perform interleaved training of both tasks (i.e. train in batches sampled from one of the tasks at a time), a setting where neural networks excel at, compared to humans who excel at blocked training, where tasks are learned sequentially (but see Flesch et al. (2022)). Figure S4b shows that the network still learns an abstract, two-dimensional continuous attractor. OOD generalization for this network is excellent, and almost identical to ID performance (median $r^2 = 0.94, 0.97$ respectively). Overall, we conclude that our framework extends to interleaved learning of a mixture of linear and nonlinear boundaries, which better reflects the challenges encountered by agents in the real world. Note that during interleaved training, linear and non-linear tasks are not performed simultaneously; yet they are in immediate succession which can also place pressure to the network to gradually learn representations that satisfy all tasks. The relation between multi-task and interleaved learning is a promising topic for future research.

### A.11 ABSTRACT REPRESENTATIONS ARE LEARNED FOR A FREE REACTION TIME, INTEGRATE TO BOUND TASK

In the main text we trained networks to produce a response at the end of the trial. However, in many situations agents are free to make a decision whenever they are certain enough. Therefore, we here seek to extend our framework to free reaction time (RT) decisions. A canonical model accounting for choices and reaction times in humans and animals is the drift-diffusion model (Krajbich et al., 2010; Brunton et al., 2013). It is composed of an accumulator that integrates noisy evidence over time, until a certain amount of certainty, represented by a bound, is reached, triggering a decision. In the linear classification task setting, the accumulated amount of evidence at time $t$ for a line with slope $\alpha$, $A_\alpha(t)$ is given by:

$$A_\alpha(t) = A_\alpha(t-1) + X_1(t) - \alpha \, X_2(t) \qquad (9)$$

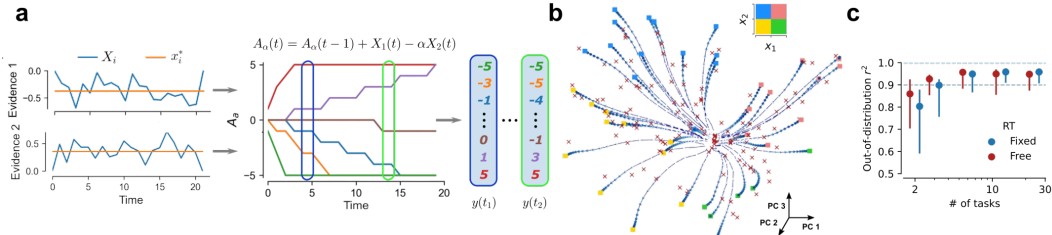

Figure S5: **Free reaction time task.** **(a)** Data generating process. Every classification line from Figure 2a now corresponds to an accumulator (see corresponding colors), and the desired output for the RNN is the accumulator values for the entire trial. The accumulator is quantized to integer values between $\pm 5$. **(b)** Representation for RNN trained on free reaction time task. The network learns a two-dimensional continuous attractor, similar to Figure 3d. A 3D rotating figure to better visualize this representation is provided in the Supplementary Material. **(c)** OOD generalization performance for the free reaction time (RT) task. Free RT outperforms fixed RT for a small number of tasks.

Intuitively, $A_\alpha(t)$ reflects the amount of **confidence** at time $t$ that the ground truth $\mathbf{x}^*$ lies above or below the classification line with slope $\alpha$. Essentially, the network has to explicitly report distance from the classification lines, not just in which side of the line $\mathbf{x}^*$ lies for that trial. We set the decision bound to $\pm 5$, and plot the accumulators $A_\alpha$ for all lines in Figure 2a. Note that once the bound is reached a decision is effectively made and $A_\alpha$ is kept constant. Also, instead of using continuous values, we quantize $A_\alpha$, because it is going to be used as target signal to train the network, and we do not want to introduce a strong inductive bias towards integrating the evidence streams.

We then train the RNN to reproduce confidence estimates from Figure S5a for the entire trial. Compared to previous experiments, the fixation input is no longer available to determine when to produce a decision. Instead, decisions evolve dynamically throughout the trial. We also use a MSE loss, change the activation function to $g = 5 \tanh$, and the Adam learning rate $\eta_0 = 3 * 10^{-3}$, but all other parameters remain the same as in the main text. To have a closer correspondence to the free RT experiments here, we also train the fixed RT task from the main text with MSE loss, symmet-

ric labels $\mathbf{y}(\mathbf{x}^*) \in \{-1, +1\}^{N_{task}}$ and output non-linearity $g = \tanh$. We find that the change of objective and loss only has minor effects on generalization performance.

Figure S5b shows that in this setting the network still learns a two-dimensional continuous attractor of the latent space. Furthermore, the free RT outperforms the fixed RT network from the main text (Figure S5c) for a small number of tasks, since it is explicitly required to report distance from the classification lines. However, as our theory shows (Lemma B.3)) the fixed RT network is also implicitly reporting distance from the boundaries, when behaving like an optimal multi-task classifier, which explains the similar performance for a larger number of tasks. Overall, we showed that our setting accounts for naturalistic free RT decisions, and provides theoretical justification for the importance of confidence signals in the brain (Rutishauser et al., 2018; Masset et al., 2020).

The importance of the confidence (i.e., calibrated likelihoods) of a network's output, is a recurring theme in machine learning too (e.g. knowledge distillation (Bhargava et al., 2024)). **We here show that confidence fundamentally connects to how neural networks construct world models**, either directly (integrate-to-bound task here) or indirectly (classification tasks in main text). Under this framework, knowledge distillation can be cast as smaller models directly copying the world models (logits) of larger ones.

### A.12 QUANTIFICATION OF SPARSITY

In the main text, we observed that RNN representations are sparse. We here seek to more precisely quantify the sparsity in these networks, and investigate how it is affected by the number of tasks $N_{task}$, latent dimensionality $D$, and specific recurrent architecture. To do so, we sample $n = 1000$ ground truth vectors $\mathbf{x}^*$ randomly for every network, and compute the sparseness (Vinje & Gallant, 2000) of a neuron in the hidden layer as:

$$S = \frac{1 - \left( \frac{\sum (z_i/n)^2}{\sum (z_i^2/n)} \right)}{1 - \frac{1}{n}} * 100\,\% \tag{10}$$

where $z_i$ is the steady-state response of the neuron to ground-truth stimulus $i$. Sparseness ranges from 0 to 100 %, with greater sparseness indicating greater selectivity of the neuron to stimuli. Then, the sparsity of a network is given as the average of the sparseness of all its neurons.

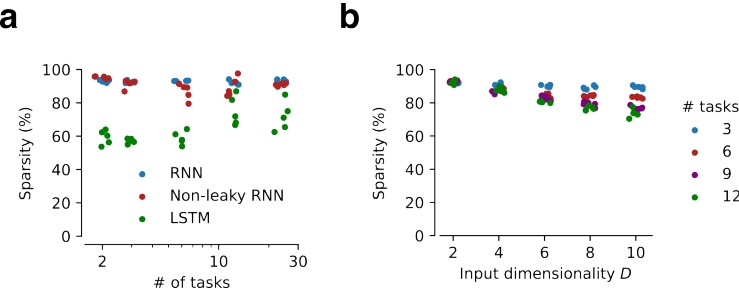

Figure S6: **Quantification of sparsity as a function of $N_{task}$, $D$ and recurrent architecture choice.** **(a)** Sparsity of a recurrent network as a function of number of tasks and network architecture. Five networks trained for each network configuration. Greater levels of sparsity indicate that the network activations are more sparse. **(b)** Sparsity of RNNs as a function of number of tasks and latent dimensionality $D$. Five network are trained for each combination of $(N_{task}, D)$.

Figure S6a shows that RNNs and non-leaky RNNs are very sparse, with sparsity values around 90 % for different values of $N_{task}$, supporting the claim in the main text. LSTMs on the other hand, which are less brain-like[4], have lower sparsity values, although interestingly sparsity increases with

---

[4]LSTMs architecturally enforce intricate, high-capacity multiplicative gating mechanisms, while in biological neural networks gating has to be learned. For other aspects of biological implausibility of LSTMs compared to RNNs, see Appendix B in (Soo et al., 2023).

$N_{task}$. Notably, we did not do anything to promote sparsity (e.g. regularization) in these networks. Therefore we conclude that sparsity naturally emerges from the optimization objective of multitask learning, particularly in architectures that are more brain-like.

Next we wondered how latent space dimensionality $D$ would affect sparsity in our trained RNNs. Figure S6b shows that networks remain very sparse for the whole range of dimensionality $D$ tested in the main text, with sparsity values above $75\%$. Greater dimensionality results in less sparsity on average, which is expected since $N_{neu} = 64$ in our networks, therefore a significant amount of their capacity must be used as $D$ increases. This effect plays in only as $N_{task}$ increases, as networks will only learn to disentangle the input dimensions that are spanned by the tasks, as our theory predicts. Overall, there seems to be a proportional relationship between the number of active neurons and dimensionality $D$, as long as there are enough tasks to uncover the $D$ latents.

## A.13 Relation to neuroscience literature

An ongoing debate in the brain sciences is whether to solve tasks the brain learns abstracts representations, or simple input-output mappings. Here we show that training RNNs to multitask results in shared, disentangled representations of the latent variables, in the form of continuous attractors. In this multitask setting, one task acts as a regularizer for the others, by not letting the representation collapse, or overfit, to specific tasks (Zhang & Yang, 2017).

Our findings directly link to two important neuroscientific findings: spatial cognition and value-based decision-making. First, the tasks here bear close resemblance to path-integration, i.e. the ability of animals to navigate space only relying on their proprioceptive sense of linear and angular velocity (Mittelstaedt & Mittelstaedt, 1980; Burak & Fiete, 2009; Vafidis et al., 2022; Sorscher et al., 2023). In path-integration animals integrate velocity signals to get location, while here we integrate noisy evidence to get rid of the noise. In path-integration, networks have to explicitly report distances, while in our setting distances are estimated implicitly (Lemma B.3)). We learn abstract representations in the form of a 2D "sheet" continuous attractor, while the computational substrate for path integration is a 2D toroidal attractor (Gardner et al., 2022; Sorscher et al., 2023) – not an abstract representation. The conditions under which a 2D sheet vs. toroidal continuous attractor is learned is a potential area of future research. Second, decision making experiments in monkeys result in a 2D abstract representation in the medial frontal cortex, which supports novel inferential decisions (Bongioanni et al., 2021). Likewise, context-dependent decision-making experiments in humans also resulted in orthogonal, abstract representations (Flesch et al., 2022).

## A.14 Biological plausibility of multi-task learning

While our theory stems from parallel processing, i.e. multi-task learning, it is not contingent upon the parallel *execution* of multiple tasks, i.e. multitasking, or the receipt of rich supervisory feedback from the environment in parallel. Behaviorally, the agent need only perform one action, the one most appropriate to its current internal state (e.g. its level of thirst vs. hunger might control the slope of the decision boundary in the 2D latent space of water & food). What we posit is that tasks that have been performed by the agent before and rely on the same input are still resolved somewhere in the brain, by the brain circuits (e.g. cortical columns Hawkins et al. (2019)) previously responsible for them, instead of the entire decision-making brain area focusing only on the current task (Mante et al., 2013). Therefore, the output of these tasks is still placing pressure on the representation, even though they are not actively driving behavior. In other words, our theory assumes **competence** at $N_{task}$ tasks, independently of when and how that competence was achieved. We feel that this is a more natural way of thinking about how the brain manages different tasks, with older tasks still leaving traces somewhere in the brain (Losey et al., 2024); after all, biological agents are remarkable *because* they achieve high performance on many tasks. This theory is also closely related to the widely observed phenomenon of memory replay (Foster & Wilson, 2006), or mental simulation of counterfactuals (Jensen et al., 2024). A future direction to further enhance the biological relevance of our work would be to investigate the relation between multi-task learning and slow, interleaved learning (see Appendix A.10), in a continual learning setting.

## A.15 LIMITATIONS AND FUTURE DIRECTIONS

A limitation of the present work is that factorization is assumed. Yet not all problems are factorizable, or should be factorized. For instance, a more coarse-grained understanding of the world, that doesn't disentangle all factors, might be more suitable in many cases, and that might be reflected in the nature of the tasks. Furthermore, we focus on canonical cognitive neuroscience tasks which are somewhat removed from standard ML benchmarks. Normally, disentanglement methods would be tested against a benchmark such as dSprites (Matthey et al., 2017); however to the best of our knowledge no such benchmark exists for sequential tasks where evidence has to be aggregated over time. Future work could endeavor to apply our setting to richer tasks, like extracting latent item attributes from item embeddings when sequential decisions are made in online retailer settings.

Our theory is agnostic to the way by which competence at multiple tasks is achieved. Thus, a natural next step is to investigate whether disentangled representations exist in a wider range of models capable of solving multiple tasks. A prime example is large language models that display excellent zero- and few-shot generalization capabilities, with progress already made in that direction (Templeton et al., 2024). Moreover, the pre-training objective for LLMs (cross entropy loss/likelihood maximization) fits well within our theoretical framing on (approximately) optimal multi-classifiers. Another application area, as already mentioned, is neuroscience; animals are naturally competent at multiple tasks, thus our work provides theoretical justification for why disentangled representations have been found in many brain areas, and motivates looking for more.

Our experiments showed parsimony of our theoretical results under conditions not covered by our theory, including non-injective observation maps (Appendix A.10) and decision boundaries (Appendix A.11) which is encouraging for testing our findings on settings beyond what is strictly covered by the theory. It would be interesting to see how the theoretical insights generalize to different task geometries, for example those implied by self-supervised learning applications (e.g. image patch-filling, next-token prediction, iterative de-noising). The connection between our framework and self-supervised learning is deep and promising. Both frameworks share a common structure, where an underlying latent truth (e.g., objects in an image and their relationships) is inferred. Each objective (e.g., filling in missing image patches) contributes synergistically to understanding the latent space as a whole. A similar logic applies to predicting words, where the latent "meaning" of a sentence is shared, whether in a causal (e.g., LLMs) or masked setting (e.g., BERT). Our study is a first effort towards understanding such parallel learning, and providing guarantees for its performance.

## B  THEORETICAL DERIVATIONS

Here we prove our main theoretical result outlined in Section 3.

**High-level summary of proof**    We prove that competence at $N_{\text{task}}$ tasks guarantees linear decodability when $N_{\text{task}} \geq D$ for non-degenerate tasks (Theorem B.5), and orthogonality when $N_{\text{task}} \gg D$ and task boundaries are sampled randomly (Corollary B.10). To that end, we first show that optimal evidence aggregation in a multi-task classification framework enforces the multi-task classifier to encode a notion of distances from classification lines (Lemma B.3). Given a suitable set of distances from classification lines, we show that one can uniquely identify an optimal estimate of the latents given noisy data in closed form (Trilateration Theorem, Theorem B.4). In addition, if the readout function of the multi-task classifier is sigmoid-like, the optimal estimate will be approximately linearly decodable from the representation (Corollary B.8,B.9). We then prove by contradiction that all the above results hold when an arbitrary injective observation map $f$ is applied to the input after noising (Theorem B.6). The theory generalizes to sub-optimal classifiers via least-mean squares approximation and the Moore-Penrose Pseudoinverse (Theorem B.7), and to different noise distributions (Section B.4). Finally, we discuss implications of the theorem for representation learning, manifold learning and the Platonic representation hypothesis, and future directions (Section B.6).

*Notation: lower case variables denote scalars (e.g., $x$), upper case variables denote random variables (e.g., $X$), and boldfaced variables denote vector quantities (e.g., $\mathbf{x}, \mathbf{X}$). We denote the $D \times D$ identity matrix as $\mathbf{I}_D$.*

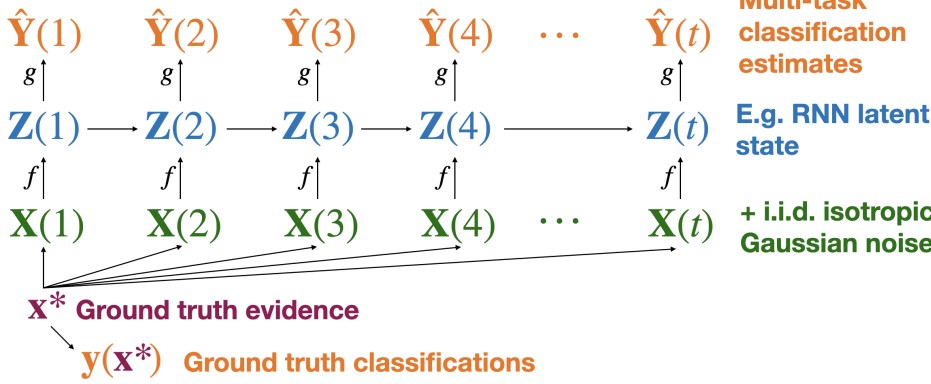

Figure S7: Bayesian graphical model framework representing our theoretical framework for multi-task classification. The agent with latent state $\mathbf{Z}(t)$ estimates the ground truth decision output $\mathbf{y}(\mathbf{x}^*) \in \{0,1\}^{N_{\text{task}}}$ from noisy observations $\mathbf{X}(t)$ transformed by injective observation map $f$. We prove that latent state $\mathbf{Z}(t)$ must encode an optimal, linearly decodable estimate of the de-noised environment state $\mathbf{x}^*$ when the decision boundary normal vectors $\{\mathbf{c}_i\}_{i=1}^{N_{\text{task}}}$ span $\mathbb{R}^D$.

**Variable Glossary:**

- $\mathbf{x}^* \in \mathbb{R}^D$ : Ground truth (un-noised) input variable of dimension $D$.
- $\mathbf{X}(t) \sim \mathbf{x}^* + \sigma \mathcal{N}(\mathbf{0}, \mathbf{I}_D)$ are i.i.d. noisy measurements of $\mathbf{x}^*$, where
    - $\sigma$ is the amount of equivariant Gaussian noise, and
    - $t$ is the discrete time index within a trial.
- $f : \mathbb{R}^D \to \mathcal{Z}$ : An injective observation map that transforms the noisy measurements $\mathbf{X}(t)$ before they reach the latent state $\mathbf{Z}(t)$ of the optimal estimator. The map $f$ is injective, meaning that it preserves the uniqueness of the input, i.e., if $f(\mathbf{x}_1) = f(\mathbf{x}_2)$, then $\mathbf{x}_1 = \mathbf{x}_2$. The codomain $\mathcal{Z}$ can be any suitable space, such as $\mathbb{C}^M$, $\mathbb{R}^\infty$, or other spaces.
- $N_{task}$ is the number of classification tasks,

- $\{(\mathbf{c}_i, b_i)\}_{i=1}^{N_{task}}$ are the classification boundary normal vectors and offsets respectively, with $\mathbf{c}_i \in \mathbb{R}^D$ and $b_i \in \mathbb{R}$. We assume each $\|\mathbf{c}_i\| = 1$.

- $(\mathbf{C}, \mathbf{b})$ are a matrix and vector representing each of the $N_{task}$ classification tasks where $\mathbf{C} \in \mathbb{R}^{N_{task} \times D}$

- $\mathbf{y}(\mathbf{x}^*) \in \{0, 1\}^{N_{task}}$ : Ground truth classification outputs, where each ground truth classification $y_i(\mathbf{x}^*)$ is given by

$$y_i(\mathbf{x}) = \begin{cases} 1 & \text{if } \mathbf{c}_i^\top \mathbf{x} > b_i \\ 0 & \text{otherwise} \end{cases} \tag{11}$$

- $\mathbf{Z}(t)$ : Latent variable of a multi-task classification model, conditional on $\mathbf{X}(1), \ldots, \mathbf{X}(t)$.

- $g$ : Map from latent state $\mathbf{Z}(t)$ to multi-task classification estimates $\hat{\mathbf{Y}}(t)$. For most of our experiments, readout map $g =$ sigmoid, for instance.

- $\hat{\mathbf{Y}}(t) := g(\mathbf{Z}(t)) \in [0, 1]^{N_{task}}$ : Output vector of the multi-task classification model at time $t$, where each $\hat{Y}_i(t)$ is a Bernoulli random variable estimator, estimating the conditional probability $\Pr\{y_i(\mathbf{x}^*) = 1\}$ given the noisy observations (via latent variable $\mathbf{Z}(t)$ – see Equation 12).

- $\hat{\mathbf{X}}(t) = \mathcal{N}(\mu(t), \Sigma(t))$ : Optimal estimate of $\mathbf{x}^*$ given measurements $\mathbf{X}(1), \ldots, \mathbf{X}(t)$, derived in Lemma B.1.

**Problem Statement:**  We consider optimal estimators of $\mathbf{y}(\mathbf{x}^*)$ in the multi-classification paradigm in Equation 12, shown graphically in Figure S7.

$$\mathbf{x}^* \to \mathbf{X}(1), \ldots, \mathbf{X}(t) \xrightarrow{f} \mathbf{Z}(t) \xrightarrow{g} \hat{\mathbf{Y}}(t) \tag{12}$$

**Contribution:**  We prove in Theorem B.6 ("Optimal Representation Theorem") that any optimal estimator of $\mathbf{y}(\mathbf{x}^*)$ described above will represent an optimal estimate of $\mathbf{x}^*$ in latent state $\mathbf{Z}(t)$. We begin by proving results on optimal estimators in Sections B.1, B.2 with identity observation map $f$, developing the linear case of the optimal representation theorem (Theorem B.5) showing that the latent state $\mathbf{Z}(t)$ must encode an estimate of $\mathbf{x}^*$ (visualized in Figure S8). We generalize this result any injective observation map $f$ in Section B.3 and derive closed-form solutions for extracting the estimate of $\mathbf{x}^*$ from $\mathbf{Z}(t)$. We derive approximation results for $g = \tanh$ in Corollary B.8 and $g =$ sigmoid in Corollary B.9 that show the representation of $\mathbf{x}^*$ in $\mathbf{Z}(t)$ will be linear-affine decodable if $g$ is in the sigmoid family of functions.

## B.1  SINGLE DECISION BOUNDARY

First, we will derive $\hat{Y}(t)$ for a single decision boundary with parameters $(\mathbf{c}, b)$. We focus on $P(\hat{Y}(t)|\mathbf{X}(1), \ldots, \mathbf{X}(t))$, reintroducing the latent variable $\mathbf{Z}(t)$ later on.

Since $y(\mathbf{x}^*)$ is a deterministic function of non-random variable $\mathbf{x}^*$, we will derive the probability distribution over $P(\mathbf{x}^*|\mathbf{X}(1), \ldots, \mathbf{X}(t))$ – denoted $\hat{\mathbf{X}}(t)$ – to determine $\hat{Y} = y(\hat{\mathbf{X}}(t))$. [5]

**Lemma B.1.** *Assuming no prior on* $\mathbf{x}^*$*, the conditional probability distribution* $\hat{\mathbf{X}}(t) \sim P(\mathbf{x}^*|\mathbf{X}(1), \ldots, \mathbf{X}(t))$ *is given by*

$$\hat{\mathbf{X}}(t) = \mathcal{N}(\mu(t), \Sigma(t)) \tag{13}$$

*where* $\mu(t) = mean(\mathbf{X}(1), \ldots, \mathbf{X}(t))$ *and* $\Sigma(t) = t^{-1}\sigma^2 \mathbf{I}_D$.

*Proof.* Since $\mathbf{X}(1), \ldots, \mathbf{X}(t)$ are i.i.d. from a Gaussian distribution with mean $\mathbf{x}^*$ and identity covariance, the sample mean is known to be distributed normally centered at the ground truth $\mathbf{x}^*$. We apply the known standard deviation of the underlying distribution (identity covariance scaled by $\sigma$) to arrive at $\Sigma(t) = t^{-1}\sigma^2 \mathbf{I}_D$ as the variance on the sample mean (derived from the central limit theorem). $\square$

---

[5] Note that the intermediate computation of $\hat{\mathbf{X}}(t)$ does not imply that a system *must* compute this value to predict $\hat{Y}$, as the full computation of $\hat{\mathbf{X}}(t)$ may not be necessary to determine $\hat{Y}(t)$.

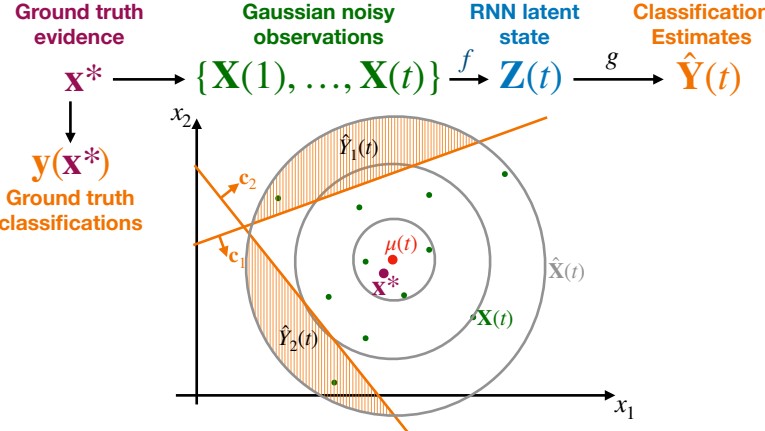

Figure S8: An overview of the classification process using an RNN with Gaussian noisy observations. The ground truth $\mathbf{x}^*$ generates the noisy observations $\{\mathbf{X}(1), ..., \mathbf{X}(t)\}$. These observations are processed by the filter-based model illustrated graphically in Figure S7, maintaining a latent state $\mathbf{Z}(t)$. The latent state $\mathbf{Z}(t)$ is then used to produce classification outputs $\hat{Y}_1(t)$ and $\hat{Y}_2(t)$. Theorem B.6 proves that $\mathbf{Z}(t)$ must encode an estimate of $\mathbf{x}^*$, visualized in this figure, shown as $\hat{\mathbf{X}}^*$, including its mean $\mu(t)$, which is the optimal estimator for $\mathbf{x}^*$ given the noisy observations.

We can use estimator $\hat{\mathbf{X}}(t)$ to construct $\hat{\mathbf{Y}}(t)$ by expanding $\hat{\mathbf{Y}}(t) = y(\hat{\mathbf{X}}(t))$ via Equation 11.

In essence, we are interested in the amount of the probability density of $\hat{\mathbf{X}}$ that lies on each side of the decision boundary. Deriving this probability is simplified by the fact that $\hat{\mathbf{X}}$ is isotropic – i.e., it inherits the spherical covariance of the underlying data generation process (Lemma B.2).

**Lemma B.2.** $\hat{\mathbf{X}}(t) = \mathcal{N}(\mu(t), \Sigma(t))$ *with isotropic covariance* $\Sigma(t) = t^{-1}\sigma^2 \mathbf{I}_D$ *and mean* $\mu(t) \in \mathbb{R}^D$. *The probability density of* $\hat{\mathbf{X}}(t)$ *on the positive side of the decision boundary* $\{\mathbf{x} : \mathbf{c}^\top \mathbf{x} > b\}$ *can be expressed as*

$$\hat{Y}(t) \triangleq \Pr\{\mathbf{c}^\top \mathbf{x}^* > b\} = \Phi(k\sqrt{t}/\sigma) \tag{14}$$

*where* $\Phi$ *is the CDF of the normal distribution and* $k = \mathbf{c}^\top \mu(t) - b$ *is the signed projection distance between the decision boundary and the mean* $\mu(t)$ *of* $\hat{\mathbf{X}}(t)$.

*Proof.* Since the $\hat{\mathbf{X}}(t)$ is isotropic, the variance on every axis is equal and independent. We may rotate our coordinate system such that the projection line between the plane and the mean of $\hat{\mathbf{X}}(t)$ aligns with an axis we denote as "axis 0". The rest of the axes must be orthogonal to the plane. Since each component of an isotropic Gaussian is independent, the marginal distribution of $\hat{\mathbf{X}}(t)$ on axis 0 is a univariate Gaussian with variance $t^{-1}\sigma^2$ mean at distance $k$ from the boundary. Equation 14 applies the normal distribution CDF $\Phi$ to determine the probability mass on the positive side of the boundary. $\square$

Observe that $\hat{Y}(t)$ in Equation 14 **monotonically scales** with the signed distance $k$ between the hyperplane and $\mu(t)$ (CDFs are monotonic).

**Lemma B.3.** *Knowledge of time* $t$ *and optimal classification estimate* $\hat{Y}(t)$ *is sufficient to determine the projection distance* $k$ *between* $\mu(t) = mean(\mathbf{X}(1), \ldots, \mathbf{X}(t))$ *and the decision boundary* $(\mathbf{c}, b)$.

*Proof.* Recall Equation 14 from Lemma B.2. We may solve for projection distance $k$ separating the decision boundary and the mean $\mu(t)$ of observations $\mathbf{X}(1), \ldots, \mathbf{X}(t)$ as

$$k = \frac{\sigma}{\sqrt{t}} \Phi^{-1}(\hat{Y}(t)) \tag{15}$$

Since $\Phi$ is the CDF of the normal distribution, and the normal distribution is not zero except at $\pm\infty$, the inverse $\Phi^{-1}$ is well-defined. □

Note that non-zero noise is required for Lemma B.3 to hold, as zero noise would yield zero probability mass on one side of each decision boundary, meaning that no distance information would be recoverable from $\hat{Y}(t)$ (and Equation (15) would lead to a $0 \cdot \infty$ indeterminacy).

## B.2 TRILATERATION VIA MULTIPLE DECISION BOUNDARIES

**To recap Section B.1** : We derived an optimal estimator of $x^*$ (denoted $\hat{X}(t)$) based on noisy i.i.d. measurements $X(1), \ldots, X(t) \sim \mathcal{N}(x^*, \sigma^2 I_D)$ in Lemma B.1. In Lemma B.2 we derived the equation for Bernoulli variable estimator $\hat{Y}(t)$ to estimate a single classification output $y(x^*)$ based on the same noisy measurements via $\hat{X}(t)$. Finally, we showed in Lemma B.3 that the uncertainty in $\hat{Y}(t)$ and the time $t$ is sufficient to determine the projection distance between the decision boundary and $\mu(t) = \text{mean}(X(1), \ldots, X(t))$ via Equation 15.

Let $\hat{\mathbf{Y}}(t)$ denote the vector of classification estimates $\hat{Y}(t)$ from Equation 15. We now have the tools to prove our final result via **trilateration**. Much like distance information from cell towers can be used to trilaterate[6] one's position, we will leverage Lemma B.3 and use distances from decision boundaries $\{(c_i, b_i)\}_{i \in [N_{task}]}$ to constrain the positions.

**Theorem B.4** (Trilateration Theorem). *If $\mathbf{C} \in \mathbb{R}^{N_{task} \times D}$ is full-rank and $N_{task} \geq D$, then $\hat{\mathbf{Y}}(t)$, $t$, $\mathbf{b}$, and $\mathbf{C}$ are sufficient to reconstruct the exact value of $\mu(t)$, the mean of $X(1), \ldots, X(t)$, which is also the optimal estimator for $x^*$.*

*Proof.* We may prove this claim by providing an algorithm to reconstruct $\mu(t) = \text{mean}(X(1), \ldots, X(t))$ from $\hat{\mathbf{Y}}(t), \mathbf{C}$, and $t$. Invoke Lemma B.3 to compute the signed projection distance between $\mu(t)$ and each decision plane $(c_i, b_i)$. Let $\mathbf{k} = [k_1, \ldots, k_{N_{task}}]^\top$ where each $k_i$ corresponds to decision boundary $c_i$. Then the mean $\mu(t)$ must satisfy

$$\mathbf{C}\mu(t) = \mathbf{k} + \mathbf{b} \tag{16}$$

Thus, for full rank $\mathbf{C}$ and $N_{task} \geq D$, we will have a uniquely determined $\mu(t)$ value. □

**Sufficient statistics and optimal estimators:** "A statistic $\mu(t)$ is called sufficient for $x^*$ if it contains all the information in $X(1), \ldots, X(t)$ about $x^*$." (from Cover and Thomas' Elements of Information Theory, 1999, Section 2.10, substituting variable names).

More formally, "A function $T(X(1), \ldots, X(t))$ is said to be a sufficient statistic relative to the family [of probability density functions indexed by $x^*$] $f(X(1), \ldots, X(t)|x^*)$ if $X(1), \ldots, X(t)$ is independent of $x^*$ given $T(X(1), \ldots, X(t))$, i.e. $x^* \to T(X(1), \ldots, X(t)) \to X(1), \ldots, X(t)$ forms a Markov chain. This is the same as the condition for equality in the data processing inequality,

$$I(x^*; X(1), \ldots, X(t)) = I(x^*; \mu(t))$$

for all distributions on $x^*$. Hence sufficient statistics preserve mutual information and conversely." (Cover and Thomas' Elements of Information Theory, 1999, Section 2.10, substituting variable names)

$\mu(t) = \text{mean}(X(1), \ldots, X(t))$ is a sufficient statistic for $x^*$ given measurements $X(i) \sim x^* + \sigma\mathcal{N}(0, I_D)$: For Gaussian noise it's a well known result that the sufficient statistic for the underlying mean given i.i.d. samples is the sample mean of the observations (Cover and Thomas, Elements of Information Theory, 1999, Section 2.10).

---

[6]Trilateration differs from triangulation, and it is more frequently used in practice. Triangulation is when one has angle information w.r.t. the cell towers. Usually, this is not available – so one **trilaterates** their position Oguejiofor et al. (2013). This more closely matches our setting, where we just have distances information w.r.t. the decision boundaries and must determine the position.

**Theorem B.5** (Optimal Representation Theorem, Linear Case). *Any system that optimally estimates classification probabilities $\hat{\mathbf{Y}}(t)$ based on noisy measurements $\{\mathbf{X}(1), \ldots, \mathbf{X}(t)\}$ must implicitly encode a representation of $\mu(t) = mean(\mathbf{X}(1), \ldots, \mathbf{X}(t))$ in its latent state $\mathbf{Z}(t)$ if decision boundary matrix $\mathbf{C}$ is full rank and $N_{task} \geq D$.*

*Proof.* We showed in Theorem B.4 (Trilateration Theorem) that if $\mathbf{C} \in \mathbb{R}^{N_{task} \times D}$ is full-rank and $N_{task} \geq D$, then $\hat{\mathbf{Y}}(t)$, $t$, $\mathbf{b}$, and $\mathbf{C}$ are sufficient to reconstruct the exact value of $\mu(t)$, the mean of $\mathbf{X}(1), \ldots, \mathbf{X}(t)$. Rearranging Equation 16 and applying Equation 15,

$$\mu(t) = (\mathbf{C}^\top \mathbf{C})^{-1} \mathbf{C}^\top \left( \frac{\sigma}{\sqrt{t}} \Phi^{-1}(\hat{\mathbf{Y}}(t)) + \mathbf{b} \right)$$

Replacing $\hat{\mathbf{Y}}(t) = g(\mathbf{Z}(t))$ from our problem setup reveals that $\mu(t)$ is a deterministic function of $\mathbf{Z}(t)$.

Therefore, optimal multi-task classifier latent state $\mathbf{Z}(t)$ contains a sufficient statistic $\mu(t)$ of $\mathbf{x}^*$, which implies that $\mathbf{Z}(t)$ must also contain all information about $\mathbf{x}^*$ given noisy measurements $\mathbf{X}(1), \ldots, \mathbf{X}(t)$ if $\mathbf{C} \in \mathbb{R}^{N_{task} \times D}$ is full-rank and $N_{task} \geq D$. ☐

Theorem B.5 boils down to the observation that the confidence associated with each $\hat{Y}_i$ in $\hat{\mathbf{Y}}(t)$ are measures of distance between an implied estimate of $\mathbf{x}^*$ (denoted $\mu(t)$) and classification boundary $i$ (denoted $(\mathbf{c}_i, b)$). $\hat{\mathbf{Y}}$ specifies the position of $\hat{\mathbf{X}} = \mu$ via "coordinates" defined by decision boundary normal vectors $\mathbf{c}_1, \ldots, \mathbf{c}_{N_{task}}$.

For sub-optimal estimators of $\hat{\mathbf{Y}}$, we may still obtain an understanding of the implied estimate $\hat{\mathbf{X}}$ using the same methods. In fact, the machinery of least-squares estimation for $\mathbf{A}\mathbf{x} = \mathbf{b}$ provides a readily accessible formula for $\tilde{\mu}$ in sub-optimal estimators of $\hat{\mathbf{Y}}$ (Equation 16) in the form of the Moore-Penrose pseudoinverse:

$$\tilde{\mu} = (\mathbf{C}^\top \mathbf{C})^{-1} \mathbf{C}^\top (\mathbf{k} + \mathbf{b}) \tag{17}$$

Conveniently, if the estimation errors in sub-optimal $\hat{\mathbf{Y}}$ have a mean of zero, additional decision boundaries in $\mathbf{C}$ (e.g., beyond the minimum $D$ linearly independent boundaries) result in improved estimation of $\mathbf{x}^*$ by the central limit theorem, thus generalizing our results to sub-optimal estimators (see Corollary B.7).

### B.3   OPTIMAL REPRESENTATION THEOREM (GENERAL CASE)

We extend the results from the linear case (Theorem B.5) to the general case where observations are transformed by an injective observation map $f$ in Theorem B.6.

**Theorem B.6** (Optimal Representation Theorem). *Let $\mathbf{x}^* \in \mathbb{R}^D$ be a latent representation for linear binary classification task $\mathbf{y}(\mathbf{x}^*) \in \{0, 1\}^{N_{task}}$ and $\mathbf{X}(t) = f(\mathbf{x}^* + \sigma \mathcal{N}(\mathbf{0}, \mathbf{I}_D))$ be noisy observations transformed by an injective observation map $f$.*

*If $\mathbf{C} \in N_{task} \times D$ is a full-rank matrix representing the decision boundary normal vectors in $\mathbb{R}^D$ and $N_{task} \geq D$, **then any optimal estimator of $\mathbf{y}(\mathbf{x}^*)$ must encode an optimal estimator $\mu(t)$ of the latent variable $\mathbf{x}^*$ in its latent state $\mathbf{Z}(t)$.** Furthermore, $\mu(t)$ is a sufficient statistic of $\mathbf{x}^*$, ensuring that all the information about $\mathbf{x}^*$ contained in $\{\mathbf{X}(t)\}$ is also contained in $\mathbf{Z}(t)$. Consequently, $\mu(t)$ – the optimal estimate of $\mathbf{x}^*$ based on $f(\mathbf{X}(1)), \ldots, f(\mathbf{X}(t))$ – can be written as a deterministic function (Equation 18) of latent state $\mathbf{Z}(t)$.*

$$\mu(t) = (\mathbf{C}^\top \mathbf{C})^{-1} \mathbf{C}^\top \left( \tfrac{\sigma}{\sqrt{t}} \Phi^{-1} \big( g(\mathbf{Z}(t)) \big) + \mathbf{b} \right) \tag{18}$$

*Proof.* We use proof by contradiction to extend the linear case of the general representation theorem to account for injective observation maps $f$ that map $\mathbf{X}(t)$ before they are input to $\mathbf{Z}(t)$.

Assume toward a contradiction that there exists a superior way of computing $\hat{Y}$ based on injectively mapped $f(\mathbf{X}(t))$ other than learning $f^{-1}$ and following the same procedure as when $\mathbf{X}(t)$ was fed in directly (which we derived the optimal estimator for in Lemma B.1 and Lemma B.2). This assumption implies there is some additional information in $f(\mathbf{X}(t))$ that is not in $\mathbf{X}(t)$, violating the data processing inequality.

Formally, consider the following Markov chain:

$$\mathbf{x}^* \rightarrow \{\mathbf{X}(1), \ldots, \mathbf{X}(t)\} \xrightarrow{f} \mathbf{Z}(t) \rightarrow \hat{Y}(t) \rightarrow \mu(t). \tag{19}$$

Since $f$ is injective, $f^{-1}$ exists, making $f(\mathbf{X}(t)) \rightarrow \mathbf{X}(t)$ an equivalent transformation in terms of information content. Hence, any optimal estimator that processes $f(\mathbf{X}(t))$ can only perform as well as if it had directly processed $\mathbf{X}(t)$.

To complete the proof, we show that $\mu(t)$ can be reconstructed from $\mathbf{Z}(t)$. Given the full-rank matrix $\mathbf{C}$, we can use the same trilateration process as in the linear case. The optimal estimate $\mu(t)$ can be written as:

$$\mu(t) = (\mathbf{C}^\top \mathbf{C})^{-1} \mathbf{C}^\top \left( \frac{\sigma}{\sqrt{t}} \Phi^{-1}\big(g(\mathbf{Z}(t))\big) + \mathbf{b} \right), \tag{20}$$

where $g(\mathbf{Z}(t))$ represents the transformation from the latent state to the classification probabilities.

Since $\mu(t)$ is the optimal estimator (and sufficient statistic) for $\mathbf{x}^*$ given measurements $\{\mathbf{X}(1), \ldots, \mathbf{X}(t)\}$, it contains all information about $\mathbf{x}^*$ contained in the measurements (Cover & Thomas (1991)). In other words, $\mu(t)$ is a deterministic function of $\mathbf{Z}(t)$, implying that $\mathbf{Z}(t)$ will contain all information about $\mathbf{x}^*$ contained in the measurements $\{\mathbf{X}(t)\}$. $\qquad\square$

**Corollary B.7** (Recovery of $\mu(t)$ for Sub-Optimal Classifiers). *Let $\hat{\mathbf{Y}}(t) \in [0, 1]^{N_{task}}$ represent the output of a sub-optimal classifier with zero-mean independent errors, i.e., $\hat{Y}_i(t) = \Pr\{y_i(\mathbf{x}^*) = 1\} + \epsilon_i$, where $\mathbb{E}[\epsilon_i] = 0$ and $Var[\epsilon_i] = \sigma_\epsilon^2$ for all $i \in \{1, \ldots, N_{task}\}$.*

*If $\mathbf{C} \in \mathbb{R}^{N_{task} \times D}$ is a full-rank and well-conditioned matrix of decision boundary normal vectors with $N_{task} \geq D$, the estimated mean $\tilde{\mu}(t)$ of $\mathbf{x}^*$ can be recovered using the Moore-Penrose pseudoinverse:*

$$\tilde{\mu}(t) = (\mathbf{C}^\top \mathbf{C})^{-1} \mathbf{C}^\top (\mathbf{k} + \mathbf{b}),$$

*where $\mathbf{k} = \frac{\sigma}{\sqrt{t}} \Phi^{-1}(\hat{\mathbf{Y}}(t))$ and $\Phi^{-1}$ is the inverse CDF of the standard normal distribution.*

*For sub-optimal classifiers, as the number of tasks $N_{task}$ increases:*

- *The redundancy in $\mathbf{C}$ reduces sensitivity to classification errors.*

- *Under the assumption of independent, zero-mean errors in $\hat{\mathbf{Y}}(t)$, the residual error in $\tilde{\mu}(t)$ is expected to decrease at a rate of approximately $\mathcal{O}(1/\sqrt{N_{task}})$, driven by the averaging effect of least-squares estimation.*

Motivated by the similarity between $\Phi(z)$ and sigmoid-like activation functions $g(z)$, we show that the two can be approximately canceled in Equation 18, implying that $\mu(t)$ can be reconstructed with high accuracy with a linear-affine transformation (e.g., linear decoding) when $g = \tanh$ or $g = $ sigmoid. This implies that $\mathbf{Z}(t)$ contains an abstract representation of $\mu(t)$ (Ostojic & Fusi, 2024).

**Corollary B.8.** *If the readout function $g$ is $\tanh$, then the reconstruction equation for $\mu(t)$ from $\mathbf{Z}(t)$ can be simplified using the approximation $\Phi(z) \approx \frac{1}{2} \tanh(\frac{\pi}{2\sqrt{3}}z) + \frac{1}{2}$. Consequently, $\mu(t)$ can be expressed directly in terms of $\mathbf{Z}(t)$ without the need for the inverse CDF.*

$$\mu(t) \approx \frac{2\sqrt{3}\sigma}{\pi\sqrt{t}} (\mathbf{C}^\top \mathbf{C})^{-1} \mathbf{C}^\top \mathbf{Z}(t) + (\mathbf{C}^\top \mathbf{C})^{-1} \mathbf{C}^\top \mathbf{b} \tag{21}$$

*Proof.* Consider the readout function $g$ given by $g(\mathbf{Z}(t)) = \hat{\mathbf{Y}}(t) = \frac{1}{2} \tanh(\mathbf{Z}(t)) + \frac{1}{2}$. To show that this function allows for linear decoding of $\mathbf{x}^*$ from $\mathbf{Z}(t)$, we need to leverage the similarity between $\tanh$ and $\Phi$.

The normal distribution CDF $\Phi(z)$ and the function $\frac{1}{2}\tanh(z) + \frac{1}{2}$ are known to be very similar, as both functions are sigmoid-like, centered at zero, and asymptotically approach 0 and 1 (Choudhury, 2014).

Page (1977) proposed a simple approximation of $\Phi$ via $\tanh$. Eliminating higher order terms, their approximation is $\Phi(x) \approx \frac{1}{2}\tanh(\sqrt{\frac{2}{\pi}}x) + \frac{1}{2}$. We found the following approximation yielded a superior mean squared error:

$$\Phi(z) \approx \frac{1}{2}\tanh\left(\frac{\pi}{2\sqrt{3}}z\right) + \frac{1}{2}.$$

Using this approximation, we can express $\Phi^{-1}$ in terms of $\tanh$:

$$\Phi^{-1}\left(\frac{1}{2}\tanh(z) + \frac{1}{2}\right) \approx \frac{2\sqrt{3}}{\pi}z.$$

Substituting this approximation into the reconstruction equation for $\mu(t)$ from Theorem B.6:

$$\mu(t) = (\mathbf{C}^\top\mathbf{C})^{-1}\mathbf{C}^\top\left(\frac{\sigma}{\sqrt{t}}\Phi^{-1}\left(\frac{1}{2}\tanh(\mathbf{Z}(t)) + \frac{1}{2}\right) + \mathbf{b}\right)$$

$$\approx (\mathbf{C}^\top\mathbf{C})^{-1}\mathbf{C}^\top\left(\frac{\sigma}{\sqrt{t}}\left(\frac{2\sqrt{3}}{\pi}\mathbf{Z}(t)\right) + \mathbf{b}\right)$$

$$= \frac{2\sqrt{3}\sigma}{\pi\sqrt{t}}(\mathbf{C}^\top\mathbf{C})^{-1}\mathbf{C}^\top\mathbf{Z}(t) + (\mathbf{C}^\top\mathbf{C})^{-1}\mathbf{C}^\top\mathbf{b}.$$

Therefore, we have shown that $\mu(t)$ can be expressed as a linear transformation of $\mathbf{Z}(t)$ when the readout function $g$ is sigmoid-like. This confirms the corollary.

$\square$

**Corollary B.9.** *For $g = $ sigmoid, linear scaling by $a_\sigma = 0.5886$ yields a mean absolute error of 0.0038699 from $Z(t)$ in the range $[-10, 10]$, enabling the following accurate linear-affine approximation of $\mu(t)$ from $\mathbf{Z}(t)$ given $g = $ sigmoid:*

$$\mu(t) \approx \frac{a_\sigma\,\sigma}{\sqrt{t}}(\mathbf{C}^\top\mathbf{C})^{-1}\mathbf{C}^\top\mathbf{Z}(t) + (\mathbf{C}^\top\mathbf{C})^{-1}\mathbf{C}^\top\mathbf{b} \tag{22}$$

*Proof.* We found $a_\sigma = 0.5886$ by computationally minimizing the mean squared error between $\Phi(a_\sigma z)$ and $\sigma(z)$ (see `sigmoid_approx_gaussianCDF.m` in supporting code). Upon computing $a_\sigma$ on successively larger optimization bounds $T = 1, 10, 100, ...$, we found that $a_\sigma$ converged to 0.5886. We note that sigmoid approximations to the Gaussian distribution CDF have existed in the literature for some time (Waissi & Rossin, 1996). $\square$

Corollary B.8 and B.9 are visualized in Figure S9, showing the close approximations to the Gaussian CDF.

**Corollary B.10.** *$N_{task} \gg D$ implies orthogonal representations in latent $Z(t)$.*

*Proof.* Recall Equation (3) from the disentangled representation theorem:

$$\mu(t) = (\mathbf{C}^\top\mathbf{C})^{-1}\mathbf{C}^\top\left(\frac{\sigma}{\sqrt{t}}\,\Phi^{-1}\big(g(\mathbf{Z}(t))\big) + \mathbf{b}\right)$$

For sigmoid-like $g$, we can approximate

$$\mu(t) \approx \frac{a_g\,\sigma}{\sqrt{t}}(\mathbf{C}^\top\mathbf{C})^{-1}\mathbf{C}^\top\mathbf{Z}(t) + (\mathbf{C}^\top\mathbf{C})^{-1}\mathbf{C}^\top\mathbf{b}$$

The orthogonality of the representations in $\mathbf{Z}(t)$ is therefore governed by the orthogonality of the rows in the matrix $\mathbf{A} := (\mathbf{C}^\top\mathbf{C})^{-1}\mathbf{C}^\top \in \mathbb{R}^{D \times N_{task}}$. If $\mathbf{A}\mathbf{A}^\top \in \mathbb{R}^{D \times D}$ is diagonal, then the rows of $\mathbf{A}$ are orthogonal.

$$\mathbf{A}\mathbf{A}^\top = \big((\mathbf{C}^\top\mathbf{C})^{-1}\mathbf{C}^\top\big)\big((\mathbf{C}^\top\mathbf{C})^{-1}\mathbf{C}^\top\big)^\top$$

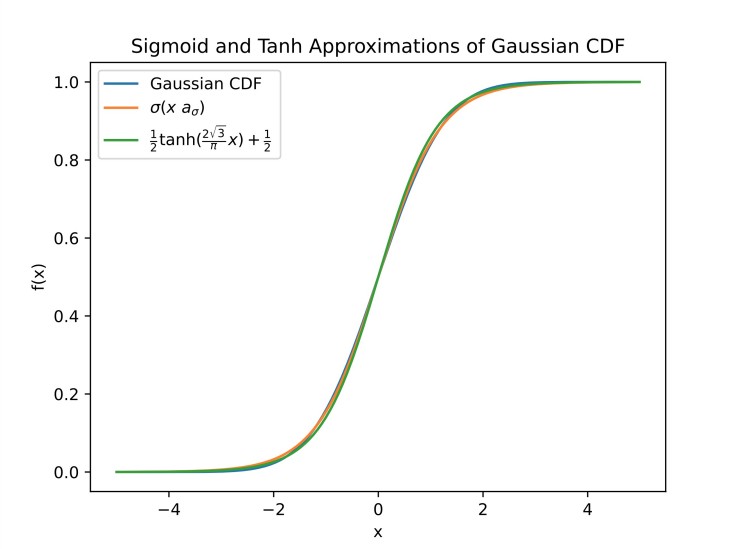

Figure S9: Sigmoid ($\sigma(\cdot)$) and $\tanh$ approximations of the normal distribution CDF $\Phi$ via horizontal scaling.

$$= \left((\mathbf{C}^\top\mathbf{C})^{-1}\mathbf{C}^\top\right)\left(\mathbf{C}((\mathbf{C}^\top\mathbf{C})^{-1})^\top\right)$$

$$= (\mathbf{C}^\top\mathbf{C})^{-1}(\mathbf{C}^\top\mathbf{C})((\mathbf{C}^\top\mathbf{C})^{-1})^\top$$

$$= ((\mathbf{C}^\top\mathbf{C})^{-1})^\top$$

$B^\top B$ is a symmetric matrix for any matrix B. Recall that the inverse of a symmetric matrix is also symmetric. So $(\mathbf{C}^\top\mathbf{C})^{-1}$ is also symmetric. Therefore

$$\mathbf{A}\mathbf{A}^\top = (\mathbf{C}^\top\mathbf{C})^{-1}$$

As the columns of $\mathbf{C}$ are high-dimensional randomly sampled vectors, their probability of being non-orthogonal vanishes as the dimensionality $N_{task}$ increases. We can also state the condition in terms of the singular value decomposition (SVD) of $\mathbf{C} = \mathbf{U}\Sigma\mathbf{V}^T$ where $\mathbf{U} \in \mathbb{R}^{N_{task}\times D}$, $\Sigma = \mathrm{diag}(\sigma_1,...,\sigma_D)$, $\mathbf{V} \in \mathbb{R}^{D\times D}$, and $\sigma_i$ is the $i$th singular value of $\mathbf{C}$ and $\mathbf{U}, \mathbf{V}$ are orthonormal. Then

$$\mathbf{A}\mathbf{A}^\top = \mathbf{V}\Sigma^{-2}\mathbf{V}^T$$

If the singular values are approximately uniform $\sigma_1 \ldots \sigma_D \approx \sigma$ then

$$\mathbf{A}\mathbf{A}^\top \approx \mathbf{V}(\frac{1}{\sigma^2}\mathbf{I}_D)\mathbf{V}^T$$

$$\mathbf{A}\mathbf{A}^\top \approx \frac{1}{\sigma^2}\mathbf{V}\mathbf{V}^T = \frac{1}{\sigma^2}\mathbf{I}_D$$

Therefore, uniform singular values in $\mathbf{C}$ is a sufficient condition to guarantee an orthogonal, disentangled representation of $\mu(t)$ in $\mathbf{Z}(t)$[7]. $\qquad\square$

As a sidenote, we would like to point out that the setting here relates to the Marchenko-Pastur law while noting important caveats: while the law typically applies to matrices with i.i.d. entries $N(0,\sigma)$, our $\mathbf{C}$ matrix consists of random row vectors in $\mathbb{R}^D$ of unit norm. This structure, while not strictly meeting the law's conditions, still supports our conclusions about orthogonalization.

---

[7]This uniformity condition in the singular value decomposition is analogous to the outcome of the LM damping technique (Levenberg, 1944; Marquardt, 1963), used for least squares inversion problems in various applications.

### B.4 SUITABLE NOISE DISTRIBUTIONS

While the original proof leverages Gaussian noise due to its mathematical convenience, the key property required for the proof is more general. Specifically, the essential requirement is that the **marginal posterior distributions along the decision boundary normals $c_i$ have invertible cumulative distribution functions (CDFs)**, allowing us to recover the distances from the observed classification probabilities.

We will now provide a precise mathematical description of the class of noise distributions where this key property holds, generalizing the disentangled representation theorem beyond Gaussian noise.

**Key Noise Property Required for Proof:** Invertibility of the Marginal Posterior CDFs Along Decision Boundary Normals. For each decision boundary normal vector $c_i$, the marginal posterior distribution of $x^*$ projected onto $c_i$ must have an invertible CDF. This allows us to map the observed classification probabilities to unique distances between the estimated mean $\mu(t)$ and the decision boundaries.

**Mathematical Description of Suitable Noise Distributions** : Let us define a class of noise distributions $\epsilon(t)$ where the key property holds and is straight forward to solve analytically.

**Definition**: The proof is immediately generalizable to noise distribution $\epsilon(t)$ if it satisfies the following conditions:

1. **Additive noise model**:
$$\mathbf{X}(t) = \mathbf{x}^* + \epsilon(t)$$
   where $\epsilon(t)$ are i.i.d. random vectors.
2. **Posterior Distribution Tractability**: The posterior distribution $P(\mathbf{x}^*|\{\mathbf{X}(s)\}_{s=1}^t$ must be analytically tractable or well-approximated, allowing us to compute the posterior mean or maximum a posteriori estimate $\mu(t)$ of $\mathbf{x}^*$ and understand its properties.
3. **Existence of Invertible Marginal Posterior CDFs**: For each decision boundary normal vector $c_i$, the marginal posterior distribution of $c_i^\top \mathbf{x}^*$ has a continuous and strictly increasing CDF $F_i(k)$, which is invertible.
4. **Support over $\mathcal{X}$**: Let $\mathcal{X} \subseteq \mathbb{R}^D$ be the connected subset of allowable $\mathbf{x}^*$ values. The noise distribution must have full support over $\mathcal{X}$, ensuring that any real-valued $\mathbf{x}^*$ is possible to trilaterate. For our proof, we assume support over the maximally permissible $\mathbb{R}^D$ is used.

**Implications:** Any suitable noise distribution allows classification task probability $\hat{Y}_i(t)$ to be expressed as
$$\hat{Y}_i(t) = \Pr\{\mathbf{c}_i^\top \mathbf{x}^* > b_i | [\mathbf{X}(s)]_{s=1}^T\}$$

$$= 1 - F_i(b_i - \mathbf{c}_i^\top \mu(t))$$
where $F_i$ is the marginal distribution of $\mathbf{c}_i^\top \mathbf{x}^*$.

Since $F_i$ is invertible, we can solve for distance $k_i = \mathbf{c}_i^\top \mu(t) - b_i$ as
$$k_i = F_i^{-1}(1 - \hat{Y}_i(t)).$$

This equation allows us to reconstruct decision boundary distances $k_i$ from optimal classification probabilities $\hat{Y}_i(t)$. Thus the proof via trilateration for the disentangled representation theorem is feasible for any suitable noise distribution $\epsilon(t)$ as described above.

### B.5 EXAMPLES OF SUITABLE NOISE DISTRIBUTIONS

#### B.5.1 ELLIPTICAL DISTRIBUTIONS

**Definition:** A multivariate distribution (Fang (2018)) is elliptical if its density function $f(\mathbf{x})$ can be expressed as:
$$f(\mathbf{x}) = |\mathbf{\Sigma}|^{-1/2} g\left((\mathbf{x} - \boldsymbol{\mu})^\top \mathbf{\Sigma}^{-1}(\mathbf{x} - \boldsymbol{\mu})\right)$$

where $g : [0, \infty) \to [0, \infty)$ is a non-negative function, $\mu \in \mathbb{R}^D$ is the location parameter, and $\Sigma \in \mathbb{R}^{D \times D}$ is the scale matrix.

**Properties:**

- Symmetric and unimodal around $\boldsymbol{\mu}$.
- Projections onto any direction $\mathbf{c}_i$ yield univariate elliptical distributions.
- Marginal distributions along $\mathbf{c}_i$ have invertible CDFs if $g$ leads to such marginals.

**Examples of Suitable Elliptical Distributions:**

- **Multi-variate Gaussians with Full-Rank Covariance Matrix** (Lemma B.11).
- **Multi-variate T-distributions with Full-Rank Scale Matrix**: Heavy-tailed alternative to the Gaussian.
- **Multivariate Laplace Distribution With Full-Rank Scale Matrix**: Has exponential tails.

### B.5.2 EXPONENTIAL POWER DISTRIBUTIONS

**Definition:** Also known as the generalized Gaussian distribution, defined by the density:

$$f(\mathbf{x}) = \frac{\beta}{2\alpha\Gamma(1/\beta)} \exp\left(-\left(\frac{|\mathbf{x} - \boldsymbol{\mu}|}{\alpha}\right)^{\beta}\right),$$

where $\beta > 0$ controls the kurtosis (Box & Tiao (2011)). Properties:

- For $\beta = 2$, it reduces to the Gaussian distribution.
- For $\beta = 1$, it becomes the Laplace distribution.
- Symmetric and unimodal.
- Marginal distributions are of the same form and have invertible CDFs.

**Generalizing the Proof**

Given the above, the proof can be generalized to any noise distribution $\epsilon(t)$ satisfying the conditions stated. The key steps are as follows:

1. Compute the Posterior Distribution:
    - Since $\epsilon(t)$ is i.i.d., the likelihood function is:

$$P(\{\mathbf{X}(s)\}_{s=1}^t \mid \mathbf{x}^*) = \prod_{s=1}^t f_\epsilon(\mathbf{X}(s) - \mathbf{x}^*).$$

    - Without a prior (uniform prior), the posterior is proportional to the likelihood.
    - The posterior distribution $P(\mathbf{x}^* \mid \{\mathbf{X}(s)\}_{s=1}^t)$ can be found (or approximated) based on the noise distribution.

2. Marginalize Along Decision Boundary Normals:
    - For each $\mathbf{c}_i$, compute the marginal posterior distribution of $\mathbf{c}_i^\top \mathbf{x}^*$.
    - Due to the symmetry and unimodality of the noise distribution, this marginal will also be symmetric and unimodal.

3. Compute Classification Probabilities: The classification probability is:

$$\hat{Y}_i(t) = \Pr\{\mathbf{c}_i^\top \mathbf{x}^* > b_i \mid \{\mathbf{X}(s)\}_{s=1}^t\} = 1 - F_i(b_i - \mathbf{c}_i^\top \mu(t)),$$

    where $F_i$ is the marginal posterior CDF of $\mathbf{c}_i^\top \mathbf{x}^*$.

4. Invert Marginal CDFs to Find Distances: Since $F_i$ is invertible, we can solve for $k_i$:

$$k_i = F_i^{-1}(1 - \hat{Y}_i(t)).$$

5. Set Up Linear System to Recover $\mu(t)$:
   - The distances $k_i$ relate to $\mu(t)$ via:
   $$\mathbf{c}_i^\top \mu(t) = k_i + b_i.$$
   - Collecting all $N_{\text{task}}$ equations:
   $$\mathbf{C}\mu(t) = \mathbf{k} + \mathbf{b}.$$

6. Solve for $\mu(t)$: If $\mathbf{C}$ is full rank, we can solve for $\mu(t)$:
$$\mu(t) = (\mathbf{C}^\top \mathbf{C})^{-1}\mathbf{C}^\top (\mathbf{k} + \mathbf{b}).$$

Implications

- Optimal Estimator Must Encode $\mu(t)$:
- The latent state $\mathbf{Z}(t)$ must contain sufficient information to recover $\mu(t)$, as it is essential for optimal classification across all tasks.
- The theorem holds for any noise distribution satisfying the stated conditions, not just Gaussian noise.

### B.5.3 EXAMPLE WITH MULTIVARIATE T-DISTRIBUTION

Suppose $\epsilon(t)$ follows a multivariate t-distribution (Kotz & Nadarajah (2004)) with degrees of freedom $\nu > 2$:

1. **Posterior Distribution**: The posterior $P(\mathbf{x}^* \mid \{\mathbf{X}(s)\}_{s=1}^t)$ is also a multivariate t-distribution.
2. Marginal Posterior Distributions: Projections onto $\mathbf{c}_i$ yield univariate t-distributions.
3. Invertible Marginal CDFs: The CDF of the t-distribution is known and invertible.
4. Recover Distances: Use the inverse t-CDF to find $k_i$:
$$k_i = s_t \cdot T_\nu^{-1}(1 - \hat{Y}_i(t)),$$
where $s_t$ is the scale parameter, and $T_\nu^{-1}$ is the inverse CDF of the t-distribution with $\nu$ degrees of freedom.
5. Proceed with the proof: Follow the same steps as before to reconstruct $\mu(t)$.

### B.5.4 EXAMPLE WITH ANISOTROPIC GAUSSIAN NOISE

**Lemma B.11.** *Suppose $\epsilon(t)$ follows an anisotropic multi-variate Gaussian distribution with full-rank covariance matrix $\Sigma$ and zero mean. Then we can update Equation 14 from Lemma B.2 as*

$$\hat{Y}(t) \triangleq \Pr(\mathbf{c}^\top \mathbf{x}^* > b) \tag{23}$$
$$= \Phi\left(\frac{k\sqrt{t}}{\sqrt{\mathbf{c}^\top \Sigma \mathbf{c}}}\right) \tag{24}$$

*Proof.* Anisotropic noise results in the quadratic form $\mathbf{c}^\top \Sigma \mathbf{c}$ in the denominator representing the variance of the marginalized anisotropic noise distribution along decision boundary normal vector $\mathbf{c}_i$. As long as $\Sigma$ is non-singular, the remainder of the disentangled representation proof may proceed substituting Equation 14 with Equation 24. $\square$

### B.5.5 CONCLUSION

The key property enabling us to recover distances from classification probabilities is the invertibility of the marginal posterior CDFs along the decision boundary normals. This property is not exclusive to Gaussian noise but is shared by a broader class of noise distributions, including but not limited to:

- Elliptical distributions (e.g., multivariate t-distributions, Laplace distributions).

- Exponential power distributions.
- Other symmetric and unimodal distributions with invertible marginals.

Therefore, the proof of the Disentangled Representation Theorem generalizes to any noise distribution satisfying the conditions outlined above. The essential requirement is that we can uniquely map the observed classification probabilities to distances along the normals, allowing us to reconstruct the posterior mean $\mu(t)$ and establish that any optimal estimator must encode this information in its latent state $\mathbf{Z}(t)$.

Correspondence in the structure of noise distribution CDF along marginals $F_i$ and point-wise activation functions $g$ (the activation function $\hat{\mathbf{Y}}(t) = g(\mathbf{Z}(t))$).

## B.6 DISCUSSION

The theoretical results presented in this appendix, particularly the Optimal Representation Theorem (Theorem B.6), provide insights into the factors driving representational convergence and alignment in neural networks and, more generally, any optimal multi-task classifier in the setup shown in Figure S7. This theorem establishes a clear connection between the latent representations learned by optimal multi-task classifiers and the true underlying data representation, offering a principled explanation for the emergence of disentangled representations aligned with the intrinsic structure of the data.

**Connection to Manifold Hypothesis:** Our theoretical results have important implications for the manifold hypothesis, which posits that real-world high-dimensional data tend to lie on or near low-dimensional manifolds embedded in the high-dimensional space (Fefferman et al., 2016; Olah, 2014). The key insight is that our proofs show an optimal multi-task classifier must encode an estimate of the disentangled coordinates of the true underlying environment state in its latent representation. Consider the disentangled space in which $\mathbf{x}^*$ resides, denoted $\mathcal{X}^*$. The injective observation map $f : \mathcal{X}^* \to \mathcal{X}$, where decision boundaries $y_i : \mathcal{X}^* \to \{0, 1\}$ are linear. Our results imply that an optimal classifier's latent state $\mathbf{Z}(t)$ must encode disentangled coordinates in $\mathcal{X}^*$ rather than ambient coordinates $\mathcal{X}$.

The injective observation map $f$ aligns closely to the typical conception of a data manifold (e.g., if $f \in C^1$ or $f \in C^n$, as described in Tu (2017)). The disentangled space $\mathcal{X}^*$ can be seen as the intrinsic coordinate system of the manifold, while $f$ maps these coordinates to the high-dimensional observation space $\mathcal{X}$. Our findings suggest that an optimal classifier will implicitly learn to invert this mapping to recover the disentangled coordinates. Moreover, for natural data where the manifold hypothesis holds, the learned latent representation would plausibly capture the manifold structure, as this is essential for disambiguating noisy observations and estimating the true underlying state. The low-dimensional manifold structure is a key prior that an optimal classifier can (and in our case must) exploit to improve its performance.

**Relation to Autoregressive Language & Multi-Modal Transformers:** Consider an analogy with masked autoencoder vision foundation models, where $\mathbf{x}^*$ is the "ground truth" of a scene (objects, positions, states, and relationships), the measurement variable $\mathbf{X}$ is an image with missing patches (Dosovitskiy et al., 2020; He et al., 2021), and the model predicts the missing patch data $y(\mathbf{x}^*)$. The model's latent variable $\mathbf{Z}$ exhibits some "understanding" of $\mathbf{x}^*$ in the form of abstract representations useful for downstream tasks. This analogy extends to masked language models (Devlin et al., 2018) and autoregressive language models (Radford et al., 2019), where $\mathbf{x}^*$ is "meaning" in a semantic space, $\mathbf{X}(t)$ are words, and $y$ is the next word. Localizing $\mathbf{x}^*$ from $\mathbf{Z}(t)$ relates to constructing a world model, showing that $\mathbf{Z}$ represents $\mathbf{x}^*$ abstractly and with high fidelity.

**Ordering of Noise and Observation Map:** The ordering of the noise and the non-linear observation map matters for the latent space representation. When the noise is applied before the observation map, the noisy observations are constrained to lie on a manifold with the same intrinsic dimension as the true latent space $\mathcal{X}^*$. In contrast, when the noise is applied after the observation map, the noisy observations can deviate from the low-dimensional manifold, potentially introducing degeneracy where two noised observations arising from different $\mathbf{x}^*$ may appear identical (i.e., non-injective). Imagine $\mathcal{X}^*$ as a 2D piece of paper. An injective, smooth, continuous observation map

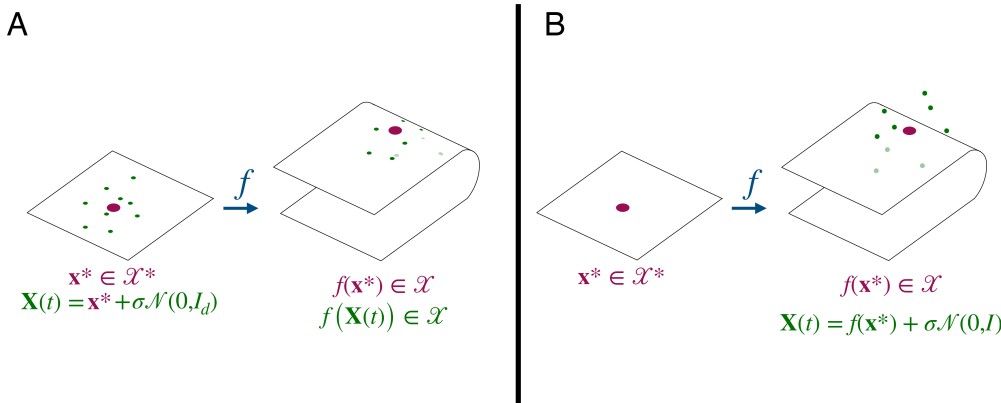

Figure S10: (A) $\mathbf{x}^*$ is noised before being transformed by injective observation map $f$, resulting in observations $f(\mathbf{X}(t))$ lying on the image of $f$ (here $f$ is a 2D folded surface). (B) $\mathbf{x}^*$ is first transformed by injective observation map $f$ and noise is added afterward, resulting in observations $f(\mathbf{x}^*) + \sigma\mathcal{N}(0, I)$ that do not lie on the image of $f$.

$f : \mathcal{X}^* \to \mathcal{X}$ where $\mathcal{X}$ is a 3-dimensional space "crumples" the sheet of paper $\mathcal{X}^*$ into a crumpled ball in $\mathcal{X}$. If you add noise after the mapping, a point on one corner of the paper could get "popped out" of the 2D manifold by the noise and end up very far away on the crumpled surface if you were to examine it flattened out (illustrated in Figure S10).

The curvature of the observation map $f$ and the level of noise $\sigma$ are fundamental factors influencing the extent of the degeneracy introduced by the noise after the observation map. High curvature in $f$ can make the intrinsic geometry of the data more challenging to identify (e.g., more tightly crumpled paper). Large noise levels can push observations further from the underlying manifold, similarly worsening the potential degeneracy in the observations. The reach of the manifold $f$ (Aamari et al. (2019)) can be used as an immediate loose bound for post-observation map noise $\epsilon_2(t)$ to ensure that the derived theorems still hold. Interestingly, this observation map degeneracy is a key concern in work on visual predictive coding for map learning by Gornet & Thomson (2024). They demonstrate that a predictive coding network can learn to map a Minecraft environment with visually degenerate states by integrating information to perform predictive coding along trajectories within the environment.

**Connection to the Platonic Representation Hypothesis:** Our results provide a new perspective on the Platonic representation hypothesis (Huh et al., 2024). The Platonic representation hypothesis suggests that the convergence in deep neural network representations is driven by a shared statistical model of reality, like Plato's concept of an ideal reality. Convergence of representations is analyzed in terms of similarity of distances between embedded datapoints among AI models trained on various modalities. While the authors of the hypothesis argue that energy constraints might lead to divergence from a shared representation for specialized tasks, our Optimal Representation Theorem suggests that the key factor driving convergence is the diversity and comprehensiveness of the tasks being learned. As long as the tasks collectively span the space of the underlying data representation, convergence to a shared, reality-aligned representation can occur, even in the presence of energy or computational limitations. Our theoretical results amount to a necessary condition for optimal multi-task classifiers to represent a disentangled representation of the data within their latent state. With energy constraints, extraneous network activity may be regularized out of the model, resulting in greater alignment between disentangled representations in energy constrained models that "understand" the Platonic nature of reality. The very energy constraints Huh et al. (2024) suggest may lead to divergence could actually facilitate convergence of the platonic representations, as they may encourage the learning of simple, generalizable features that capture the essential structure of the data. This insight opens up interesting avenues for future research on the interplay between task diversity, energy constraints, and the emergence of shared representations. Finally, energy constraints have been shown to naturally lead to predictive coding Rao & Ballard (1999); Ali et al. (2022), tightening

the relationship between energy efficiency, prediction, and cognitive map learning. A relationship between predictive coding and optimal Bayesian estimation has also been established (Rao, 1999).

**Implications and Future Directions:** The theoretical analysis presented in this appendix sheds light on the factors driving the emergence of disentangled representations in neural networks and their alignment with the intrinsic structure of the data. By formalizing the conditions under which learned representations recover the true underlying data manifold, our work provides a foundation for understanding the remarkable success of representation learning across diverse domains. Avenues for future research include exploring the sample complexity of learning under different observation maps and noise levels, and empirically validating the convergence of representations across models and modalities in the context of task diversity and energy constraints.

## C  SUPPLEMENTARY FIGURES & TABLES

Table S1: **Hyperparameter values for RNN training.** These values apply to all simulations, unless otherwise stated. $\tau = 100ms$ was chosen as a conservative estimate of membrane time constant. $\sigma$ was varied in some simulations (e.g. Figure 5c). We found that free RT tasks benefited from a higher learning rate. Other hyperparameters worked out of the box.

| PARAMETER | VALUE | EXPLANATION |
|---|---|---|
| $\Delta t$ | 100 MS | EULER INTEGRATION STEP SIZE |
| $\tau$ | 100 MS | NEURONAL TIME CONSTANT |
| $N_{neu}$ | 64 | NUMBER OF HIDDEN NEURONS |
| $\sigma$ | 0.2 | INPUT NOISE STANDARD DEVIATION |
| $T$ | 20 | TRIAL DURATION (IN $\Delta t$S) |
| $\eta_0$ | 0.001/0.003 | ADAM LEARNING RATE FIXED/FREE RT |
| $B$ | 16 | BATCH SIZE |
| $N_{batch}$ | $10^5$ | NUMBER OF TRAINING BATCHES |
| $D$ | 2 | DIMENSIONALITY OF LATENT SPACE |
| $N_{layer}$ | 1 | RNN/LSTM NUMBER OF LAYERS |

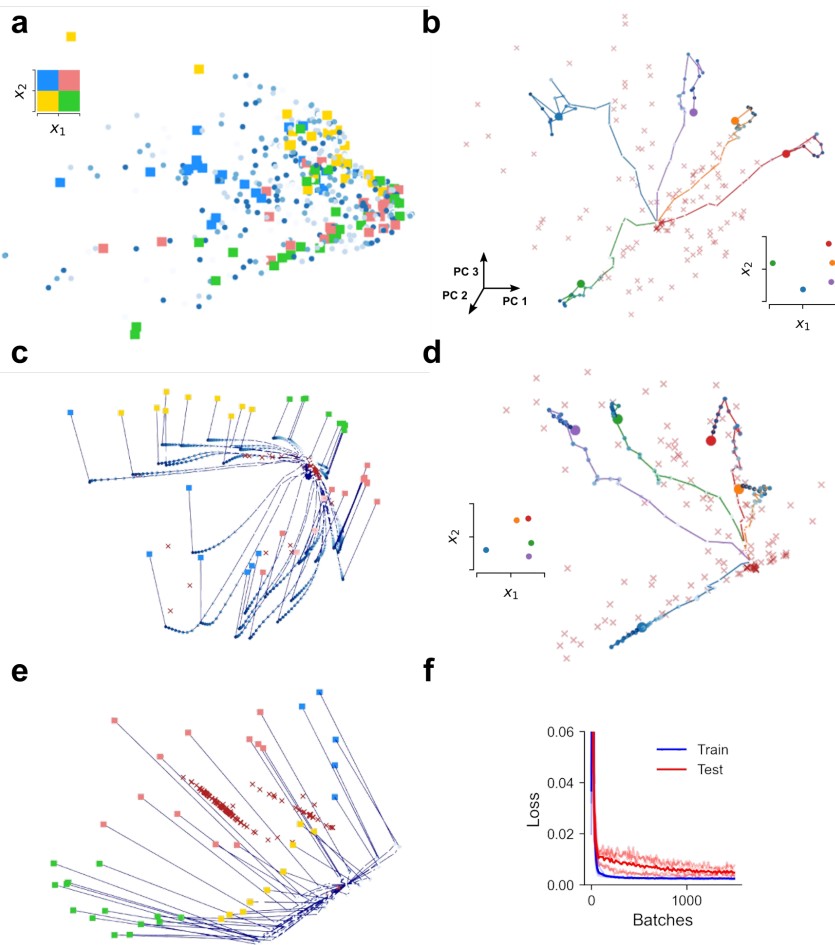

Figure S11: **Details of learned representations and of learning.** **(a)** Representation after the decoder. 3 out of 40 features were chosen randomly to be displayed. Compared to Figure 3d the representations wrap around non-linearly, and the quadrants are overlapping. The RNN needs to invert the non-linear mapping and remove the noise to arrive at disentangled representations. **(b)** Individual trial examples from network in Figure 2b. Plotting conventions same as in Figure 3d, except here every trial has been mapped to a separate color (lines and final dot). The ground truth $\mathbf{x}^*$ for these example trials is shown in the bottom right, and the attractors have been made transparent for better visibility. Note that these trials include noise, like the ones the network has been trained on. As can be seen, the network maintains a sense of metric distances in the 2D space: examples close in state space are also close in representation space. **(c)** Representation early in learning, for a network trained with 1/4 of the examples compared to Figure 3d. The representation is not disentangled yet, however it is visible how the quadrants start separating and the attractors start spreading in the 2D manifold. **(d)** Same as in **b**, but for network with a delay period of 500 ms (5 darker dots at the end of trajectories). Activity remains localized after the removal of the evidence streams, maintaining a short-term memory of the joint evidence with only minor leaks. **(e)** Representation learned in a network trained without input noise ($\sigma = 0$). Trajectories separate from the beginning, and there is no pressure to learn a 2D continuous attractor anymore. **(f)** Train and test errors for linear decoder for the OOD generalization task. Transparent lines correspond to different quadrants while opaque lines to the average across quadrants for one network.

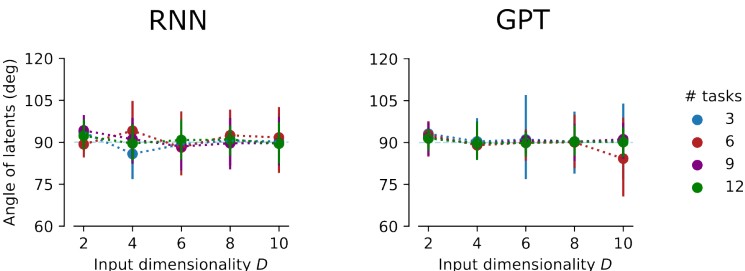

Figure S12: **Angles between latent factor decoders in higher dimensions.** Angles converge to 90 degrees as $N_{\text{task}} \gg D$ for RNNs, and as early as $N_{\text{task}} \geq D$ for transformers (see Appendix A.3 for angle estimation details). This confirms that multi-task learning leads to orthogonal, disentangled representations, in some cases even earlier than our theoretical guarantees.

