# OpenReview forum: "Disentangling Representations through Multi-task Learning"
_ICLR.cc/2025/Conference — ICLR 2025 Poster_

### Official Review · Reviewer_b5qB · 2024-10-19

**Soundness:** 3
**Presentation:** 3
**Contribution:** 3
**Rating:** 8
**Confidence:** 4

**Summary:**

This paper demonstrates the emergence of disentangled representations in recurrent neural networks and transformer architectures in classification tasks based on evidence aggregation or accumulation over time. The conditions under which disentangled (latent) representations emerge are studied in relation to the number of classification criteria, relative to the dimensionality of the input. The paper motivates its focus with reference to the neuroscience literature on evidence accumulation and offers a number of theorems predicated on mutual information and the functional forms of nonlinearities in various mappings, implicit in the network architectures that could be adopted.

**Strengths:**

The strengths of this paper rest upon a detailed treatment of the functional forms of mappings — implicit in neural network architectures — and how they can be unpacked in relation to maximising the mutual information between inputs and (classification) outputs. Another nice feature of this paper is its reference to neuroscience; in the sense of potential implications for neuronal processing in the brain.

**Weaknesses:**

The specific contributions of the analysis are not foregrounded sufficiently. For example, one could read the setup considered by the authors as characterising a mapping between inputs x and classification estimates (Y^) using a bottleneck architecture (i.e., mapping through a low dimensional latent space or manifold). If any such mapping were optimised with respect to cross entropy or mutual information, would the latent representations not be disentangled? In other words, is it a linear (orthogonal) disentanglement that qualifies as interesting disentanglement? And, if so, does this inherit from the assumptions and functional form of the mappings in question?

The second weakness is the rather colloquial appeal to the neuroscience literature. For example, evidence aggregation or accumulation would not be considered canonical in the cognitive neurosciences. Canonical paradigms will be things like oddball paradigms, psychophysical paradigms, working memory paradigms, attentional paradigms, and so on. Furthermore, most of neuroscience nowadays cast disentanglement in terms of predictive processing under hierarchical or deep generative world models (i.e., disentanglement is a way of describing the inversion of a generative model of how independent causes become entangled). This inversion is usually articulated in terms of predictive coding or belief propagation. Given that the authors mention Bayesian filtering — and predictive coding can be read as extended Kalman (Bayesian) filtering — there might have been a missed opportunity to join the dots between the authors work and current formulations of perceptual synthesis and evidence accumulation in the brain.

Related to this, there is a long history of sparse coding models in neuroscience that speak to the current questions; for example, sparse coding models, variants of independent component analysis, liquid computation, echo state machines, and so on (Gros, 2009; Hu et al., 2020; Maass et al., 2002; Olshausen and Field, 1996; Simoncelli and Olshausen, 2001; Suh et al., 2016). More recently, people have been looking at latent representations at various hierarchical levels to address compositionality and disentanglement; especially in relation to spatiotemporal receptive fields that can be assessed empirically in the brain e.g., (Ficco et al., 2021; George et al., 2021; Rao et al., 2023).

Ficco, L., et al., 2021. Disentangling predictive processing in the brain: a meta-analytic study in favour of a predictive network. Scientific Reports. 11, 16258.
George, D., et al., 2021. Clone-structured graph representations enable flexible learning and vicarious evaluation of cognitive maps. Nature Communications. 12, 2392.
Gros, C., 2009. Cognitive Computation with Autonomously Active Neural Networks: An Emerging Field. Cognitive Computation. 1, 77-90.
Hu, H.-y., et al., 2020. RG-Flow: a hierarchical and explainable flow model based on renormalization group and sparse prior. Machine Learning: Science and Technology. 3.
Linsker, R., 1990. Perceptual Neural Organization - Some Approaches Based on Network Models and Information-Theory. Annual Review of Neuroscience. 13, 257-281.
Maass, W., Natschlager, T., Markram, H., 2002. Real-time computing without stable states: a new framework for neural computation based on perturbations. Neural Comput. 14, 2531-60.
Olshausen, B.A., Field, D.J., 1996. Emergence of simple-cell receptive field properties by learning a sparse code for natural images. Nature. 381, 607-9.
Rao, R.P., 1999. An optimal estimation approach to visual perception and learning. Vision Res. 39, 1963-89.
Rao, R.P.N., Gklezakos, D.C., Sathish, V., 2023. Active Predictive Coding: A Unifying Neural Model for Active Perception, Compositional Learning, and Hierarchical Planning. Neural Computation. 36, 1-32.
Simoncelli, E.P., Olshausen, B.A., 2001. Natural image statistics and neural representation. Annu Rev Neurosci. 24, 1193-216.
Suh, S., et al., 2016. Echo-State Conditional Variational Autoencoder for Anomaly Detection. In: 2016 International Joint Conference on Neural Networks. IEEE International Joint Conference on Neural Networks (IJCNN), Vol., ed.^eds., pp. 1015-1022.

**Questions:**

For the neuroscience reader, it would be good to clarify what multi-task means in this setting. Does it mean that there are many different classification criteria or decision boundaries that have to be met? In other words, is the multi-task aspect a way of saying that the number of classification criteria have to exceed the dimensionality of the underlying (ground truth) evidence? Perhaps it would be useful to have a simple example that people can imagine applying to a perceptual classification task in a psychophysics laboratory?

By evidence aggregation, did you mean evidence accumulation in the spirit of drift diffusion and race to bound models?

Is there any relationship between the functional form of your formulation and schemes based upon the principle of maximum mutual information (a.k.a. infomax principle) (Linsker, 1990). Canonical schemes here would include canonical variates analysis and non-linear versions, such as independent component analysis that take us into the realm of sparse coding models. One interesting aspect of independent component analysis is the identification of latent variables that have a non-Gaussian structure under the assumption that the random fluctuations are Gaussian. Is this related to your interesting observation that noise is required for the kind of disentanglement you are considering?

Can you formulate your evidence accumulation scheme as a Bayesian (e.g., extended Kalman Bucy) filter? If so, can you relate this to predictive coding in the brain (Rao, 1999)?

On a related note, can you formulate your description in terms of inverting a generative model? In other words, can you turn your scheme on its head and generate ground truth evidence from the ground truth classifications; i.e. causes. If so, is this one perspective on the requisite invertibility of f?

---

> ### Author Response · Authors · 2024-11-22
> **Response to the reviewer (1/2)**
>
> We thank the reviewer for the thoughtful comments and suggestions, and for acknowledging the contributions of this work.
>
> Here we address the key points raised:
>
> ## Theoretical Contributions and Linear Disentanglement
> The reviewer asks whether linear (abstract) and orthogonal (disentangled) representations constitute interesting disentanglement. Linear representations are interesting for a couple of reasons: 1) they tend to emerge in neural networks (e.g. Templeton et al., 2024), and we don't really understand why, and 2) they make it easy to extract meaningful latents one layer downstream, which is a useful property since linear probes are often used. Disentanglement (orthogonality) is also useful for this reason, i.e. it allows linear decoders to tease apart individual latents with minimal interference from others.  We show that abstract representations emerge from optimal (or approximately optimal) multi-task classification on decision boundaries that are linear in the “ground truth” latent space (implying $N_{task} \geq D$), despite potentially complex observation maps $f$. We also show that disentangled representations (linear + orthogonal) have  to emerge from (potentially approximately) optimal multi-task classifications when $N_{task} \gg D$ span the ground truth latent space. We also foreground that our theory is derived for affine classification boundaries in the latent space.
>
> ## Clarification of Multi-Task Learning
> Theoretically, the tasks fundamentally define the latents the model must represent within our problem formulation. If tasks span only a subset of dimensions in $D$, we guarantee linear representation of that subset if the model is optimal. For sub-optimal models with zero-mean error, the machinery of least squares can be applied to combat prediction error if there if $N_{task} > D$ (equation 19, lines 1507-1520). We foreground this result in the new version of the manuscript in the theory in the main text, and as a corollary of Theorem B.5 (Corollary B.7).
>
> Furthermore, we want to clarify that indeed by evidence aggregation we mean evidence accumulation, and we also include experiments more closely mirroring drift diffusion tasks in Appendix A.11. We will also think of an example task that we could best explain this multi-task setting to the neuroscience reader.
>
> ## Connection to Predictive Processing
> Our methods complement predictive processing approaches in the following way. Our key theoretical contribution is establishing a connection between
> * A system’s **tasks/classification outputs** and
> * A system’s representation of underlying latent variables.
>
> The central question for connecting to the predictive processing view is: **“What set of latent variables is implied by the task of predictive processing?”.**  This represents a promising direction for future work, particularly in analyzing the latent representations in pre-trained autoregressive language models.
>
> ## Bayesian Filtering
> We note interesting connections to state space models and particle filters on manifolds, which relate to our framework when the observation map $f$ is a manifold. A current area of active research is extending the theory to include noise both before and after a manifold observation map $f$. Conditions relating to the reach of the manifold $f$ and its Fisher information are of great interest for understanding the conditions under which the observation map $f$ is expected to be invertible, particularly for computationally or energetically bound filters/observers.
> ## Sparse Coding
> We appreciate the connection to the sparse coding literature. Of particular interest is the connection between sparse representation learning and the particular distribution of decision boundaries in the latent space, noise properties, and the structure of the observation map $f$.
>
> Theoretically, our approach centers on connecting the multi-task predictions $\hat Y_i(t)$ produced by the evidence aggregator and the structure of the underlying latents $x^*$. A key challenge in deriving sparsity results on $Z(t)$ -- as opposed to $\hat Y(t)$. We apply the data processing inequality to show that $Z(t)$ contains the information of $\mu(t)$ -- the optimal estimate of $x^*$ given the classifier’s input data. The specific structure of this representation in $Z(t)$ is challenging to constrain by our approach, except for the final layer L of $Z(t)$ is mapped through element-wise activation function $g$ to $\hat Y(t)$ (i.e., $\hat Y(t) = g(Z_L(t))$).
>
> The specific structure of the representations in $Z(t)$ before the final layer are challenging to constrain in terms of sparsity or lack thereof as different regularization schemes on the same RNN that achieves similar performance may lead to different coding schemes for non-final layers in $Z(t)$. Yet, we now rigorously quantify sparsity in our trained RNNs, and show that it consistently emerges across all values of $N_{task}$ and $D$ of our trained networks (Appendix A.12).

---

> > ### Comment · Reviewer_b5qB · 2024-11-26
> >
> > Thank you for the clarification (and elaboration)

---

> ### Author Response · Authors · 2024-11-22
> **Response to reviewer (2/2)**
>
> A future direction of interest for understanding the advent of sparsity within our framework is to develop theory on (approximately) optimal multi-task classifiers under energetic constraints on the representations in $Z(t)$. This also relates to neuroscientific theories of efficient coding.
>
> ## Information Maximization and Noise
> Our results relate to the infomax principle but focus specifically on mutual information between ground truth latents $x^*$ and classifier state $Z(t)$, rather than mutual information maximization between the inputs of the network $f(X_1) ... f(X_t)$ and outputs $\hat Y(t)$.
> Our work particularly resembles nonlinear independent component analysis, with (approximately) optimal multi-task performance guaranteeing inversion of nonlinear observation map $f$ under certain conditions. The parallel with random Gaussian fluctuations around potentially non-Gaussian distributed ground truth latents is especially noteworthy. While the noise distribution in the main theorem is Gaussian, the theorem does not require Gaussian-distributed latent variables $x^*$.
>
> Importantly, our framework reveals that the tasks effectively define which latent dimensions of $x^*$ are represented in $Z(t)$ -- if your tasks only span a subset of $x^*$’s dimensions in $\mathbb R^D$, an optimal classifier is only guaranteed to represent those task-specific dimensions. This aligns with information theoretic intuitions: the mutual information between $Z(t)$ and the dimensions of $x^*$ irrelevant to the classification tasks need not be preserved for optimal performance. This provides a simple, principled way to think about how tasks demands shape representation learning. In contrast to infomax and nonlinear ICA, the multiple classification tasks in our framework specify which latent dimensions are relevant to the tasks at hand.
>
> This **task-driven perspective on representation learning** aligns with neuroscientific observations of task-specific representations in different brain areas (i.e., various levels of hierarchical organization). Different neural circuits appear to disentangle only those features relevant to their computational role. This suggests that task demands, in addition to general information preservation, may be a useful organizing principle for both biological and artificial representation learning.
>
> ## Generative Model Formulation
> We can indeed formulate our approach as a generative model by $x^*$ by sampling from $\hat X = \mathcal N(\mu(t), t^{-1} \sigma^2 I_D)$, where $\mu(t)$ is a deterministic function of the multi-classifier’s state $Z(t)$.
>
> This requires calibration of expected noise $\sigma$ with knowledge of time $t$. For non-Gaussian distributions (covered in B.5), knowledge of the particular shift-and-scale distribution is required. The generative model’s accuracy depends on available data at time $t$ -- fewer samples results in higher variance in the optimal estimator $\mu(t)$ about $x^*$ due to limited evidence aggregation.
> Since $\mu(t)$ is a deterministic function of classifier state $Z(t)$ and outputs $\hat Y(t)$, we are particularly interested in leveraging the reviewer’s observation to understand how predictive multi-task classification models (e.g., autoregressive language models) model the world differently based on their probability distribution over next tokens (analogous to $\hat Y(t)$). Our current work suggests a path to *conjuring* latent variables/underlying causes $x^*$ from a set of existing $\hat Y(t)$ (discussed above under connections to predictive processing, information maximization and noise). We are especially interested in this as a technique to characterize the “world models” developed by such systems.
>
> Finally, we would like to thank the reviewer for the references; we have cited some of that literature when talking about predictive coding (Rao & Ballard (1999), Rao (1999)).
>
> ### References:
> 1. Snoussi, Hichem. "Particle filtering on riemannian manifolds. application to covariance matrices tracking." In Matrix information geometry, pp. 427-449. Berlin, Heidelberg: Springer Berlin Heidelberg, 2012.
> 2. Tompkins, Frank, and Patrick J. Wolfe. "Bayesian filtering on the Stiefel manifold." In 2007 2nd IEEE International Workshop on Computational Advances in Multi-Sensor Adaptive Processing, pp. 261-264. IEEE, 2007.

---

> > ### Comment · Reviewer_b5qB · 2024-11-26
> >
> > Thank you for these responses and clarifying the link with predictive coding (and implicit world models)

---

> ### Author Response · Authors · 2024-12-04
> **Thank you**
>
> We would like to sincerely thank the reviewer for their thoughtful review, insightful comments, and support for our work.

---

### Official Review · Reviewer_xvfc · 2024-11-02

**Soundness:** 3
**Presentation:** 4
**Contribution:** 3
**Rating:** 6
**Confidence:** 4

**Summary:**

This paper studies the problem of learning disentangled representations using a multi-task evidence aggregation approach. In contrast to previous works which have studied a similar problem of abstract and disentangled representations using contextual information, the authors here adopt an approach which simultaneously collects multiple streams of evidence to enable linear classification in multiple tasks at each step. They prove the emergence of an abstract representation of ground truth data given an optimal multi-task classifier, which is shown to be learnable using RNN, LSTM, and GPT-2 based transformer architectures.

**Strengths:**

The key contribution is an extension of the Johnston and Fusi (2023) feedforward framework to recurrent networks. They show that abstract variables are indeed represented in the latent states of RNNs, LSTMs and GPT-2 style transformer architectures when these models are trained using the multi-task paradigm. The results are interesting and point to a marked advantage in adopting the multi-task paradigm for understanding canonical neuroscience experiments in an AI framework. The discussion of these results in the context of contemporary neuroscience literature is also well framed and places this paper in a good position to appeal to both the broader neuroscience and task-learning communities.

**Weaknesses:**

While the experimental results are solid contributions, the theoretical contributions are less appealing. The key theoretical results follow from fairly straightforward arguments, but appear a bit overstated. This can potentially be remedied with some re-wording and clarifying statements.

In my opinion, the following two sentences in the abstract are overstated for reasons that I expand further below:  "The key conceptual finding is that, by producing accurate multi-task classification estimates, a system implicitly represents a set of coordinates specifying a disentangled representation of the underlying latent state of the data it receives.....Overall, our framework puts forth parallel processing as a general principle for the formation of cognitive maps that capture the structure of the world in both biological and artificial systems, and helps explain why ANNs often arrive at human-interpretable concepts, and how they both may acquire exceptional zero-shot generalization capabilities".

The conditions under which the authors prove the theoretical result are rather strong. Specifically, the authors require 1) that the network is an *optimal* classifier in the sense that the network correctly computes the N_task logits associated with each of the N_task tasks, 2) that decision boundaries in the world are linear in the space of abstract variables, and 3) a highly simplified evidence accumulation problem.

It follows from 1) and 2) that the output of an optimal network encodes the distance to the N_task decision boundaries. The optimal network will thus represent a linearly transformed version of the input if N_task is larger than the dimensionality of the input (provided the decision boundaries are not degenerate). The mathematical proofs presented in this paper formalize this argument. I believe it will help clarify the results if the authors provide this intuition upfront. The proofs have various conceptual issues, for example, in the use of the data processing inequality in Theorem B.5 and the mix-up of random variables and their estimators (lines 1442-1445).

The authors state that their framework offers a "general principle for the formation of cognitive maps". I believe it will be helpful if the authors can clarify what this general principle is -- is it the proposal that decision boundaries in the world are linear in the space of the abstract variables or that biological networks are solving multiple tasks at the same time? How is the current work related to the formation of cognitive maps and zero-shot generalization?

**Questions:**

Comments:

Line 231: “…in the column space of (C^TC)^{-1}C^T potential with some element-wise non-linearity.” This statement seems too strong given that the restriction on the activation function g has been significantly relaxed. Are there any preliminary results to qualify this statement?

Corollary B9: While I agree with the results of this corollary, it would be good to explicitly connect the uniformity of the singular values to the corollary statement using the Marchenko-Pastur law. Also in this corollary, at line 1613, it is stated that the probability of being orthogonal vanishes as the dimensionality N_task increases. Was this supposed to be “non-orthogonal” instead?

Figure 2: This figure is particularly unclear, and it is perhaps one of the important foci in explaining all the experiments to follow. Upon first viewing with the caption, it is not clear what the arrows (two grey and one red) in Fig. 2a are supposed to imply.

Line 294: “…in the RNN’s hidden layer, when tasks span the latent space.” Does this really imply anything about the tasks formally spanning the latent space?

Figure 4a: I agree that the RNN representations are sparse, and this looks like a good spot to test if there is any lower threshold on the size of the hidden units before the latent encoding fails.

Line 450: “…which might explain their superior generalization performance to RNNs for lower N_task.” It is not really clear from Fig. S10 that the RNNs are indeed worse at orthogonalization than the GPT.

Line 1119/1120: “Figure S3b shows that the network still learns a disentangled, two dimensional continuous attractor.” Not convinced that this actually shows disentanglement since the metric for that qualification from before was the orthogonalization of different tasks. Did you mean to say abstract here? Also, small typo in the spelling of “disentangled”.

Minor comments:

Line 1321: \hat{Y}_i(t) is noted as a Bernoulli random variable. This implies the support of the estimator is {0,1}, but it should really be [0,1].

Line 1408: Similar comment here to Line 1321 on the use of Bernoulli variable for \hat{Y}(t).

Line 1426: In Equation 16, boldfaced t should be scalar.

Line 1592: “Proof: Recall equation (3)…” should be equation (4).

Line 1608: “B^TB is a symmetric for any matrix B.” Some LaTeX formatting errors here and possibly missing the word matrix following “symmetric”.

Line 309: “disentaglement” misspelled

Fig. 3d Caption: May be good to include a short statement about the inset on PCA components accounting for variance.

Line 1070: “Once the factors are perfectly, performance…” missing the word correlated after “perfectly”

Line 1162: “…for all lines in Figure S4a.” Did you mean all lines from Fig. 2a, shown in Fig. S4a?

---

> ### Author Response · Authors · 2024-11-22
> **Response to reviewer (1/2)**
>
> We would like to thank the reviewer for the very thorough and useful review, and for taking the time to read the paper including the proofs very carefully. In regards to the specific points raised:
>
> ## Theoretical Framework and Mathematical Precision
>
> ### Theoretical Framework Clarifications
> The reviewer correctly highlights the constraint implicit in our problem formulation that the $N_{task}$ “ground truth” task boundaries $y_i(x^*) \in \\{ 0, 1 \\} , x^* \in \mathbb R^D$ are linear. In other words, our theoretical results rely on the multi-task classifier making optimal (or approximately optimal) predictions on $N_{task}$ linear classifications on ground truth latents $x^*$ based on noisy, non-linearly mapped observations. We now explicitly mention that in the theory section of the main text (paragraph “More general decision boundaries”) along with desiderata and propositions on generalizing our main theoretical results on linear boundaries in latent space to smooth manifold decision boundaries. We also include these considerations here for reviewer convenience:
>
> ### Extending theoretical results to nonlinear boundaries on latents
> Our existing results on linear boundaries $y_i(x^*)$ in latent space function as a solution to locally linearized smooth manifold decision boundaries. Specifically, we can apply our existing results on linear decision boundaries to manifold decision boundaries $y_i(x^*)$ when $\tau_i \gg \sigma$, where $\tau_i$ is the reach of manifold $y_i$. The reach of a manifold is the largest distance from the manifold at which every point in space has a unique closest point on the manifold. Deriving maximally permissive conditions on multi-task manifold decision boundaries $y_i(x^*)$ that guarantee the emergence of disentangled/abstract representations of latents $x^*$ is left for future work.
>
> We have also re-worded and clarified the statements that the reviewer mentioned in the abstract. In particular, we now specify that the theoretical results are on “classification tasks affine in the latent space” and we removed the word “general” from the last sentence of the abstract.
>
> ### Simplicity
> We acknowledge that our main theoretical results build on straightforward arguments. We believe this simplicity is a strength, as it provides an accessible foundation for extending our results on the disentanglement of $x^*$ in the state of optimal multi-task classifiers. **Our results aim to clearly and simply show the conceptual insight that distances in latent space are reflected in an optimal (or approximately optimal) evidence aggregating multi-task classifier’s uncertainties.**
>
> ### Extension to sub-optimal classifiers
> While our main results are stated for optimal classifiers, Equation 19 (line 1515) handles sub-optimal classifiers with zero mean error, inheriting from least squares estimation. We formalize this into a corollary of theorem B.5 (Corollary B.7) and give it more prominence in the main text.
> ### Data processing inequality
> The application of the data processing inequality (DPI) in Theorem B.5 follows from the markov chain $x^* \to {X(t)} \to Z(t) \to \hat Y(t) \to \mu(t)$, which simplifies to $x^* \to Z(t) \to \mu(t)$. The DPI states that if $X \to Y \to Z$, then $I(X; Y) \geq I(X; Z)$. Applying this to our chain yields $I(x^*; Z(t)) \geq I(x^*; \mu(t))$, which directly supports our theorem’s main claim about information content in the latent state $Z(t)$. We would be thankful for further clarification from the reviewers of errors committed in this line of reasoning.
>
> ## Technical Refinements
> ### Column space and activation functions
> We acknowledge that our statement about column space representations in line 231 was overstated. We have removed that statement completely, focusing on empirical results instead (e.g. robustness to noise distribution/activation function mismatch in Appendix A.9).
>
> ### Singular value analysis
> Regarding corollary B.10, we incorporate a discussion of the Marchenko-Pastur law while noting important caveats: while the law typically applies to matrices with i.i.d. entries $N(0, \sigma)$, our $C$ matrix consists of random row vectors in $R^D$ of unit norm. This structure, while not strictly meeting the law’s conditions, still supports our conclusions about orthogonalization. We also corrected the statement in line 1683 to read “non-orthogonal” instead of “orthogonal”.
>
> ### Statistical precision
> We improve precision in our treatment of random variables and estimators in lines 1496-1499. Additionally, we clarify that $\hat Y_i(t)$ represents a Bernoulli random variable parameter estimator for assigning probabilities to binary classes, rather than a Bernoulli random variable itself in other places of Appendix B.

---

> > ### Comment · Reviewer_xvfc · 2024-11-26
> > **Data-processing inequality**
> >
> > > The application of the data processing inequality (DPI) in Theorem B.5 follows from the markov chain $x^* \to {X(t)} \to Z(t) \to \hat Y(t) \to \mu(t)$, which simplifies to $x^* \to Z(t) \to \mu(t)$. The DPI states that if $X \to Y \to Z$, then $I(X; Y) \geq I(X; Z)$. Applying this to our chain yields $I(x^*; Z(t)) \geq I(x^*; \mu(t))$, which directly supports our theorem’s main claim about information content in the latent state $Z(t)$. We would be thankful for further clarification from the reviewers of errors committed in this line of reasoning.
> >
> > The authors state in line 1550 (of the current version) that $I(x^*; \mu(t)) = H(\mu(t))$. By the definition of mutual information, $I(x^*; \mu(t)) = H(\mu(t)) - H(\mu(t)|x^*)$. The authors' statement then implies that $H(\mu(t)|x^*) = 0$. This cannot be true as $\mu(t)$ is a random variable that depends on the stochastic draws $X(1),X(2),\dots,X(t)$.
> >
> > Line 1551: "Therefore $I(x^*; Z(t)) ≥ H(\mu(t))$, implies that $I(Z(t); \mu(t)) = H(\mu(t))$".  why?

---

> ### Author Response · Authors · 2024-11-22
> **Response to reviewer (2/2)**
>
> ## Cognitive maps
> We state in the manuscript that our work offers a principle for the formation of cognitive maps (or “world models”). The suggested principle is that cognitive maps (world models) emerge from parallel processing (multi-task learning). In biological systems, this could be implemented at the level of cortical columns which have been proposed to operate independently from each other (Hawkins et al, 2019), and in artificial neural networks this bears close resemblance to self-supervised learning regimes (see l. 1908-16 in the manuscript). Cognitive maps is a concept often invoked to describe brain representations that capture the structure of the real world, in a spatial (e.g. grid cells) or a more high-level semantic level. Abstract, or disentangled, representations do exactly that, as they separate latent factors present in the real world in orthogonal axes. Finally, the link to zero-shot generalization was established in the experiments throughout the paper where we report zero-shot generalization performance $r^2$. We hope this helps clarify how these concepts come together in this work.
>
> ## Decreasing dimensionality of $Z$
> To examine when the latent encoding fails, we decrease the number of neurons when $D=10$, i.e. when we need the most amount of neurons to disentangle all latents. For $N_{task} = 24$ and $N_{neu}=64$, median $r^2 = 0.9$, therefore representations are abstract. However when decreasing $N_{neu}=32$, median $r^2 = 0.52$, and when $N_{neu}=16$, median $r^2 = 0.19$, i.e. generalization performance is completely hampered. Therefore we conclude that while there is no strictly defined threshold, $N_{neu}$ should be a multiple of $D$ to ensure enough capacity to perform all the functions that the network does (encode the factors, invert nonlinear encoder, denoise etc.).
>
> ### In regards to the reviewer’s other questions:
> * We thank the reviewer for pointing out the vagueness in Fig 2a. We now describe how the right plot is created by adding noise, and how the color-coded boolean variables in $y$ are determined by the classification lines of the corresponding color.
> * Line 298: This is more of a statement here, rather than a finding, because for $D=2$ the classification lines will always span the space. This assumption is really tested in Figure 5b.
> * We agree that Fig. S10 (now Fig. S12) might be a bit hard to read, but it can be seen that the mean angle for GPT is always close to 90 degrees (with the exception of $(N_{task}=6, D=10)$ which is below the required $N_{task} \geq D$), while for RNNs it can vary (especially for $D=2$ and $D=4$).
> * We thank the reviewer for catching the “disentangled” vs. “abstract” statement regarding Fig. S3b (now Fig. S4b). We have corrected it.
>
>
> We have also addressed all minor comments of the reviewer in the new version of the manuscript. Again, we thank the reviewer for the very careful reading of the paper.

---

> ### Author Response · Authors · 2024-11-29
> **Response to reviewer**
>
> We thank the reviewer for their clarification of the issues with the current proof. We have revised the proofs for Theorem B5-B6 to remove the inaccurate argument as follows:
>
> Since $\mu(t)$ is the optimal estimator (and sufficient statistic) for $\mathbf x^*$ given measurements $\{\mathbf X(1), \dots, \mathbf X(t)\}$, it contains all information about $\mathbf x^*$ contained in the measurements (Cover & Thomas, 1991).
> In other words, $\mu(t)$ is a deterministic function of $\mathbf Z(t)$, implying that $\mathbf Z(t)$ will contain all information about $\mathbf x^*$ contained in the measurements $\{\mathbf X(t)\}$
>
> An alternative way to answer this question would be to condition mutual information on the noisy samples, which are known to the agent. We chose this way because it is cleaner. Happy to answer any more questions that the reviewer may have, as the discussion period winds down.

---

> > ### Comment · Reviewer_xvfc · 2024-11-29
> >
> > Thank you for the clarification and the revisions made to the manuscript. I am afraid the current version does not address my concern regarding the proofs to the main theorems. The new section in the proof of B.5 reads:
> >
> > > Since µ(t) is the optimal estimator (and sufficient statistic) for x* given measurements
> > {X(1), . . . , X(t)}, it contains all information about x* contained in the measurements (Cover &
> > Thomas (1991)). In other words, µ(t) is a deterministic function of Z(t), implying that Z(t) will
> > contain all information about x* contained in the measurements {X(t)}
> >
> > I am not sure how to interpret this argument, which also seems disconnected from the rest of the proof. The same argument is copied in the proof of B.6.
> >
> > I will not change my score, but I encourage the authors to please fix the proofs (or omit them entirely) in future versions of the manuscript.

---

> ### Author Response · Authors · 2024-12-02
> **Response to reviewer**
>
> We thank the reviewer for their continued assistance in improving the clarity of the proof. Here we streamline the logical reasoning for Theorem B.5, showing that $\mathbf Z(t)$ must contain all information about $\mathbf x^*$ given the noisy measurements $\mathbf X(1), ..., \mathbf X(t)$. We show below how that ties in with the notion of sufficient statistics which we introduced in our previous response. Most of the proof is already in the paper, we outline new additions in detail:
>
> ## Proof Sketch: Theorem B.5
> **Goal**: Show that optimal multi-task classifier latent state $\mathbf Z(t)$ contains all information about $\mathbf x^*$ given noisy measurements $\{\mathbf X(1), \dots, \mathbf X(t)\}$ if $\mathbf C \in \mathbb R^{N_{task} \times D}$ is full-rank and $N_{task} \geq D$.
>
> **Background on sufficient statistics:** "A statistic $\mu(t)$ is called sufficient for $\mathbf x^*$ if it contains all the information in $\{\mathbf X(1), \dots, \mathbf X(t)\}$ about $\mathbf x^*$." (verbatim from Cover and Thomas' Elements of Information Theory, 1999, Section 2.10, substituting variable names). Therefore, if we show that $\mathbf Z(t)$ contains a sufficient statistic $\mu(t)$ of $\mathbf x^*$, we have completed the proof.
>
> More formally, "A function $T(\{\mathbf X(1), \dots, \mathbf X(t)\})$ is said to be a sufficient statistic relative to the family \[of probability density functions indexed by $\mathbf x^*$\] $f(\{\mathbf X(1), \dots, \mathbf X(t)\} | \mathbf x^*)$ if $\{\mathbf X(1), \dots, \mathbf X(t)\}$ is independent of $\mathbf x^*$ given $T(\{\mathbf X(1), \dots, \mathbf X(t)\})$, i.d., $\mathbf x^* \to T(\{\mathbf X(1), \dots, \mathbf X(t)\})$ forms a Markov chain. This is the same as the condition for equality in the data processing inequality, $$I(\mathbf x^*; \{\mathbf X(1), \dots, \mathbf X(t)\}) = I(\mathbf x^*; \mu(t))$$
> for all distributions on $\mathbf x^*$. Hence sufficient statistics preserve mutual information and conversely." (Cover and Thomas' Elements of Information Theory, 1999, Section 2.10, substituting variable names)
>
> $\mu(t) = \text{mean}(\mathbf X(1), \dots, \mathbf X(t))$ **is a sufficient statistic for** $\mathbf x^*$ **given measurements** $\mathbf X(i) \sim \mathbf x^* + \sigma \mathcal N(0, \mathbf I_D)$: For Gaussian noise it's a well known result that the sufficient statistic for the underlying mean given i.i.d. samples is the sample mean of the observations (Cover and Thomas, Elements of Information Theory, 1999, Section 2.10).
>
> **Showing that $\mu(t)$ is a deterministic function of $\hat {\mathbf Y}(t)$:** We showed in Theorem B.4 (Trilateration Theorem) that if $\mathbf C \in \mathbb R^{N_{task} \times D}$ is full-rank and $N_{task} \geq D$, then $\hat {\mathbf Y}(t)$, $t$, $\mathbf b$, and $\mathbf C$ are sufficient to reconstruct the exact value of $\mu(t)$, the mean of $\mathbf X(1), \dots, \mathbf X(t)$. Rearranging Equation 16 and applying Equation 15,
>
> $$
> \mu(t) = (\mathbf C^\top \mathbf C)^{-1} \mathbf C^\top (\frac{\sigma}{\sqrt{t}} \Phi^{-1}(\hat{\mathbf Y}(t)) + \mathbf b)
> $$
>
> Replacing $\hat{\mathbf Y}(t) = g(\mathbf Z(t))$ from our problem setup reveals that $\mu(t)$ is a deterministic function of $\mathbf Z(t)$.
>
> Therefore, optimal multi-task classifier latent state $\mathbf Z(t)$ contains a sufficient statistic $\mu(t)$ of $\mathbf x^*$, which implies that $\mathbf Z(t)$ must also contain all information about $\mathbf x^*$ given noisy measurements $\{\mathbf X(1), \dots, \mathbf X(t)\}$ if $\mathbf C \in \mathbb R^{N_{task} \times D}$ is full-rank and $N_{task} \geq D$.
> $$\square$$
>
>
> We hope this clarifies our proof to the reviewer. We cannot make changes to the manuscript now, but we will include these new clarifications in the new version. The same fix applies to theorem B.6.

---

### Official Review · Reviewer_Mvb9 · 2024-11-02

**Soundness:** 2
**Presentation:** 2
**Contribution:** 1
**Rating:** 3
**Confidence:** 4

**Summary:**

This paper extends the work of [Johnston and Fusi (2023)](https://www.nature.com/articles/s41467-023-36583-0)  by investigating how multiple supervised binary classification tasks can lead to abstract or disentangled representations in recurrent neural networks that accumulate evidence from noisy inputs. The authors present theoretical and experimental evidence suggesting that optimal multi-task classifiers can learn abstract representations of underlying ground truth factors.

**Strengths:**

The paper introduces a theoretical framework connecting optimal multi-task classification to disentangled representations, building upon the empirical observations in [Johnston and Fusi (2023)](https://www.nature.com/articles/s41467-023-36583-0) . To support these theoretical claims, the paper provides a range of experiments exploring different architectures, task structures, and decision boundary geometries.

**Weaknesses:**

A significant weakness of this paper is its heavy reliance on dense supervised signals to achieve abstract representations. Similar to [Johnston and Fusi (2023)](https://www.nature.com/articles/s41467-023-36583-0), this work utilizes multiple binary classification tasks, framed as "multi-task" learning. This framing implies a diversity of tasks that is not truly reflected in this setup. The dependence on numerous, highly similar, supervised tasks raises questions about the biological plausibility and real-world applicability of the proposed mechanism.

The requirement of a large number of tasks (N_task >> D) to achieve disentanglement adds to concerns regarding the model's biological plausibility and practical relevance. Biological and artificial agents operating in real-world environments rarely encounter such an abundance of supervised signals.

The authors assert (lines 486-493) that their Theorem 3.1 implies "the key factor driving convergence [to a Platonic representation of reality] is the diversity and comprehensiveness of the tasks being learned." This claim is not supported by the results presented. The tasks in this paper are essentially multiple variations of the same supervised classification task, a far cry from the genuinely diverse tasks (e.g., multimodal vision and language tasks) considered in the Platonic representation paper ([Huh et al., 2024](https://arxiv.org/abs/2405.07987)).

It is well-established that achieving disentangled representations in real-world, unsupervised settings is incredibly difficult, if not impossible, without incorporating appropriate inductive biases ([Locatello et al., 2019](https://arxiv.org/abs/1811.12359); which the authors cite). A substantial body of research, spanning decades, has demonstrated that source signals become non-identifiable when mixed non-linearly ([Hyvarinen and Pajunen, 1999](https://www.sciencedirect.com/science/article/pii/S0893608098001403)). Considering this extensive history of work in machine learning dedicated to tackling this complex problem, this paper's approach—attaining abstract or disentangled representations by providing the model with an abundance of supervised signals in the form of binary classification tasks—appears contrived and ultimately of limited practical value. The reliance on such a heavily supervised setting makes it difficult to see how these findings could generalize to more realistic scenarios where supervision is scarce and the underlying structure of the data must be inferred, rather than provided.

Overall, this paper makes incremental progress on the work of [Johnston and Fusi (2023)](https://www.nature.com/articles/s41467-023-36583-0)  but fails to address the fundamental concern of over-reliance on dense supervised signals. As suggested by [Johnston and Fusi (2023)](https://www.nature.com/articles/s41467-023-36583-0), exploring unsupervised mechanisms such as "predicting the sensory consequences of our actions," might offer a more promising path forward.

Instead of relying on a multitude of supervised signals to enforce disentanglement, future work in this area should investigate whether disentanglement can emerge from a smaller number of more diverse tasks, where the need for efficient generalization across tasks drives abstraction. This shift in focus would significantly enhance the biological plausibility, practical relevance, and overall impact of this line of research to understanding disentangled representation learning.

**Questions:**

**Sparsity:**

The emergence of sparsity in Fig. 4 is interesting. Could the authors specify if they applied any particular architectural constraints, such as regularization techniques or sparsity-promoting mechanisms, to encourage this behavior?

Additionally, quantifying the sparsity (e.g., lifetime sparsity, [Vinje and Gallant, 2000](https://www.science.org/doi/10.1126/science.287.5456.1273)) and examining how this metric varies with N_task, the number of latent dimensions, and specific RNN architecture choices would provide further insights into the effects of task density on network representation.

**More nonlinear encoding:**

The authors' use of a piecewise linear MLP+ReLU encoding of ground truth signals raises questions about the robustness of their empirical results under more complex nonlinear encodings.

Could the authors explore how the N_task vs. out-of-distribution r2 relationship (Fig. 3) might change if the encoding network incorporated more nonlinear functions such as exponential or power-law activations, to simulate more biologically plausible ([Priebe et al., 2004](https://www.nature.com/articles/nn1310)) nonlinear encoding? This line of inquiry is particularly relevant given that the choice of activation function significantly influences the geometry of representations ([Alleman et al., 2024](https://openreview.net/forum?id=k9t8dQ30kU)).

---

> ### Author Response · Authors · 2024-11-22
> **Response to reviewer (1/4)**
>
> We thank the reviewer for their review. In regards to the specific points raised:
> ## Definition of multi-task learning
> The reviewer seems to disagree with framing our setting as multi-task learning, citing the lack of diversity in the tasks. They also point out that related work by Huh et al (2024) met this diversity requirement, by training networks on multimodal tasks. However, diversity in the sense of multimodality is not a requirement for multi-task learning. Multi-task learning is defined as the “training of [related] tasks in parallel while using a shared representation” (Caruana (1997)). Therefore, our work falls within the multi-task learning framework. Furthermore, our definition of task diversity is mathematical, not colloquial (“multimodality”). By diversity of task demands we refer to whether the decision boundaries *span* the latent space, involving all latent variables in the decisions, which is a condition for disentangling all factors. So while we do provide results from a mixture of classification boundary geometries in the appendix, we keep the tasks as simple as possible to derive theoretical insight into the nature of generalization and world model construction by means of rigorous proofs, which would be difficult to get a grasp on if we used rich multimodal datasets instead (as **reviewer 2Aef** neatly points out). We agree that it would be interesting to extend the experiments to multi-modal tasks, but this would be a new follow-up study on its own.
> ## Regarding the feasibility of disentanglement
> There seem to be two concerns here, that are of different nature and which we therefore address separately.
> First, from a theoretical standpoint, the reviewer argues that disentangling representations might be difficult, or even impossible, citing previous work on unsupervised learning (Locatello et al (2019), Hyvarinen & Pajunen (1999)), which is not what we do. Meanwhile, we mathematically prove conditions under which disentangling representations is not just possible, but *guaranteed*. In particular, a system being approximately optimal at tasks involving the same latents *is* an inductive bias; therefore our results do not contradict Locatello et al (2019). Furthermore, the “impossibility result” in Locatello et al rests on the narrow view of disentangled representations as axis-aligned, i.e. a single ground truth factor change should lead to a single change in the representation, similar to the single neuron coding assumption in neuroscience. We argue in favor of a mixed representation (Rigotti et al 2013) view, because by looking at single units we might be missing cases where disentanglement exists at the population level; a point also raised by Johnston & Fusi (2023).
>
> On a second note, the reviewer raises a concern about the supervised structure of our tasks, claiming that such supervised datasets would be difficult to find in real-world settings. We disagree, as in fact modern self-supervised learning involves converting an unsupervised objective (“scene understanding”) into a well-posed supervised objective (image patch-filling) (see l. 1909-16 in the manuscript). Filling missing patches *is* a multi-task objective, with a common, shared latent space (objects present in an image, and their relations), and solving each one of the objectives acts synergistically in understanding the image latent space as a whole. The same logic can be extended to other modalities, for example in language, when predicting masked words that rely on a common latent sentence “meaning”. Our study is a first effort towards understanding such parallel learning, and providing guarantees for its performance. In the future, we plan to extend the insights from this study to real-world dynamic datasets.
> ## Regarding “abundance of supervised tasks”
> One of the main concerns of the reviewer is the apparent use of “dense supervised signals”. Yet, our setting requires the minimum possible number of tasks ($N_{task}$) mathematically required for identifiability in $D$ dimensional space. To uniquely identify a point in $D$ dimensional space, one needs distance information from at least $D$ boundaries, i.e. $N_{task} \geq D$. This is exactly the condition we find for linear decodability in our setting (eqs 3 and 4). Therefore, for linear decodability, no abundance of supervised tasks is required - only the minimum $N_{task}$ that would make identifiability mathematically feasible.
>
> Regarding orthogonality, our theoretical bound is less tight here ($N_{task} \gg D$). However, empirically we find that neural networks disentangle much earlier (figure 5b), and in particular, transformers disentangle for the minimum possible number of tasks that identifiability is possible (again, $N_{task} = D$). Compared to alternatives for representation learning in the brain (Mante et al 2013), which requires the construction of task-specific subspaces scaling exponentially with $D$, our work confers enormous algorithmic advantages.

---

> ### Author Response · Authors · 2024-11-22
> **Response to reviewer (2/4)**
>
> The reviewer concludes by saying that:
>
> > “future work in this area should investigate whether disentanglement can emerge from a smaller number of more diverse tasks, where the need for efficient generalization across tasks drives abstraction”
>
> However, this is exactly what our work is showing. Task diversity is defined by the dimensionality of the subspace they span in the latent space, and $N_{task} = D$ tasks that span the entire latent space is the minimal task set that ensures identifiability. Furthermore, one of our main findings is not that the need for generalization drives abstraction; rather generalization is a by-product that comes about by the pressures imposed when learning to solve multiple tasks optimally, given noise and time constraints.
>
> On that note, even having less tasks than the dimensionality $D$ is not a problem: in that case, we expect that our networks would learn to disentangle the subspace spanned by the available tasks. We expand more on the dynamic interplay of $N_{task}$ and $D$ implied by our work in the discussion and main rebuttal above.
>
> We would also like to note that the reviewer’s assertion that the
> >“ underlying structure of the data… [is] provided”
>
> in our setting is incorrect; the only thing provided to the networks are binary decisions driven by that underlying structure, not the underlying structure itself. The structure is discovered by means of optimally performing (classification) tasks in parallel. Being able to extract latents just from solving multi-task learning objectives is an exciting guarantee that opens up a lot of future research directions.
>
> ### Conflation of “competence at N tasks” with “receiving feedback signals” on N tasks
>
> The reviewer seems to conflate “competence at N tasks” with “receiving feedback signals on N tasks”. Our theory doesn’t require external signals. It just requires approximate optimality on the multi-task classification objective -- regardless of how and when that was learned. E.g., you could find such reps in a pre-trained LLM if you prompt it right. **Our contribution is that competence on N tasks -> uncovering latent variables.** In other words, the labels provided to the network in Fig. 2b need not come externally. They can originate internally, for tasks that the organism is already competent at. Biological agents are remarkable *because* they achieve high performance on many tasks, and as a result they uncover latent variables that underlie the tasks (as we show here). We expand more on this in the next section.
>
> ## Biological plausibility of multi-task learning
> The reviewer mentions that this multi-task learning setting might be not biologically relevant, as organisms rarely receive such rich feedback from the environment. However, while our theory stems from parallel processing, i.e. multi-task learning, it is not contingent upon the parallel execution of multiple tasks, i.e. multitasking, or the receipt of such feedback in parallel. Behaviorally, the agent need only perform one action, the one most appropriate to its current internal state (e.g. its level of thirst vs. hunger might control the slope of the decision boundary in 2D space of water & food). What we posit is that tasks that have been performed by the agent before and rely on the same input are still resolved somewhere in the brain, by the brain circuits (e.g. cortical columns Hawkins et al. (2019)) previously responsible for them, instead of the entire decision-making brain area focusing only on the current task (Mante et al., 2013). Therefore, the output of these tasks is still placing pressure on the representation, even though they are not actively driving behavior. We feel that this is a more natural way of thinking about how the brain manages different tasks, with older tasks still leaving traces somewhere in the brain ([Losey et al., 2024](https://www.sciencedirect.com/science/article/abs/pii/S0960982224002987)). This is in a similar vein to the quotation from Johnston & Fusi (2023) about prediction of outcomes under different hypotheticals that the reviewer provides (see [Jensen et al, 2024](https://www.nature.com/articles/s41593-024-01675-7)).  A link to the predictive coding could also be established here (see **reviewer b5qB’s** insightful comments). The key question for connecting to predictive processing-centric views is: “What set of latent variables is implied by the task of predictive processing?”
>
> Furthermore, we also show that our setting generalizes to interleaved learning of tasks with different decision geometry structure; the relationship between interleaved and multi-task learning is an interesting direction for future research, and would more closely mimic neuroscience experiments where new tasks are introduced sequentially. We have added these comments in Appendix A.14 of the revised manuscript.

---

> ### Author Response · Authors · 2024-11-22
> **Response to reviewer (3/4)**
>
> ## Regarding “incremental progress” claim
> The reviewer argues that our work is incremental progress over [Johnston & Fusi (2023)](https://www.nature.com/articles/s41467-023-36583-0). While our paper builds on their work, and we did utilize the experimental framework of this study, for its conceptual simplicity and amenability to theory, **our findings go way beyond the findings of Johnston & Fusi (2023) in several ways** (see final paragraph in section 1.1, “Related work” in the original manuscript), which we repeat here for reviewer convenience:
> * We extend the findings of Johnston & Fusi (2023) to **autoregressive architectures** that can update their representations as further information arrives. Going from feedforward to recurrent/autoregressive architectures already presents technical challenges that make the work non-incremental, and opens up the framework to a family of **dynamic tasks** that feedforward architectures are not suited for. We go beyond that to:
> * **Prove theorems that guarantee** the emergence of linearly decodable and orthogonal representations in **any** optimal multi-task classifier, and experimentally confirm the findings
> * Showcase the **importance of noise for disentanglement**, showing that more noise during pretraining results in better out-of-distribution generalization. Johnston & Fusi (2023) did not include noise in their experiments, which is an essential component of biological neural networks
> * Explore **a range of values for $D$**, providing experimental validation of our theory
> * Include **correlations between latent factors**, rendering entire areas in state space virtually unseen by the network during training
> * **Quantify disentanglement**, in addition to abstractness
> * Show that the substrate for disentangled representations in RNNs is **continuous attractors**, that maintain a short-term memory of the current evidence and updating it over time
> * Show the advantage of our approach over previously utilized frameworks for representation learning in the brain (“context-dependent computation”, Mante et al 2013)
> * Demonstrate that **transformers are ideally suited for disentangling representations**, achieving linearly decodable, orthogonal representations at the theoretically minimum possible number of tasks ($N_{task} \geq D$). This is of particular relevance for the machine learning community at large, since there is a lot of interest in what makes transformers so good at world understanding.
>
> While judgments about novelty are always subjective, and as acknowledged in the manuscript we build our work on the important contribution of Johnston & Fusi (2023), **it is hard for us to see how these advancements constitute incremental progress, since Johnston & Fusi (2023) focused on feedforward architectures that cannot handle noisy dynamical tasks, and did not provide any theoretical proofs for disentanglement or, as a matter of fact, abstractness.**
>
> As a side note, we would like to note that the experimental framework that we utilized was also used in [Maziarka et al. (2023)](https://dl.acm.org/doi/10.1007/978-3-031-26387-3_38) which has a very similar setting to Johnston & Fusi (2023) (generate synthetic objectives based on some sampled ground truth) and arrived at similar conclusions (multi-task setting results in disentanglement in feedforward architectures), although less unequivocally. In addition, the idea that multitask learning can lead to better generalization has been around for decades (Caruana, 1997). Our personal inspiration stemmed from the fruit fly head direction system, where an explicit representation of head direction is enforced presumably due to its downstream usage by many circuits for flight control etc. (Wilson, 2023). **Our work is, to our knowledge, the first time that theoretical guarantees for disentanglement through multi-task learning in autoregressive architectures have been provided and experimentally confirmed.**

---

> ### Author Response · Authors · 2024-11-22
> **Response to reviewer (4/4)**
>
> ## Sparsity
> We now rigorously quantify sparsity in our trained networks using the sparseness metric suggested by the reviewer (Vinje and Gallant, 2000). We find that RNNs are consistently sparse across values for number of tasks $N_{task}$ and latent dimensionality $D$ (see Appendix A.12 in updated manuscript). Therefore we conclude that sparsity is a phenomenon that consistently emerges in our trained RNNs, an attribute they share with their biological counterparts.
> Furthermore, as we now note in Appendix A.12, we do not impose any regularization or other constraints for sparsity; rather it naturally emerges from the optimization objective and network architecture. We would like to note though that **while sparsity is an interesting emergent phenomenon in our networks, our work presents much more far reaching and interesting findings than sparsity, as recognized by all other reviewers.**
>
> ## More nonlinear encoding
> In new experiments, we replace the ReLUs in the encoder with power-law (quadratic) nonlinearities (Appendix A.5 in revised manuscript). We find no impact on generalization performance. Therefore, we conclude that our setting is robust to the choice of encoder nonlinearity, even when the nonlinearity is not injective, going beyond our theoretical proofs. We have also cited the paper suggested by the reviewer ([Alleman, Lindsey & Fusi 2024](https://arxiv.org/abs/2401.13558)), as motivation for these experiments. Finally, we would like to clarify that the observation map is determined by the environment, not the brain, therefore the units of the encoder are not modeling biological neurons.
>
> Overall, we feel that these new experiments enhance the overall presentation of our paper, and quantify aspects (e.g. sparsity) that were not quantified before.

---

> ### Comment · Reviewer_Mvb9 · 2024-11-27
> **Questioning the framing of this work as 'multi-task learning'**
>
> I appreciate the authors’ thorough responses, the additional experiments, and the revisions they implemented in their submission. However, my primary concern remains: the **supervised paradigm** fundamentally limits the work's biological plausibility and its relevance to real-world, unsupervised scenarios.
>
> ---
>
> Below are my detailed responses to specific points made by the authors:
>
> ## 1. Framing this work as “multi-task learning” is misleading
>
> The paper frames multiple binary classification tasks as multitasking, which I find misleading.
>
> > They also point out that related work by Huh et al (2024) met this diversity requirement, by training networks on multimodal tasks
>
> The multimodal example I mentioned is not central to my critique. Multitasking does not require multimodality: rather, it requires distinct but related tasks that differ in their objectives.
>
> > By diversity of task demands we refer to whether the decision boundaries span the latent space … Task diversity is defined by the dimensionality of the subspace they span in the latent space
>
> This definition does not align with the standard understanding of task diversity in multitask learning. Tasks that share the same objective—such as binary classification across different decision boundaries—do not constitute diverse tasks in the traditional sense.
>
> For example, in a single-modality multitask learning scenario, such as image-based tasks, diversity arises from fundamentally different objectives: identifying the primary object in the image (*Object Recognition*), predicting bounding boxes around objects (*Object Localization*), classifying the type of scene (*Scene Understanding*), estimating pixel depth (*Depth Estimation*), and reconstructing pixel values (*Reconstruction*). Despite using the same modality, each task in this example interprets the input differently and requires specialized processing.
>
> In contrast, this paper's setup appears more like **single-task learning with multiple outputs**, as the tasks are essentially variations of the same binary classification task. While decision boundaries may span the latent space, the lack of task-specific diversity (i.e., distinct objectives) makes the classification framework less representative of what is traditionally understood as multitask learning.
>
> To address this, I recommend amending the paper's title to reflect its content more accurately.
>
> - The current title, "*Disentangling Representations through Multi-task Learning*," may create expectations of true multitask learning involving diverse objectives.
>
> - A more precise title could be: "*Disentangling Representations through Multiple Binary Classification Tasks*".
>
> While this revised title may sound less broad or ambitious, it more accurately represents the work and its contributions.

---

> ### Comment · Reviewer_Mvb9 · 2024-11-27
> **The limited relevance of disentanglement through strongly supervised frameworks**
>
> ## 2. Disentanglement in unsupervised settings
>
> > modern self-supervised learning involves converting an unsupervised objective (“scene understanding”) into a well-posed supervised objective (image patch-filling) (see l. 1909-16 in the manuscript)
>
> I appreciate the mention of modern self-supervised learning techniques, but I find this tangential to the discussion. The authors' approach is fundamentally supervised, and drawing parallels with self-supervised methods may introduce unnecessary conflation between the two paradigms. While self-supervised learning can incorporate elements of supervision through engineered objectives, it fundamentally differs from the explicitly supervised framework employed here. (Minor: It also appears the cited line references [1909-16] may be incorrect.)
>
> > The same logic can be extended to other modalities, for example in language, when predicting masked words that rely on a common latent sentence “meaning”.
>
> This example of masked word prediction in language models again seems tangential. My critique is not about the applicability of disentanglement to other modalities, but rather the reliance on a supervised framework in this paper. The focus should remain on the relevance of the current setup to disentanglement in unsupervised or weakly supervised settings, which this paper does not address.
>
> > To uniquely identify a point in D dimensional space, one needs distance information from at least D boundaries, i.e. N_task ≥ D. This is exactly the condition we find for linear decodability in our setting (eqs 3 and 4)
>
> Thank you for clarifying this. The linear relationship between the number of necessary tasks and input dimensionality is indeed less problematic than I initially thought. However, the reliance on supervision still limits the broader applicability of this work to real-world, unsupervised scenarios. The strength of disentangled representations lies in their emergence from data without requiring extensive supervision, and the current paradigm misses this key point.
>
> > the only thing provided to the networks are binary decisions driven by that underlying structure, not the underlying structure itself
>
> Indeed, the underlying structure is not directly provided, but the supervision signals effectively encode it through the binary decisions spanning the latent space. Proving this theoretically is neither surprising nor interesting. This approach positions the paper closer to the fully supervised end of the learning spectrum. Within supervised learning, the reliance on many supervision signals spanning the entire latent space limits the paper's relevance to unsupervised learning, which has been an essential focus of disentanglement research for decades.
>
> > Conflation of “competence at N tasks” with “receiving feedback signals” on N tasks
>
> From the perspective of building AI systems that learn to disentangle representations, your results appear to advocate for providing extensive supervised signals across all tasks (please correct me if not). Is this the most scalable or realistic approach? This reliance on supervision restricts the relevance of the findings for advancing models capable of disentanglement in real-world, minimally supervised scenarios.
>
> > They can originate internally, for tasks that the organism is already competent at.
>
> This raises an important question: How does the organism become competent in these tasks to begin with? Supervised or unsupervised learning? Without addressing how such competence is acquired, the paper does not provide actionable insights for models that have to learn disentangled representations from scratch. This omission limits the real-world applicability of the results, leaving them speculative at best.
>
> > The reviewer mentions that this multi-task learning setting might be not biologically relevant, as organisms rarely receive such rich feedback from the environment.
>
> To clarify, my primary concern lies in the reliance on a supervised paradigm. While I question the biological plausibility of this feedback-rich setup, my critique extends beyond biological relevance to the fundamental limitation of this work in advancing disentanglement beyond supervised frameworks.

---

> ### Comment · Reviewer_Mvb9 · 2024-11-27
> **Incremental contributions within a supervised paradigm**
>
> ## 3. Clarifying my comment regarding incremental progress
>
> I appreciate the authors highlighting how their work extends Johnston & Fusi (2023) and agree that it provides several valuable results beyond that study. However, my central critique remains unchanged: the **supervised paradigm** fundamentally limits the scope and impact of this line of work.
>
> Johnston & Fusi (2023) acknowledge in their discussion the inherent limitation of their supervised framework and emphasize the importance of future work moving beyond this artificial setup. By building on Johnston & Fusi (2023) without transitioning to an unsupervised framework, the current submission remains firmly grounded in the same supervised paradigm. While the paper refines and extends Johnston & Fusi (2023), it does not take the critical leap toward unsupervised disentangled representation learning that the field has been striving for.
>
> ## Final thoughts
>
> Researchers grappling with the complexities and challenges involved in unsupervised disentanglement recognize that advancing the field requires adhering to unsupervised, or at least weakly supervised, frameworks, which have been the focus of this area for decades. From this perspective, I maintain that the present work—while rigorous and well-executed—is incremental in essence, as it does not address the broader challenges or opportunities for unsupervised disentanglement.

---

> ### Author Response · Authors · 2024-12-02
> **Our theory is agnostic to the specific learning paradigm, focus of our work**
>
> **We are glad our responses resolved a lot of the reviewer’s previous confusions and contentions**, including the concern about requiring an abundance ($N_{task}>>D$) of tasks, the claim that we directly hint the underlying latent structure to the network, and that our work constitutes incremental progress over Johnston & Fusi (2023), and that **the reviewer acknowledged that our work is rigorous and well-executed.**
>
> We will address some contentions that seem to persist or have emerged in their new round of responses, but first it would be useful to establish what our work is and what it is not about. **More generally, it seems that the reviewer has a fundamentally different view of what our paper should be. Even though we understand their particular interests, we think we should be allowed the freedom to set our own goals for our paper.**
>
> ## Focus of our work
>
> > The focus should remain on the relevance of the current setup to disentanglement in unsupervised or weakly supervised settings, which this paper does not address.
>
> **While we appreciate and share the reviewer’s interest in the area of unsupervised disentanglement, our focus is not in this area.** Instead, we take a different perspective, **focusing on providing a formal link between competence at multiple tasks and disentanglement**, and experimentally confirming and extending the theory on tasks canonical in the cognitive neuroscience literature. While we recognize that proposing **new algorithms for unsupervised disentanglement** is a particularly important area of research, it **is not the only important question when it comes to disentangled representations** (see section *“Multiple ways towards understanding disentangled representations”* below).
>
> In addition, our theory goes beyond our experiments on supervised disentanglement, because it does not assume how competence at tasks was achieved, leaving the door open for representations that have been learned through supervised, self-supervised, or unsupervised means; our supervised experiments are simply the fastest way to get good performance at multiple tasks in a tractable manner, and test our theory. Therefore, **overly focusing on the supervised nature of the experiments misses a key message of our study.**
>
> >The strength of disentangled representations lies in their emergence from data without requiring extensive supervision, and the current paradigm misses this key point.
>
> Our theory is agnostic to the specific learning paradigm by which disentangled representations emerge. **It provides a formal argument why disentangled representations may emerge in a system: because the system is competent at multiple tasks.** Key strengths of disentangled representations include the fact that they separate latent factors present in the real world along orthogonal directions, enabling OOD generalization, efficient downstream learning etc, therefore providing any guarantees on their emergence is valuable.
>
> >How does the organism become competent in these tasks to begin with? Supervised or unsupervised learning?
>
> This is an important question, but given the reviewer’s frequent references to neuroscience, they surely understand that it cannot be fully addressed within a single study. Also, it is not the focus of this study.
>
> >the paper does not provide actionable insights for models that have to learn disentangled representations from scratch.
>
> We have to respectfully disagree here. Our work shows that if you train a network from scratch on a set of objectives, the network learns a disentangled representation of the latents that are spanned by these tasks. Our experiments are on supervised learning, but as mentioned throughout our responses and in the paper, the obvious next step is scaling that to self-supervised learning in future work.
>
> ## Self-supervised learning
>
> The connection between our framework and self-supervised learning is not tangential, but deep and promising. As we mentioned in the section *"Regarding the feasibility of disentanglement"* above, the two frameworks share a common structure of an underlying latent truth (e.g. objects in an image and their relationships), and solving each one of the objectives (e.g. filling each one of the missing image patches) acts synergistically in understanding the latent space as a whole. Same logic applies to predicting masked words that rely on a common latent sentence “meaning”. Our study provides guarantees on the disentanglement of representations learned as a function of the number of objectives solved in parallel. As such, self-supervised is a natural extension of our work, and we will be testing how the main findings of our theory generalize in this setting. The same points are mentioned in the new l. 1962-70 of the final version of the manuscript (apologies for the wrong line numbers before, as we added some more things in the meanwhile).

---

> ### Author Response · Authors · 2024-12-02
> **Our title is not misleading, etc**
>
> Other points:
>
> >From the perspective of building AI systems that learn to disentangle representations, your results appear to advocate for providing extensive supervised signals across all tasks (please correct me if not).
>
> We are not trying to advocate in favor of scaling supervised learning here. **Supervised training is our way of getting to our goal of competence at $N_{tasks}$ efficiently, and it is not at all our suggestion of how to scale up disentanglement efforts.** Self-supervised approaches offer much more promise at that, as just mentioned.
>
> > While I question the biological plausibility of this feedback-rich setup, my critique extends beyond biological relevance to the fundamental limitation of this work in advancing disentanglement beyond supervised frameworks.
>
> As just mentioned, we are not trying to suggest scaling up disentanglement through supervised training efforts. And there’s more than one way to advance the field of disentangled representations, for more see section *“Multiple ways towards understanding disentangled representations”* below.
>
> >but the supervision signals effectively encode it through the binary decisions spanning the latent space. Proving this theoretically is neither surprising nor interesting
>
> This looks like one of the main findings of our work. Judgements as to whether a certain result is surprising or not are of little practical value, especially in hindsight.
>
> ## Framework is multi-task learning by definition
>
> The reviewer claims that calling our framework multi-task learning is misleading. At the risk of sounding like a broken record, we will reiterate the definition of multi-task learning from the formative work of [Caruana (1997)](https://link.springer.com/article/10.1023/A:1007379606734), which is widely accepted:
>
> > Multi-task learning is defined as the “learning of [related] tasks in parallel while using a shared representation”
>
> Therefore, our work neatly falls within multi-task learning. Furthermore, we clearly stated our definition of diversity of tasks, which is mathematical, well-motivated and easily understandable by anyone. Tasks are diverse when they engage a lot of latent factors. The less latent factors that are engaged, the less diverse the set of tasks. There’s different ways to get to task diversity (i.e. engage more latents), including multimodality, different vision objectives that the reviewer references, or the way we sample classification lines here. Our way happens to be clean and amenable to theory.
>
> Regarding the title, while we appreciate the reviewer’s suggestion, we will keep it as is. This is not because the current title is more ambitious (why is multi-task learning ambitious?), but because it better reflects the nature of our work. In principle, it doesn’t have to be binary classification tasks, and in practice we also show that our framework extends to integration-to-bound tasks where networks directly report their confidence, not just a binary decision (see Appendix A.11). **The importance of the confidence of a network’s output is a recurring theme in machine learning (e.g. knowledge distillation etc.). We here show that it fundamentally connects to how neural networks construct world models, either directly (integrate-to-bound task) or indirectly (classification task).**

---

> ### Author Response · Authors · 2024-12-02
> **Reviewer seems to hold the view that only unsupervised disentanglement is not incremental**
>
> ## Regarding incremental progress
>
> We are glad that the reviewer now seems to agree that our work is not incremental with regards to Johston & Fusi (2023). The title that the reviewer chose for this section of their response *“Incremental contributions within a supervised paradigm”* can be a bit confusing, but based on context we will assume it means that they think our work is incremental because our experiments are on supervised learning; not because our results are incremental with regards to other work that used supervised learning for their experiments (e.g. Johnston & Fusi (2023)).
>
> Just to reiterate, **the supervised paradigm is simply our way to ensure that the networks are competent at multiple tasks which is a requirement of our theory, and secondary to the main message of our work, which is that competence at $N_{task} \geq D$ tasks leads to disentangled representations.**  The theoretical insights stand separately from the specific paradigm used, and apply to networks that have achieved competence at multiple tasks through other means, including unsupervised learning.
>
> We acknowledge that a limitation and future direction of our work would be to apply our setting to richer tasks (e.g. self-supervised learning) in the paper and throughout the rebuttals. In the limitation section of our paper (now appendix A.15) we also mention other limitations, such as that factorization and feature-based generalization might not be the best strategy always. We cannot make any more modifications to the paper at this point, but we will make sure to connect our mentions of self-supervised learning to the importance of advancing the field of unsupervised learning of disentangled representations. Finally, the discussion in Johnston & Fusi the reviewer is referencing is centered around offline prediction for more biologically plausible versions of the multitasking framework, which we already addressed in the previous round of rebuttals (section *"Biological plausibility of multi-task learning"*).
>
> >I maintain that the present work—while rigorous and well-executed—is incremental in essence, as it does not address the broader challenges or opportunities for unsupervised disentanglement.
>
> **It seems from the above that while the reviewer no longer considers our work incremental in regards to Johnston & Fusi (2023) as they did before, they now consider it incremental because it doesn’t focus on unsupervised disentanglement.** This brings us to the next point.
>
> ## Multiple ways towards understanding disentangled representations
>
> Throughout the rebuttals, the reviewer seems to **argue strongly in favor of a particular approach for advancing the field of disentangled representations**: in particular, they seem to argue that if one is to advance the field of disentangled representations, they should focus on answering the question of how neural networks learn disentangled representations through unsupervised learning (also see section “Focus of our work” above). This seems quite restricted to us. While we recognize the importance of and the complexities associated with the unsupervised learning of disentangled representations, **this clearly cannot be the only important research question.** Two other key ways to advance the field of disentangled representations that come to mind are:
>
> 1. **Provide theoretical guarantees** for the existence of disentangled representations under some conditions - **our work makes important progress on that front**, irrespective of how these representations were learned and
>
> 2. Define new metrics that can **better detect** disentangled representations in existing networks. Our study informs thinking in this space, by arguing in favor of a mixed representation approach to disentanglement (fig. 4), while most metrics to date look for axis-aligned representations. **Our work, along with others** (e.g. Johnston & Fusi, 2023) **suggests that we might be missing disentanglement in networks that perform well in many tasks, due to the metrics used.**
>
> We are sure that there will be many other possible ideas and research directions that will be useful as the field of disentangled representations moves forward, and we are excited for the future of the field.
>
> Finally, we genuinely hope that our remaining interactions with the reviewer are constructive, and that the reviewer takes the time to carefully consider our responses.

---

> > ### Comment · Reviewer_Mvb9 · 2024-12-03
> >
> > I appreciate the authors' efforts and their thorough responses.
> >
> > ## On Theoretical Guarantees for Disentangled Representations
> >
> > > Provide theoretical guarantees for the existence of disentangled representations under some conditions
> >
> > The authors emphasize providing theoretical guarantees for the emergence of disentangled representations under specific conditions. This is indeed the crux of the discussion. The impact of such theoretical contributions fundamentally depends on the viability and applicability of these conditions. In my view, guarantees in unsupervised or minimally supervised settings would be far more impactful. While the authors stress the role of competence rather than supervision, the theoretical framework and most of the experimental results explicitly rely on supervised signals to achieve this competence. Without addressing how such competence could emerge in real-world, unsupervised scenarios, the broader significance of the results remains unclear.
> >
> > ## On Mixed Representation Approaches and Metrics
> >
> > > Our work, along with others (e.g. Johnston & Fusi, 2023) suggests that we might be missing disentanglement in networks that perform well in many tasks, due to the metrics used.
> >
> > I commend the authors for advocating a mixed representation approach to disentanglement. It aligns with ongoing discussions in both neuroscience and machine learning. However, this perspective is not new and has already been explored extensively in the literature. For instance:
> >
> > - In neuroscience, the concept of “*untangling*,” or explicit (e.g., linear) decodability of some ground truth signal from representations, has been around for nearly two decades ([DiCarlo & Cox, 2007](https://www.sciencedirect.com/science/article/abs/pii/S1364661307001593); [DiCarlo et al., 2012](https://www.cell.com/neuron/fulltext/S0896-6273(12)00092-X)).
> > - In machine learning, metrics such as "*Informativeness*" ([Eastwood & Williams, 2018](https://openreview.net/forum?id=By-7dz-AZ)) or "*Explicitness*" ([Ridgeway & Mozer, 2018](https://papers.nips.cc/paper_files/paper/2018/hash/2b24d495052a8ce66358eb576b8912c8-Abstract.html)) have already addressed similar ideas, with clear connections to the authors’ definition of in-distribution (ID) performance. Additionally, the authors' framing of out-of-distribution (OOD) generalization closely resembles what [Lachapelle et al., (2023)](https://openreview.net/forum?id=R6KJN1AUAR) have described as "*Cartesian-product extrapolation*."
> >
> > The authors suggest that existing metrics might fail to detect disentanglement in networks performing well on many tasks. Yet, they do not cite any of the aforementioned well-known works in this area, which demonstrate that researchers in both neuroscience and machine learning are already aware of these limitations. Consequently, while the current paper contributes to the broader discussion, it is unlikely to uniquely inspire new research in this area. That said, this work adds momentum to the ongoing shift toward embracing mixed-selective representations, which is a valuable direction.
> >
> > ## On Multi-Task Learning
> >
> > > The reviewer claims that calling our framework multi-task learning is misleading. At the risk of sounding like a broken record, we will reiterate the definition of multi-task learning from the formative work of Caruana (1997), which is widely accepted: Multi-task learning is defined as the “learning of [related] tasks in parallel while using a shared representation”
> >
> > Multi-task learning has traditionally involved distinct but related objectives, as demonstrated by the three explicit examples in section 2 of Caruana (1997): 1D-ALVINN, 1D-DOORS, and Pneumonia Prediction. Each of these involves tasks with separate yet interconnected objectives. Similarly, more recent overviews, such as [Ruder, (2017)](https://arxiv.org/abs/1706.05098), also emphasize multi-task settings where objectives are clearly delineated.
> >
> > In contrast, the authors' framework employs multiple binary classification boundaries within the same latent space, which diverges from the traditional understanding of multitasking. This difference may confuse readers who expect multi-task learning to involve distinct objectives. I encourage the authors to clarify this distinction in their paper to better align with established conventions and avoid potential misunderstandings.
> >
> > ## Final Thoughts
> >
> > I appreciate the authors' hard work and engagement during the rebuttal process. However, the strong reliance on supervised signals in the current work limits its applicability to broader, real-world settings where supervision is scarce. Moving beyond this paradigm toward weakly supervised or self-supervised approaches, as the authors themselves discuss, would significantly enhance the relevance and impact of this line of work.
> >
> > Given these considerations, I will maintain my score.

---

> ### Author Response · Authors · 2024-12-04
> **Unfortunately, the reviewer repeats the same objections after we addressed them**
>
> We thank the reviewer for their response, and **we are glad that they now agree that testing our theory on self-supervised learning is an actionable next step.**
>
> We are not the first to come up with the idea for mixed disentangled representations (e.g. we cite Johnston & Fusi 2023), however a lot of influential work focuses on axis-aligned representations (Higgins et al. 2017, Kim & Mnih 2018, Locatello et al. 2019); we want to advocate in favor of mixed representations before axis-alignment becomes an indispensable part of the definition of disentanglement in the CS community.
>
> Regarding multi-task learning, we will be sure to outline our definition of task diversity, which is mathematical in nature.
>
> **When it comes to our theoretical guarantees and the origin of competence at multiple tasks however, it seems that the reviewer is still misunderstanding our work, and repeating their previous objections that we just addressed** (e.g. section [*Addressing reviewer Mvb9’s misunderstanding of a key contribution of our work*](https://openreview.net/forum?id=yVGGtsOgc7&noteId=ez9jQvXiOy)).
>
> ## Our theoretical guarantees apply to representations learned through unsupervised means
>
> > the theoretical framework and most of the experimental results explicitly rely on supervised signals to achieve this competence
>
> > In my view, guarantees in unsupervised or minimally supervised settings would be far more impactful.
>
> As mentioned several times in our rebuttal, and contrary to the reviewer’s assertion, **our theoretical framework does not rely on supervised signals** to achieve competence at tasks; it just assumes competence. **The conditions for our theory are not related to the learning paradigm** (supervised or unsupervised), **but rather to the geometry of the tasks on which competence is achieved.** Therefore, **our theory also covers cases where competence was achieved by unsupervised or minimally supervised means.**
>
> ## Focus of our work
>
> > Without addressing how such competence could emerge in real-world, unsupervised scenarios, the broader significance of the results remains unclear.
>
> **Addressing how competence at tasks can emerge in real-world settings through unsupervised means is an entirely different question** than our focus here. As we mentioned before, there are more than one ways to make progress towards understanding disentangled representations (see corresponding section [here](https://openreview.net/forum?id=yVGGtsOgc7&noteId=ikR5XGez9y)), and **we focus on providing theoretical guarantees on disentanglement, and rigorously testing them in tractable settings** (see [here](https://openreview.net/forum?id=yVGGtsOgc7&noteId=QjrzaVKFN0)). Testing our theory in self-supervised settings would introduce significantly more overhead, that goes far beyond the scope of a single study (see below).
>
> ## Our experimental results already extend past our theoretical guarantees
>
> > Moving beyond this paradigm toward weakly supervised or self-supervised approaches, as the authors themselves discuss, would significantly enhance the relevance and impact of this line of work.
>
> In our discussion that the reviewer quotes, we were referring to the experiments specifically. While it remains to be seen how the theoretical insights generalize to self-supervised settings, this is not because of the learning paradigm per se, but because of the different **task geometry** implied by the specific application (e.g. image patch-filling, next-token prediction, iterative de-noising). **We have experimentally shown that our findings generalize to other task geometries beyond what is strictly guaranteed from our theory** (e.g. see Appendix A.10, Appendix A.5), **which is promising for the broader application of our findings across task geometries, including those that might be encountered in self-supervised settings** (also mentioned under *“New research directions”* in the [main rebuttal](https://openreview.net/forum?id=yVGGtsOgc7&noteId=0GglZZbzCV)). Extending our results to self-supervised settings would likely require the introduction of new datasets that can take advantage of the temporal dimension, like dynamic versions of dSprites; **thus, this would constitute an entirely new paper on its own.**
>
> ## Summary
>
> Overall, it appears that the reviewer has a specific vision of what our paper should address, which differs from the actual scope and contributions of our work. **It remains unclear what the reviewer expects in response to their repeated objections regarding “strong reliance on supervised signals”, despite our repeated clarification that supervised signals are not necessary for our theory;** competence on multiple tasks is. We do not agree with the reviewer, and it seems that they remain unwilling to reconsider their stance despite thorough clarifications.

---

### Official Review · Reviewer_w8Z5 · 2024-11-03

**Soundness:** 3
**Presentation:** 3
**Contribution:** 3
**Rating:** 6
**Confidence:** 4

**Summary:**

The paper presents theoretical and empirical results showing that, in multi-task evidence aggregation classification tasks, representations become disentangled as the number of tasks N_task greatly exceeds the input dimensionality 𝐷 when the agent solves the tasks optimally.

**Strengths:**

1. Originality: the study of the emergence of disentangled representation in temporal tasks and models is relatively new.
2. Quality: the empirical validation is comprehensive.
3. Clarity: the theory is well explained. The background section is also very nicely written and thorough.
4. Significance: The paper addresses the key problem of learning the world model in representation learning and also discusses in detail the biological relevance. The results offer meaningful insights for future research on disentangled representation in both artificial systems and the brain.

**Weaknesses:**

1. The theory assumes that agents are optimal multi-task classifiers, which may not be achievable in realistic settings where the input dimension D is already large and the number of tasks N_task >>D. This raises questions about the practical relevance of the regime considered in the paper. Additionally, it’s difficult to imagine a large number of orthogonal tasks in real-world settings. For instance, in the dSprite dataset, as the authors noted, how could a meaningful, larger set of tasks be constructed, and what implications would this have for the applicability of the framework?
2. The connection to neuroscience also appears somewhat speculative. The authors could clarify how their findings might directly relate to brain function. For example, can the theory predict the relationship between the disentanglement of neural representations and the level of noise or variability in different brain regions?
3. While explaining zero-shot generalization capabilities in artificial neural networks (ANNs) is a central claim, this remains a substantial challenge for most networks. The theory should address whether prior models fail in zero-shot generalization due to insufficient N_task and how practitioners could estimate an adequate N_task, especially in settings where underlying latent factors are unknown, making it impractical to sample random classification tasks as in the paper.

**Questions:**

1. Most examples and empirical validations focus on low-dimensional cases (e.g., D=2). Could you provide examples using moderately higher-dimensional datasets, such as CIFAR or dSprites, to illustrate how N_task scales with D in these settings?
2. For a given D, what is the maximum number of tasks N_task can be learned with low error? Additionally, how do the test and generalization errors scale with N_task if we keep the number of training data in at a realistic scale?

---

> ### Author Response · Authors · 2024-11-22
> **Response to reviewer (1/2)**
>
> We would like to thank the reviewer for the in depth review and comments.
>
> Regarding the individual comments:
>
>
> ## Dimensionality of latents
> One thing that we might need to specify here is that $D$ is the dimensionality of the latents (e.g. features like ripeness, fruit identity, etc.), not the dimensionality of the observation (image). Therefore, $D$ is of much lower dimension than the dimensionality of the input, and only corresponds to underlying latent features that are relevant for the tasks.
>
> Also, while $N_{task} \gg D$ is the condition for orthogonality, the condition for linear decodability (abstractness), for which equations (4) and (5) hold, is $N_{task} \geq D$. In addition we find empirically that disentanglement (orthogonality) is also achieved for $N_{task} \geq D$, particularly in transformers. Therefore both our theory and experiments point to a linear relationship between the number of tasks required to disentangle factors, and the dimensionality of said factors $D$. We believe that this makes our setting applicable to real life settings, in particular because as noted in the discussion, there exists an interplay between $N_{task}$ and $D$: multi-task learning guarantees that neural networks will learn to disentangle the subspace within the $D$-dimensional latent space that is spanned by the decision boundaries. In other words, we are guaranteed to learn disentangled representations of all latent factors that are involved in the classification tasks, which is a very powerful guarantee. Furthermore, the tasks themselves do not need to be orthogonal; just to collectively span the latent subspace of interest; then the latents can become orthogonal. Finally in regards to optimality, we expect near-optimal performance to be achieved by any efficiently trained network (e.g. one trained with backpropagation), if adequate training samples are provided. Sub-optimal classifiers with zero-mean error are also handled in Corollary B.7 in the Appendix.
>
> Regarding dSprites, the latent factors for this dataset are $D=5$: Shape, Scale, Orientation, X and Y Position. Therefore, we would need at least 5 binary classification tasks that collectively depend on all of these variables (example task: $+1$ if we have a shape of scale$>0.7$ centered at $(x,y)$ where $x>y$, $0$ otherwise).
>
> ## Connection to neuroscience
> We thank the reviewer for allowing us to elaborate on this topic. Disentangled (or abstract) representations have been discovered in a multitude of brain areas, serving processes as disparate as memory (Boyle et al, 2022), emotion (Saez et al, 2015) and decision making (Bongioanni et al, 2021). Therefore, it seems natural to ask how such disentangled representations may come about in neural networks, including the brain. A main takeaway from this study is that parallel processing (multi-task learning) enforces the learning of such representations, with mathematical guarantees. Incidentally, it turns out that the cortex itself is built for parallel processing (Hawkins et al, 2019), with individual columns thought to be able to perform computation independently. Therefore, we believe that might be how the brain might build disentangled world models that allow efficient downstream out-of-distribution generalization (as we show). Of course this is still an assumption, and we hope this study motivates experimental neuroscientists to have a closer look at computation in cortical columns. Furthermore, other brain areas, such as the thalamus, could be involved in this parallel processing, as we mention in the manuscript.
>
> In regards to the second point, it is hard to quantify the level of noise in the brain. Furthermore, by noise in our setting we refer to noise in the latents (perceptual noise) not noise in the brain (neural noise), which in vision would correspond to viewing an object under different angles/lighting positions etc, which will allow the agent to “localize” the object in latent factor space. This idea of building faithful visual representations by being robust to perturbations like rotation etc. has been around in the literature for a while (e.g. DiCarlo, J. J., Zoccolan, D., & Rust, N. C. (2012). How does the brain solve visual object recognition? Neuron, 73(3), 415–434).

---

> ### Author Response · Authors · 2024-11-22
> **Response to reviewer (2/2)**
>
> ## Determining the number of tasks for disentanglement
>
> The reviewer raises a great point. Indeed most of the time ANNs are not trained with multi-task learning, which might explain their poor out-of-distribution generalization capabilities. Note, however, that our framework neatly falls within the modern machine learning practices, and more specifically self-supervised learning (see l. 1908-16 in the manuscript). In short, self-supervised learning involves converting an unsupervised objective (“scene understanding”) into a well-posed supervised objective (image patch-filling). Filling missing patches *is* a multi-task objective, with a common, shared latent space (objects present in an image, and their relations), and solving each one of the objectives acts synergistically in understanding the image latent space as a whole. Therefore in this case, if a lack of out-of-distribution generalization is detected, our theory proposes a simple remedy: increase the number of masked out patches that have to be predicted from the image, which will result in a construction of a more descriptive latent space of the features underlying images. Of course, that prediction has to be tested, which would be an interesting topic for future research.
>
> Another dimension we would like to point out here is that sometimes we don’t necessarily want to uncover the entire latent space. As we note in the discussion, our work reveals an interplay between the number of disentangled latents and the number of tasks. We guarantee that multi-task learning will uncover the latents that are spanned by the tasks we train for, which in many cases is adequate for the tasks under consideration. In other words, there is no need to uncover all possible latents, but only the ones that are relevant for the tasks we are interested in. In that sense, it is the tasks that define the set of useful, disentangled latents, and not the other way around.
>
> Regarding to the reviewer’s questions:
>
> ## Higher $D$
>
> While a lot of the main text figures focus on the $D=2$ case, a main reason we do this is that for $D=2$ it is easier to visualize the representations. However we also conduct experiments in high-dimensional latent spaces (up to $D=10$) in Figure 5b, experiments that confirm our theoretical findings, showing excellent out-of-distribution generalization (abstractness) for $N_{task} \geq D$ in transformers and RNNs. Extending these findings to more real-world datasets (like CIFAR or dSprites) would be an interesting direction. However these datasets lack the dynamical structure of the tasks we focus on here, where there is some background truth buried in perceptual noise that can be uncovered if multiple independent views of the object are afforded. Therefore, a promising direction would be to develop novel datasets like dynamic versions of dSprites that afford multiple views of the same object under different lighting conditions/angles. However, constructing and testing such datasets with CRNN or ViT architectures would be an independent study of its own, and far beyond the scope of this paper.
>
> ## Higher $N_{task}$
>
> In general, learning more classification tasks with low error is not a problem. In particular, increasing the number of tasks while keeping $D$ constant does not negatively affect the performance of these tasks (we tried values of $N_{task}$ up to $192$). Similarly, if the number of training samples is kept constant, increasing the number of tasks does not negatively affect performance. Each sample provides information useful for each task.

---

> ### Comment · Reviewer_w8Z5 · 2024-11-25
>
> Thank you for the detailed rebuttal. I appreciate the clarification regarding D and the number of tasks. However, I still hold the view that the current implications of the paper are vague. Specifically, the main message that multi-task learning enforces disentangled representations in the brain feels overly generic and lacks causal evidence. The paper would be much stronger if it could provide any quantitative predictions. That said, I recognize the valuable contributions this paper makes and believe it remains acceptable in its current form. I will maintain my score.

---

> ### Author Response · Authors · 2024-11-26
> **Thank you**
>
> We thank the reviewer for the kind response.
>
> While we share the interest in potential causal connections to neuroscience, we emphasize that **this work focuses on representation learning** rather than computational neuroscience, and that we cannot provide causal evidence about brain function – this would require experimental validation through wet lab studies. **Our primary goal is to advance understanding of how multi-task learning enforces generalizable representations in neural networks, with theoretical guarantees, independently of whether such processes occur in the brain in this exact way or not.** We hope that our work provides motivation and a theoretical foundation that others (experimentalists or theoreticians) may use to test or explore connections to brain function in future research.
>
> Once again we would like to thank the reviewer for their thoughtful feedback, and for supporting our work.

---

### Official Review · Reviewer_2Aef · 2024-11-04

**Soundness:** 3
**Presentation:** 3
**Contribution:** 3
**Rating:** 6
**Confidence:** 2

**Summary:**

This paper explores the important question of how abstract (linear and approximately orthogonal) and disentangled (orthogonal without the necessity of linearity) representations can emerge in biological and artificial agents.
The authors present both theoretical and experimental results demonstrating that multi-task learning, specifically within the framework of evidence aggregation classification tasks, can lead to the development of such representations.

**Strengths:**

The paper presents theoretical results that establish specific conditions—relating to the number of tasks, input dimensionality, input noise, and more—that lead to the emergence of abstract and disentangled representations in agents solving multi-task evidence aggregation classification tasks. The authors conduct thorough experiments across several architectures (RNNs, LSTMs, and transformers) to validate their theoretical results, showing that even architectures like GPT-2 can exhibit these properties. The work bridges AI and neuroscience, with potential explanations for how abstract, human-aligned representations might arise in artificial neural networks, making it valuable for both fields.

**Weaknesses:**

One notable limitation of this work is the assumption of factorization, as acknowledged by the authors. Additionally, the theoretical framework is tailored to a specific type of multi-task learning problem—evidence aggregation classification with linear decision boundaries—which may not capture the full diversity of tasks and decision boundaries that agents encounter in dynamic environments. It would be valuable to explore how these ideas generalize to other multi-task learning scenarios. Furthermore, the experiments focus on synthetic data with simple latent structures. While these experiments effectively support the theoretical results, testing the framework on slightly more complex data could provide further insights.

**Questions:**

- In Equation 6, could you clarify if $x_{in}$ is the encoded input, i.e., $f(x)$?

- Does the dimensionality of the latent state Z influence the results? Also, does the dimensionality of the encoded input play a significant role? Have these aspects been examined?

- Have you conducted experiments with other encoding functions, such as a quadratic mapping?

- In Figure 5b, the blue curve for GPT appears to increase at the end. Could you clarify the reason behind this behavior?

- Since the noise at each time step is independent, temporal information isn’t modeled in this setup. How might the results differ if a transformer with self-attention layers was used? Would such a setup still yield disentangled representations?

- On the other hand, if the inputs contained some temporal structure, how might this affect the nature of the representations?

- Have you tested the framework with much larger values of $N_{task}$ and $D$ (e.g., over 100)?

---

> ### Author Response · Authors · 2024-11-22
> **Response to reviewer (1/2)**
>
> We would like to thank the reviewer for the thoughtful and useful review.
>
> ## Main comments:
>
> While most of the experimental results are on linear classification boundaries in latent factor space, we would like to point out that the classification boundaries are rendered non-linear by the encoder $f$, therefore the network is de facto still performing non-linear classification tasks. In addition, in the appendix (A.10) we extend our framework to non-linear ($1/x$) classification boundaries, showing that abstract representations are also learned in this setting. Therefore, this generalizes scenarios where agents have to multiply latent factors to make decisions, like for example in situations where one latent corresponds to reward magnitude and the other to reward probability (Bongioanni et al, 2021). Finally, while we would like to extend our framework to more naturalistic tasks, like dynamic versions of dSprites that afford multiple views of the same object under different lighting conditions/angles, which would correspond to noise in the latents as in our setting, we are not currently familiar with any such datasets, and constructing and testing such datasets would be an independent study of its own, beyond the scope of this paper.
>
> ## Impact of dimensionality of network and of the encoder
>
> We have tested different sizes of networks (dimensionality of $Z$), ranging from $32$ to $512$ neurons. The dimensionality of $Z$ should be greater than $D$, to ensure that the network has enough capacity to capture all the latents, as well as perform other computations (average evidence, invert non-linear mappings, etc.). As a standard machine learning practice, we choose $Z$ to be a few times greater than the largest $D$ we would like to capture. We found that $64$ neurons were enough for $D$ up to $10$. To examine when the latent encoding fails, we decrease the number of neurons when $D=10$, i.e. when we need the most amount of neurons to disentangle all latents. For $N_{task} = 24$ and $N_{neu}=64$, median $r^2 = 0.9$, therefore representations are abstract. However when decreasing $N_{neu}=32$, median $r^2 = 0.52$, and when $N_{neu}=16$, median $r^2 = 0.19$, i.e. generalization performance is completely hampered. Therefore we conclude that while there is no strictly defined threshold, $N_{neu}$ should be a multiple of $D$ to ensure enough capacity to perform all the functions that the network does (encode the factors, invert nonlinear encoder, denoise etc.).
>
> Similarly, the output dimensionality of the encoder (let’s call it $E$) should be greater than the largest $D$, to avoid throwing away information at the encoding stage. We have tried all the way down to $E=D$, e.g. by feeding the latents directly to the network. Realistically, $E$ will be much larger than $D$, as an observation (e.g. image) typically has much larger dimensionality than its latents.
>
> ## More nonlinear encoding
>
> In new experiments, we replace the ReLUs in the encoder with quadratic nonlinearities. We find no reduction in OOD generalization performance (see Appendix A.5 in updated manuscript).
>
> ## Transformer architecture
>
> Self-attention (or simply, attention) layers are a central piece of the transformer architecture in all situations, and unrelated to whether a transformer received independent temporal information or not. Did the reviewer mean positional encodings here? If we would include positional encodings in the form of learnable parameters, while the noise is sampled IID, we would expect to see no improvement in performance, but no deterioration in terms of disentangled representations either. The transformer would simply not use the positional encodings, since they are not relevant for the execution of the task (the inputs can be treated as a set). Since the transformer has to learn that positional encodings are irrelevant, this might slightly slow down learning, depending on how strong the initialization of the positional encodings is, but the effect will likely not be noticeable.
>
> ## Noise correlations
>
> We have already shown that our setting works in the case of autocorrelated noise which introduces temporal structure to the inputs in section A.9 in the appendix. If we wanted to extend these findings to transformers, performance would benefit from including positional encodings, since now position matters (the inputs are no longer IID, and adjacent samples have correlated noise).

---

> ### Author Response · Authors · 2024-11-22
> **Response to reviewer (2/2)**
>
> ## Much greater values of $N_{task}$ and $D$
>
> We have tested values of $N_{task}$ up to $192$, with no negative impact on performance. Testing for much larger values of $D$ poses a problem, because of the increased computational runtime and memory resources associated with increasing $D$. Specifically, in our OOD generalization code, we perform quadrant splits which scale exponentially with $D$, making testing generalization performance for large $D$ difficult. However, we did train networks for $D=30$, $N_{task}=256$, $E=150$ and $N_{neu} = 256$ up to ~ $90$ percent classification accuracy in new experiments, and we also trained and tested networks with $D=20$, $N_{task}=128$, $E=80$ and $N_{neu} = 128$ resulting in great generalization performance  (median OOD $r^2 = 0.86$).
>
> ### About the more minor points:
> * Yes indeed, $x_{in}$ is the encoded input $f$ concatenated with the fixation input (see Fig. 2a). We have added this information in the paper
> * For the experiments with higher $D$, we resorted to randomly sampling classification boundaries. This naturally introduces some randomness in the performance of the networks, which is why RNN performance when $D=10$ can happen to be slightly higher than performance for $D=8$ (the two are not that different when $N_{task}$ is only 3), and also why variance in the fits for $D=8$ is high. In both cases, performance is far below what would be considered as good generalization performance, as expected by the theory.

---

> ### Author Response · Authors · 2024-11-29
> **Any other questions we can answer?**
>
> Dear Reviewer,
>
> Thank you for taking the time to review our paper and providing valuable feedback. We noticed that there hasn’t been further engagement with our rebuttals during the discussion period. As the discussion is nearing its conclusion, we wanted to kindly ask if there are any additional questions or concerns we could address to help clarify any remaining points.
>
> We greatly appreciate your efforts in this review process.
>
> Sincerely,
> The Authors

---

### Author Response · Authors · 2024-11-22
**Main response (2/2)**

## Biological plausibility of multi-task learning
**Reviewer Mvb9** mentioned that our framework may lack biological relevance because of its “reliance on supervisory signals”. **While our theory stems from *parallel processing*, i.e. multi-task learning, it is not contingent upon the parallel execution of multiple tasks, i.e. multitasking, or the receipt of such feedback in parallel.** Behaviorally, the agent need only perform one action, the one most appropriate to its current internal state (e.g. its level of thirst vs. hunger might control the slope of the decision boundary in 2D space of water & food). What we posit is that tasks that have been performed by the agent before and rely on the same input are still resolved somewhere in the brain, by the brain circuits (e.g. cortical columns Hawkins et al. (2019)) previously responsible for them, instead of the entire decision-making brain area focusing only on the current task (Mante et al., 2013). In other words, **our theory assumes competence at $N_{task}$ tasks, independently of when and how that competence was achieved**. We feel that this is a more natural way of thinking about how the brain manages different tasks, with older tasks still leaving traces somewhere in the brain ([Losey et al., 2024](https://www.sciencedirect.com/science/article/pii/S0960982224002987)), and biological agents are remarkable *because* they achieve high performance on many tasks. This theory is also related to the widely observed phenomenon of memory replay ([Foster & Wilson, 2006](https://pubmed.ncbi.nlm.nih.gov/16474382/)), or mental simulation of counterfactuals ([Jensen et al, 2024](https://www.nature.com/articles/s41593-024-01675-7)).

At the same time, we recognize that parallel processing is still an assumption, and the study already presents fundamental insights in the nature of generalization and world model construction in neural networks as is, independently from whether this exact process materializes in biological neural networks or not. Future work will explore the relationship between multi-task learning and interleaved training (see Appendix A.10 for preliminary results), which more closely mimics neuroscience experiments where new tasks may be introduced to the animal.

---

> ### Comment · Reviewer_Mvb9 · 2024-11-27
> **Core limitation: Reliance on a supervised paradigm**
>
> Since I was called out singularly in this segment of the authors’ global rebuttal, I feel it is necessary to respond directly.
>
> ---
>
> I appreciate the authors' efforts and their response, but I am genuinely struggling to understand the broader significance of this paper—and this line of work in general—where abstract representations are obtained through strong supervised signals. Could the authors clarify how their work contributes to advancing the field?
>
> The discussion of parallel processing seems peripheral to my original critique. To ensure clarity, let me restate my concern in concrete terms, highlighting why I find the study of disentangled representation learning impactful and where this paper appears to fall short:
>
> - **From a neuroscience perspective**: Abstract representations have been observed across various brain regions. It is an interesting question to explore how and why these representations emerge in evolved biological systems.
>
> - **From a machine learning and AI perspective**: Disentangled representations are hypothesized to improve efficiency and facilitate out-of-distribution generalization. Understanding how artificial systems can autonomously develop such representations is a valuable goal.
>
> The core issue with this paper is its reliance on supervised signals to achieve abstract representations. In my view, this undermines its relevance to understanding how biological systems develop such representations, as supervision is rare in real-world or naturalistic environments (outside highly controlled laboratory settings). Similarly, the supervised paradigm limits the practical utility of this work in developing artificial systems capable of learning disentangled representations directly from experience. In both neuroscience and AI, the scarcity of reliable supervision in real-world scenarios renders this approach less impactful.
>
> To illustrate this point, I'd like to reference a related study from ICLR last year ([Whittington et al., 2023](https://openreview.net/forum?id=9Z_GfhZnGH)), which the authors themselves cite. Whittington and colleagues showed that biologically plausible constraints, such as non-negativity of representations and a metabolic cost term, can lead to disentangled representations in a fully unsupervised framework.
>
> Their work is impactful because:
>
> - **In neuroscience**: It starts from biologically plausible assumptions, rather than starting with something contrived like multiple binary classification tasks, followed by forcing a post-hoc biological interpretation.
> - **In AI**: It demonstrates how these constraints lead to disentangled representations that outperform established benchmarks (e.g., beta-VAEs) on certain tasks while providing clear, actionable paths for future work to contribute to improving disentanglement in a way that is biologically grounded.
>
> Crucially, Whittington et al. operate in an unsupervised setting, addressing the challenges of both biological plausibility and real-world AI applications.
>
> My goal here is not to directly compare the two papers. Instead, I hope to convey, clearly and concretely, why I am struggling to see the significance of the current work. The key questions remain:
>
> - How does this paper help us understand the emergence of abstract representations in biological systems, which rarely receive supervision?
> - How do these results inform the design of models that develop abstract representations in real-world, unsupervised settings?
>
> If these questions are outside the scope of the authors' intended contribution, I would appreciate clarification on what specific insights this work provides to the broader community. Is the contribution primarily theoretical, with the aim of laying groundwork for future extensions toward more realistic and impactful scenarios? If so, how does this work pave the way for that future research?
>
> My struggle, again, is understanding how providing models with extensive supervised signals to achieve abstract representations could positively influence future research in disentangled representation learning. Given the significant gap between the paper's assumption of abundant supervision, versus real-world scenarios where supervision is rare, I am not yet convinced of its relevance. I would genuinely appreciate it if the authors could articulate how their work is meaningful despite this fundamental limitation, while staying focused on the core critique.

---

> ### Author Response · Authors · 2024-12-03
> **Clarifying a key reviewer misunderstanding on competence at tasks vs. supervision**
>
> We thank reviewer Mvb9 for their overview of Whittington et al., 2023, which is excellent work. To briefly address the key concerns related to the usage of supervised learning and its impact on the broader significance of our work:
> 1. Conceptually and theoretically, our proof on disentanglement in potentially suboptimal multi-task classifiers clarifies the mechanism by which **competence** on $N_{task}$ classifications implicitly represents latent variable estimator $\mu(t)$. **Competence in $N_{task}$ tasks does mean the ability to accurately assign likelihoods to $N_{task}$ classifications necessarily arose from supervised training signals**. A direct counterexample is pre-trained large language models that exhibit low perplexity/accurate likelihood assignment on novel few-shot tasks. We previously mentioned this point regarding **competence vs. supervision** in reviewer Mvb9’s previous round of individual rebuttals (see section *Conflation of “competence at N tasks” with “receiving feedback signals" on N tasks*), but we will further elaborate below.
> 2. Unsupervised learning objectives (as in Whittington 2023) are not mutually exclusive with applying the theoretical and experimental results of our work. A direct comparison of our work with Whittington et al, which has a different focus, approach and architectures might not be particularly useful here.
> ## Addressing reviewer Mvb9’s misunderstanding of a key contribution of our work
> To address reviewer Mvb9’s contention regarding supervision, we briefly outline a main contribution of our work here, to make it extremely easy to parse. Of course it’s impossible to capture the full nuance of our work in a few lines.
> * **Theoretical contribution**: we prove that competence in $N_{task}\geq D$ enforces abstract/disentangled representations.
> * **Experimental confirmation**: we confirm the key predictions of our theory with regards to $N_{task}$, effect of noise, etc in autoregressive models that achieve near-optimal (fig. 5a) competence at the tasks
> * **How we achieve competence at tasks**: multi-task learning with supervised labels. We choose this because it is the most straightforward way to become competent at multiple tasks, however our theory is not limited to this supervised framework. **Our theory establishes a formal link between competence at $N_{task}\geq D$ tasks and disentanglement, irrespectively of how that competence was achieved.**
>
> **The focus on the supervised nature of our tasks hence misses a key point of our work, namely that the supervised paradigm is simply our way of guaranteeing the theory requirement of competence at multiple tasks, and as such it is secondary to the key message above.** Scaling up supervised disentanglement efforts is not our suggestion for future research directions.
> ## Future directions
> Since our theory is agnostic to the way by which competence at multiple tasks is achieved, our work raises exciting applications in several areas of modern machine learning. As already mentioned in the manuscript (l. 1962-70), and the previous rebuttal to reviewer Mvb9 (section *Regarding the feasibility of disentanglement*), the most straightforward application is self-supervised learning, that relies on the prediction of multiple objectives which share a common latent space, and as such shares commonalities with our experimental approach here.
>
> Some actionable next steps spun by this work that we are actively pursuing are:
> * Extending our experiments beyond the supervised tasks considered here, to real-world datasets to see **how well the main conclusions of our theory hold in self-supervised settings**
> * **Investigating representations in pretrained language models, which are competent at many tasks**, and hence are likely to display disentanglement according to our theory
> * **Explaining emergence of disentangled reps in biological organisms naturally competent at many tasks**, and investigating more biologically plausible ways to induce it (e.g. continual learning)
>
> Of course we cannot expand too much on these as they are ongoing research. We are sure this work will spark more research directions, as it gets shared with the research community.
>
> **Our work also motivates the development of metrics that detect disentanglement without relying on the axis-alignment hypothesis, as now we have good reason to believe that capable models contain disentangled representations, just not necessarily at the individual unit level.** The same shift happened in neuroscience, where people switched from looking for single neuron coding of variables, to population coding.
>
> We expand more on reviewer Mvb9’s response at their [individual rebuttal below](https://openreview.net/forum?id=yVGGtsOgc7&noteId=QjrzaVKFN0) to avoid repeating things, including what our paper is about and what it isn’t, which we believe is not accurately reflected in the reviewer’s framing and questions.

---

### Author Response · Authors · 2024-11-22
**Main response (1/2)**

We thank the reviewers for the time and effort they put in reading the manuscript. The vast majority of reviewers recognized the contribution of the work, and their comments and suggestions have proven very useful in improving its presentation.

Here we summarize additions made in the revised manuscript, and clarify some aspects of our paper that were confusing for more than one reviewer.

## Summary of changes

In the updated version of our manuscript we:

* Rigorously quantify sparsity of our trained networks, showing that RNNs are consistently sparse across hyperparameter choices ($N_{task}$, $D$), similar to the brain (see **new Appendix A.12**). We also show that the number of active neurons is proportional to $D$, as long as there are enough tasks to uncover all latents ($N_{task} \geq D$) (**reviewers Mvb9** and **b5qB**).

* Demonstrate that our framework is robust to the choice of encoder nonlinearity, even handling non-injective observation maps (**new Appendix A.5**) (**reviewers 2Aef** and **Mvb9**)
* Expand upon the biological plausibility of multi-task learning (**Appendix A.14**) (**reviewer Mvb9**)
* Include a reproducibility statement
* Specify and clarify a lot of the pain points identified by the reviewers (e.g. description of Fig. 2).
* Summarize robustness of the theory under different noise distributions, readout functions and suboptimal classifiers in the new paragraph “Robustness” in section 3 of the main text
* Include another paragraph on extending the theory to non-linear boundaries in latent space in the theory section (section 3) (**reviewer xvfc**)
* Include a corollary in Appendix B (**Corollary B.7**) discussing extensions of the theory to sub-optimal multi-task classifiers (**reviewers b5qB** and **xvfc**)
* Move limitations to Appendix A.15 to make space in the main text.
* Increase the accuracy of our treatment of mutual information of the point estimator $\mu(t)$ and latent state $Z(t)$ with $x^*$ (l. 1550-3, 1617-9) (**reviewer  xvfc**)

All additions are marked with green font in the manuscript, for reviewer convenience. We also provide more experiments to address specific reviewer comments in the individual rebuttals. **Overall, we feel that we have addressed most reviewer comments, making this a compelling contribution for how autoregressive architectures can disentangle their representations, with provable guarantees.**

## Theoretical results and number of tasks

Our theory provides separate conditions for disentanglement and abstractness, which created some confusion (e.g. **reviewers w8Z5** and **Mvb9**). We show that linear identifiability is guaranteed when $N_{task} \geq D$ and orthogonality when $N_{task} \gg D$. Equations 3 and 4 hold under linear identifiability, i.e. when $N_{task} \geq D$ (condition 3 in Theorem 3.1 doesn’t have to hold). Empirically, we find that our networks, in particular transformers, readily disentangle when $N_{task} \geq D$, going beyond our theoretical guarantees. This might have caused some confusion to **reviewer Mvb9**, who mentions that an abundance of supervised signals is required, which is clearly not the case. Our networks require the minimum number of tasks that theoretically guarantee linear identifiability.


## Interplay between $N_{task}$ and $D$

Extending the above, having less tasks than latents is not a problem either. Our theory guarantees that the networks would disentangle the subspace in latent space that is spanned by the classification objectives. Therefore, as we note in the discussion, our work reveals an interplay between  $N_{task}$ and $D$; in a rich world where a lot of latents may exist, we are guaranteed to uncover the latents that are relevant for the tasks we are concerned with, as long as the subspace of these latents is spanned. If we would like to uncover more latents (i.e. understand the world in more detail), we would have to incorporate more tasks.

---

### Author Response · Authors · 2024-12-03
**Main contributions & implications**

We would like to thank reviewers for their feedback, and for finding our work interesting. For reviewer convenience, we summarize here some of the key contributions of our work, their implications, and potential future steps. This section acts synergistically with the more technical *Contributions* section in the main text.
## Main contribution
A main contribution of our work is that it **establishes a formal link between competence at many tasks, and the emergence of disentangled representations.** We first establish this argument theoretically, and then confirm and extend it experimentally. While our experiments are on supervised multi-task learning, **our theory is agnostic to the way by which competence was achieved**, and leaves the door open for other possibilities (e.g. self-supervised or unsupervised pre-training).
## Role of experiments
**Our supervised learning experiments are the most efficient way to confirm and extend our theoretical findings, and allow for greater tractability of results**, compared to unsupervised approaches. Note that scaling up supervised disentanglement efforts is **not** our suggestion for future research, due to scalability concerns. Instead, future work will endeavor to extend our results to **self-supervised learning**, which shares a lot of commonalities with our framework (i.e. making several predictions in parallel that all rely on a common latent ground truth; also see l. 1962-70 in the manuscript), and test how well our theoretical findings hold up in this setting.
## New research directions
Since our theory is agnostic to the way by which competence at multiple tasks is achieved, **a natural next step is to investigate whether disentangled representations exist in a wider range of models capable of solving multiple tasks**. A prime example is large language models that display excellent zero- and few-shot generalization capabilities, with some progress already made in that space. Moreover, the pre-training objective for LLMs (cross entropy loss/likelihood maximization) fits well within our theoretical framing on (approximately) optimal multi-classifiers.

**Our experiments showed parsimony of our theoretical results under conditions not covered by our theory**, including non-injective observation maps (Appendix A.5) and decision boundaries (Appendix A.10) which is encouraging for testing our findings on settings beyond what is strictly covered by our theory. Another direction we are actively pursuing is extending the theory to handle these more general cases (see “Discussion” on pages 36-38 of the manuscript).
## Need for better metrics
Our work also motivates new metrics that can **better detect** disentangled representations in existing networks. We argue in favor of a mixed representation approach to disentanglement (fig. 4), while most metrics to date look for axis-aligned representations. **Our work suggests that we might be missing disentanglement in networks that perform well in many tasks, due to the metrics used.**
## Impact for neuroscience
Our findings are also relevant for neuroscience, since the evidence integration tasks we use are canonical decision-making neuroscience tasks. Naturally, animals are capable of solving multiple tasks, thus our work provides theoretical justification for why disentangled representations have been found in many brain areas, and motivates looking for more. Note that we do not make strong specific claims about how competence at multiple tasks is achieved by organisms here.

**One hypothesis put forth by this work is that parallel computation may be happening**, where an animal may be executing only one task at a time but its predictions for previously encountered, similar tasks are also computed. Expanding upon this idea and other alternatives ways to achieve competence at multiple tasks in biological organisms (e.g. continual learning) will be a topic of future research. The potential benefits of this line of research are clear: **multi-task learning is extremely more efficient than previously proposed mechanisms for representation learning in the brain** (Mante et al, 2013), which is particularly relevant for organisms that support diverse computations within a limited neural substrate.
## Confidence and world models
Finally, our work emphasizes the importance of the confidence (i.e., calibrated likelihoods) of a network’s output, which is a recurring theme in machine learning (e.g. knowledge distillation etc.). **We here show that confidence fundamentally connects to how neural networks construct world models, either directly** (integrate-to-bound task in Appendix A.11) **or indirectly** (classification tasks). Under this framework, knowledge distillation can be cast as smaller models directly copying the world models (logits) of larger ones.

For more general implications of our work, we would like to point the reviewers to the discussion in the main text. Overall, we are very excited for the future of the field.

---

### Meta-Review · Area_Chair_kUzn · 2024-12-22

**Metareview:**

This paper explores the question of how abstract and disentangled representations can emerge in biological and artificial agents. The authors present both theoretical and experimental results demonstrating that multi-task learning, specifically within the framework of evidence aggregation classification tasks, can lead to the development of such representations.

Strengths: The paper introduces a theoretical framework connecting optimal multi-task classification to disentangled representations, building upon the empirical observations in Johnston and Fusi (2023). The paper provides a range of experiments exploring different architectures, task structures, and decision boundary geometries to validate these claims. The discussion of these results in the context of contemporary neuroscience literature is also well framed and places this paper in a good position to appeal to both the broader neuroscience and task-learning communities.
Weaknesses: The conditions for the theoretical results are rather strong, which may limit their practical relevance.  The experiments focus on synthetic data with simple latent structures. While these experiments effectively support the theoretical results, testing the framework on more complex data could provide further insights.

Overall, while the theory and experiments have limitations,  the paper still contributes meaningfully  to the understanding of disentangled representations and how they might arise. I think it will be valuable for the community and recommend accepting it.

**Additional Comments On Reviewer Discussion:**

There was an extended and somewhat heated discussion between the authors reviewer Mvb9 who insists that using supervision for disentanglement is the wrong direction and thus dismisses the very premise of the paper. I agree with the authors that misses the point of their work and am disregarding the scoring fo Mvb9 in my evaluation.

---

### Decision · Program_Chairs · 2025-01-22

Accept (Poster)